# The motive cocktail in altruistic behaviors

Xiaoyan Wu[1,2,3,11], Xiangjuan Ren [4,5,6,11], Chao Liu [1,2,3] & Hang Zhang [4,7,8,9,10]

Prosocial motives such as social equality and efficiency are key to altruistic behaviors. However, predicting the range of altruistic behaviors in varying contexts and individuals proves challenging if we limit ourselves to one or two motives. Here we demonstrate the numerous, interdependent motives in altruistic behaviors and the possibility to disentangle them through behavioral experimental data and computational modeling. In one laboratory experiment ($N = 157$) and one preregistered online replication ($N = 1,258$), across 100 different situations, we found that both third-party punishment and third-party helping behaviors (that is, an unaffected individual punishes the transgressor or helps the victim) aligned best with a model of seven socioeconomic motives, referred to as a motive cocktail. For instance, the inequality discounting motives imply that individuals, when confronted with costly interventions, behave as if the inequality between others barely exists. The motive cocktail model also provides a unified explanation for the differences in intervention willingness between second parties (victims) and third parties, and between punishment and helping.

Many people voluntarily provide resources such as shelter, food and healthcare to refugees fleeing war-torn regions, while others advocate sanctioning responsible nations, even at personal expense. This altruistic behavior, known as third-party punishment (3PP) and helping (3PH), involves sacrificing personal interests to punish transgressors or help victims. Such behaviors have been observed in both laboratory[1–3] and field studies[4,5]. What, then, motivates these actions?

According to one line of theories, third-party intervention serves as a strategic means to obtain future rewards, by signaling one's trustworthiness to potential cooperators[3,6] or deterring potential transgressors from harming oneself or valued others[7]. However, third-party intervention in one-shot, anonymous scenarios[1] aligns more with the strong-reciprocity theory[8], where individuals may reward cooperation, punish non-cooperation or more generally sanction violations of social norms[9,10], even without prospect of personal gain. These two lines of theories are not necessarily conflicting; the motives for sanctioning norm violations can be viewed as internalized external motivations. A widely observed norm in human societies is egalitarian distribution. By quantifying inequality—a violation of this norm—as a loss in a utility maximization framework, Fehr and Schmidt[11] provide a unified explanation for various socioeconomic phenomena, including altruistic punishment and helping behaviors[1,12,13]. Human representation of inequality is further supported by neuroimaging studies[12,14,15].

The power of this normative framework[1] lies in its potential to integrate different motives into one utility measure to address the complexity of human altruistic behaviors. However, this potential is far from thoroughly explored, because most previous studies only focused on one or two motives (other than self-interest, SI) and often contrasted models with distinctive motives[13,16], as if human behaviors were guided exclusively by one of the alternative motives at each moment. Such practice makes it difficult to unify the knowledge gained from different studies that examine different motives. Furthermore,

[1]State Key Laboratory of Cognitive Neuroscience and Learning & IDG/McGovern Institute for Brain Research, Beijing Normal University, Beijing, China. [2]Beijing Key Laboratory of Brain Imaging and Connectomics, Beijing Normal University, Beijing, China. [3]Center for Collaboration and Innovation in Brain and Learning Sciences, Beijing Normal University, Beijing, China. [4]School of Psychological and Cognitive Sciences and Beijing Key Laboratory of Behavior and Mental Health, Peking University, Beijing, China. [5]Max Planck Institute for Human Development, Berlin, Germany. [6]Institute of Psychology, Universität Hamburg, Hamburg, Germany. [7]PKU–IDG/McGovern Institute for Brain Research, Peking University, Beijing, China. [8]Peking–Tsinghua Center for Life Sciences, Beijing, China. [9]State Key Laboratory of General Artificial Intelligence, Peking University, Beijing, China. [10]Chinese Institute for Brain Research, Beijing, China. [11]These authors contributed equally: Xiaoyan Wu, Xiangjuan Ren. ✉e-mail: liuchao@bnu.edu.cn; hang.zhang@pku.edu.cn

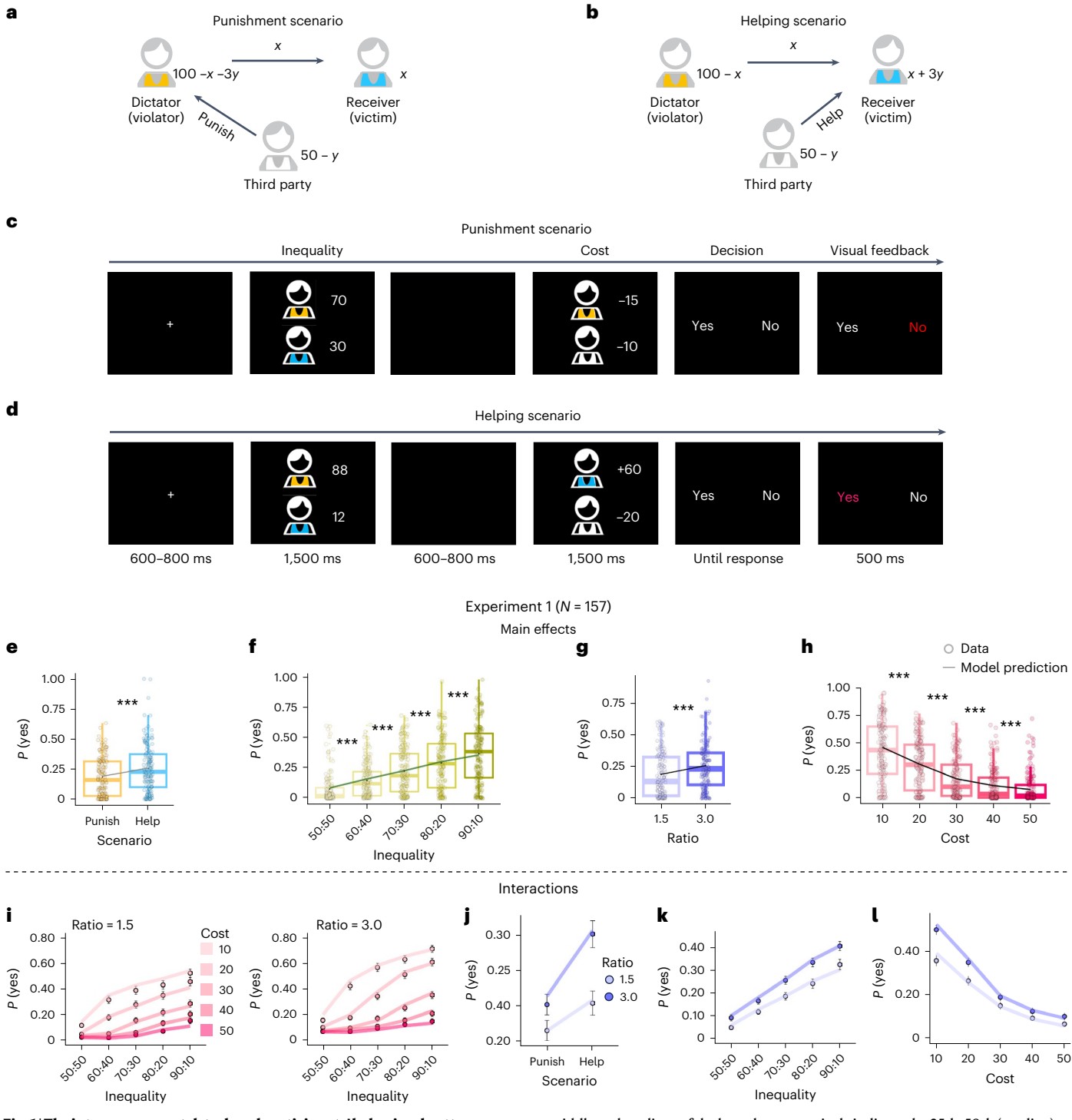

**Fig. 1 | The intervene-or-watch task and participants' behavioral patterns.**
**a,b**, Schema of the intervene-or-watch task for the punishment (**a**) and helping
(**b**) scenarios. **c,d**, Time course of a trial for the punishment (**c**) and helping (**d**)
scenarios. In each trial, participants first saw the outcome of a dictator game—out
of 100 tokens how much the dictator (transgressor, cartoon figure in orange shirt)
allocated to themselves and to the receiver (victim, blue shirt): 70 versus 30 (**c**)
or 88 versus 12 (**d**). As a third party starting with 50 tokens, participants (white
shirt) were provided with an intervention offer, such as spending 10 of their own
tokens to reduce the transgressor's payoff by 15 tokens (**c**) or spending 20 of their
own tokens to increase the victim's payoff by 60 tokens (**d**). The participants' task
was to decide whether to accept the intervention offer (press 'yes') or do nothing
(press 'no'). **e–h**, Main effects of scenario (**e**), transgressor–victim inequality (**f**),
impact-to-cost ratio (**g**) and intervention cost (**h**) on the probability of accepting the
intervention offer, $P$(yes). Each filled circle denotes one participant. The bottom,
middle and top lines of the box plot respectively indicate the 25th, 50th (median)
and 75th percentiles. The whiskers extend to the minima and maxima within
1.5 times the range between the 25th and 75th percentiles from the bottom and
top bounds of the box plot. The black dot inside each box denotes the group mean.
\*\*\**P* < 0.001 for the difference between adjacent conditions from Bonferroni-
corrected post hoc comparison (see statistical details in Supplementary Section 3).
The line superimposed on the boxes denotes the prediction of the best-fitting
model (that is, the seven-motive motive cocktail model, described later).
**i–l**, Interaction effects on $P$(yes), including an inequality × cost × ratio three-way
interaction (**i**) and two-way interactions of scenario × ratio (**j**), inequality × ratio (**k**)
and cost × ratio (**l**). Each circle denotes the mean across participants ($N$ = 157).
Error bars denote s.e.m. As in **e–h**, the lines denote the predictions of the best-
fitting model. Credit: **a–d**, head icon, X. Mai.

it limits the power of the normative framework to explain intricate behavioral patterns.

For example, when a victim seeks revenge against the transgressor, a trade-off between SI and inequality reduction would predict either no punishment or full punishment to restore equality, depending on whether the impact ratio of the punishment is below or above a certain threshold (Supplementary Fig. 1). However, people often choose to punish the transgressor without fully restoring equality[1], which some researchers explain by resorting to a separate personal tendency called 'willingness to punish'[12], a factor not motivated by socioeconomic utilities. The hesitation of previous studies to simultaneously test multiple motives may be partly due to limitations in their experimental designs[17], where different motives often yield similar predictions[18], making them empirically indistinguishable. However, practices from relatively developed modeling-reliant fields such as human decision-making[19] and working memory[20,21] suggest that including multiple motives in one model and empirically teasing them apart are both plausible and valuable for advancing our understanding of human altruistic behaviors.

In this Article we aimed to extend the normative framework of utility maximization to provide a unified explanation for a wider range of phenomena in altruistic behaviors. We constructed a series of computational models assuming that altruistic behaviors are driven jointly by multiple socioeconomic motives. These 'motive cocktail' models cover a comprehensive set of socioeconomic motives. Five of the motives are based on established theories from the literature, including two variants of self-centered inequality (SCI)[1,13], victim-centered inequality (VCI)[13], efficiency concern (EC)[14,22] and reversal preference (RP)[23,24]. While some of the established socioeconomic motives are qualitatively similar, they lead to different quantitative patterns and can thus be distinguished through computational modeling. Furthermore, we also identified two new 'compound' motives that are nonlinear combinations of more elementary motives.

To separate the effects of different socioeconomic motives, we need an experimental set-up that can systematically vary all the motives in the same context. We thus designed a third-party intervention task—the intervene-or-watch task (Fig. 1a,b), which enables an unusually rich set of experimental conditions for testing this variety of motives that would otherwise be indistinguishable. In each trial (Fig. 1c,d), participants saw the outcomes from a dictator game, where the dictator ('transgressor') allocated more to themselves than to the receiver ('victim', for example, 88 versus 12 tokens). As the unaffected third party, participants received 50 tokens in each trial and were offered an opportunity to intervene, such as spending 10 tokens (intervention cost) to reduce the transgressor's payoff by 15 tokens (impact ratio = 15/10 = 1.5). Participants decided whether to accept this intervention offer or to keep all 50 tokens to themselves. Each participant completed 300 trials in 100 different conditions that varied in the transgressor–victim inequality as well as the scenario (punishment versus helping), the cost and the impact-to-cost ratio of the intervention offer.

We performed one laboratory experiment ($N = 157$) and a preregistered online experiment ($N = 1,258$), with all major findings of the former replicated in the latter. A three-way interaction of inequality × cost × impact ratio found in participants' intervention decisions suggests utility calculations that go beyond linear combinations of different motives. Indeed, participants' behavioral patterns were best fit by a motive cocktail model whose utility calculation involves seven socioeconomic motives, including two compound motives. We called the compound motives 'inequality discounting' (ID), which refers to people's tendency to behave as if they are underestimating the inequality between others as the intervention cost increases. Individuals' cocktail motives fall into three groups: 'justice warriors', who have a strong intention to intervene whenever there is inequality, 'pragmatic helpers', who are sensitive to the impact of their intervention to help the victim, and 'rational moralists', who seek to achieve an acceptable standard of morality at the lowest cost to SI. Our model provides a

unified explanation for phenomena beyond 3PP and 3PH, such as why interveners spend more to penalize transgressors when they themselves are victims rather than unaffected third parties[1,12].

## Results

Each trial was either in a punishment scenario (as in the example above, Fig. 1a,c) or in a helping scenario (to increase the victim's payoff, Fig. 1b,d). The inequality between the transgressor and the victim (50:50, 60:40, 70:30, 80:20 or 90:10, with ±2 jitters), the intervention cost (10, 20, 30, 40 or 50) and the impact ratio (1.5 or 3.0) were also varied across trials. Each participant completed 300 trials (5 inequality levels × 5 cost levels × 2 impact ratios × 2 scenarios × 3 repetitions) of intervention decisions.

### Behavioral patterns in 3PP and 3PH

In experiment 1, there were 157 participants (all students). We first performed a generalized linear mixed model analysis (GLMM1, see Supplementary Table 1) on participants' decisions (to intervene or not) to assess the effects of each independent variable and their interactions. We found intriguing interaction effects as well as classic 3PP and 3PH behavioral effects.

**Preference for helping over punishment.** Consistent with most previous studies, participants had a higher probability to help the victim ($M = 0.25$) than to punish the transgressor ($M = 0.18$, $b$ of scenario = −1.22, 95% confidence interval (CI) [−1.64, −0.80], $P < 0.001$; Fig. 1e).

**Inequality aversion and rationality.** As we would expect from inequality aversion, participants were more willing to intervene when the transgressor–victim inequality was more extreme ($b$ (regression coefficient) = 1.61, 95% CI [1.40, 1.81], $P < 0.001$; Fig. 1f) and when the impact-to-cost ratio was higher, that is, when the same cost yielded a greater reduction in inequality ($b = 0.82$, 95% CI [0.62, 1.01], $P < 0.001$; Fig. 1g). Meanwhile, participants were also rational decision-makers who cared about their own interests, being less willing to intervene under a higher cost of intervention ($b = −2.12$, 95% CI [−2.37, −1.86], $P < 0.001$; Fig. 1h).

**Interaction effects.** Thanks to our factorial experimental design with four dimensions and 100 conditions, we also identified three two-way and one three-way interaction effects that had been seldom documented before. Under a higher impact-to-cost ratio, the preference for helping over punishment was stronger (scenario × ratio interaction: $b = −0.39$, 95% CI [−0.47, −0.30], $P < 0.001$; Fig. 1j), and the probability of intervention changed more markedly with the transgressor–victim inequality (inequality × ratio interaction: $b = −0.08$, 95% CI [−0.14, −0.02], $P = 0.017$; Fig. 1k) and with cost (cost × ratio interaction: $b = −0.08$, 95% CI [−0.14, −0.02], $P = 0.015$; Fig. 1l). According to the three-way interaction of inequality × cost × ratio ($b = −0.21$, 95% CI [−0.27, −0.15], $P < 0.001$), a higher ratio also led to a stronger modulation of the intervention cost with participants' sensitivity to inequality (Fig. 1i).

### Seven socioeconomic motives and their hypothetical effects

What socioeconomic motives may have driven the observed 3PP and 3PH behaviors? Besides SI (the core of classical economic models), we considered five classes of computationally well-defined socioeconomic motives (Fig. 2a), which expand into seven motive terms in utility calculation (see Supplementary Table 2 and 3 for examples in fictitious characters and real-life scenarios). Five of these motives are adapted from the literature, including three variants of inequality aversion[1,13], EC[14,16] and RP[23,24]. The remaining two motives, under the class of ID, are defined here to capture the interaction between SI and inequality aversion. They are partly motivated by the observed interaction effect that under higher intervention cost the participants' probability of

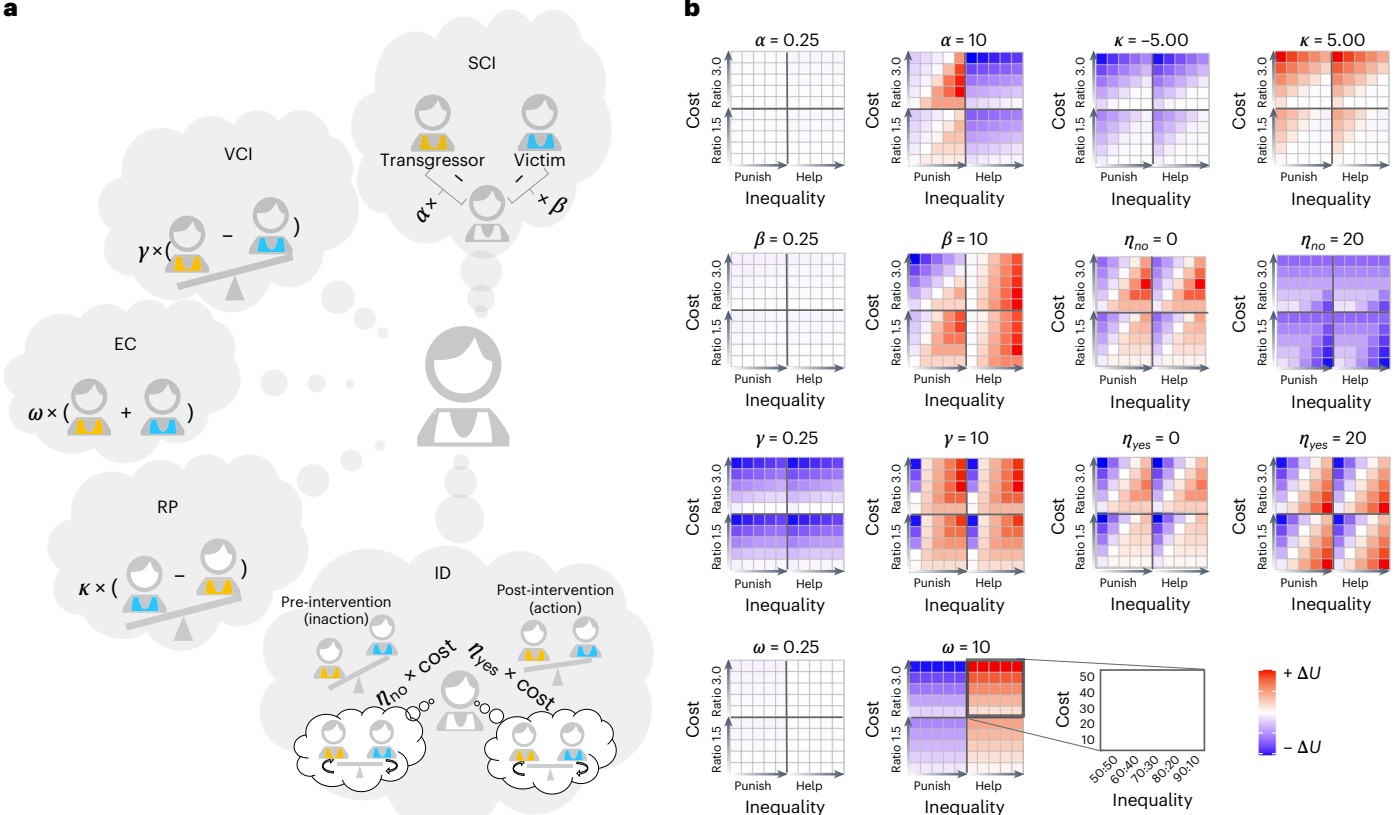

**Fig. 2 | The seven socioeconomic motives and their hypothetical effects on the third party's utility gain to intervene. a,** Five classes of computationally well-defined socioeconomic motives that expand into seven motive terms in utility calculation. Parameters $\alpha$ and $\beta$ control disadvantageous (self < other) and advantageous (self > other) inequality aversion, respectively. This illustration of disadvantageous SCI between self and transgressor but advantageous SCI between self and victim may not apply to post-intervention inequality, where the direction of SCI might be reversed. The SCI type only depends on whether self > other or self < other, regardless of the other being transgressor or victim. Parameter $\gamma$ controls victim-centered disadvantageous (victim < transgressor) inequality aversion. Parameter $\kappa$ controls the direction and strength of the RP motive (victim > transgressor after intervention). Parameter $\omega$ controls EC (maximizing others' total payoff). Parameters $\eta_{no}$ and $\eta_{yes}$ respectively control inaction and action ID (attenuated perception of inequality under higher intervention cost). **b,** Heatmaps illustrating how each motive's strength

influences $\Delta U$ (utility of choosing yes − utility of choosing no) in the third-party intervention decision. Each motive is shown by a pair of panels with the small and large parameters controlling the motive's magnitude differently. For simplicity, when the effect of a single parameter is examined, all other parameters are set to zero. The exceptions are $\eta_{no}$ and $\eta_{yes}$, for which parameter $\gamma$ is set to 1, because their utility terms are multiplied by $\gamma$. Each heatmap has four submaps: divided horizontally by scenario (punishment left, helping right) and vertically by impact ratio (1.5 bottom, 3.0 top). The x axis denotes inequality severity (near equality left to extreme inequality right), and the y axis denotes intervention cost (low bottom to high top). Color code, $\Delta U$: reddish for stronger preference to choose yes, bluish for stronger preference to choose no. For illustration purposes, the $\Delta U$ were scaled separately for each column and separately for positive and negative values. Each motive shows a distinct influence on $\Delta U$ and would thus lead to distinguishable effects on third-party intervention decision behaviors. Credit: **a,** head icon, X. Mai.

intervention not only was lower, but also increased more slowly with the transgressor–victim inequality (Fig. 1i). As unfolded below, each motive affects the utility gain from intervention relative to non-intervention (thus the tendency to intervene) in a different way (Fig. 2b).

SCI refers to the payoff difference between self and others[1]. It can be further divided into disadvantageous inequality (self < other) and advantageous inequality (self > other), controlled by parameters $\alpha$ and $\beta$ respectively. The parameter $\alpha$ implies stronger aversion to receiving lower payoff than others (for instance, self 50 versus transgressor 88), while $\beta$ implies a stronger aversion to receiving higher payoff than others (self 50 versus victim 12). Before intervention, participants had lower payoff than the transgressor but higher payoff than the victim. As the result, higher $\alpha$ motivates penalizing the transgressor to reduce disadvantageous inequality, but discourages helping the victim as it increases disadvantageous inequality with the transgressor and may create disadvantageous inequality with the victim (Fig. 2b, row 1 left pair). In contrast, higher $\beta$ motivates intervention in both the punishment and helping scenarios, unless greater punishment leads to an undesirable advantageous inequality over the transgressor (Fig. 2b, row 2 left pair).

VCI refers to the payoff difference between the transgressor and the victim[13]. This inequality aversion variant implies that participants dislike the higher payoff of the transgressor over the victim. Participants with larger $\gamma$ intervene more in most punishment and helping scenarios (Fig. 2b, row 3 left pair), unless the victim-centered disadvantageous inequality is too small (for instance, transgressor 51 versus victim 49) to compensate for intervention costs.

EC, a motive used frequently for modeling economic games[14,16] but seldom for 3PP or 3PH, assumes that people care about others' overall welfare, such as the sum of the transgressor's and the victim's payoffs in our case. Participants with larger $\omega$ are more likely to help the victim to increase the overall welfare, but less likely to penalize the transgressor to avoid reducing the overall welfare, regardless of the inequality between others (Fig. 2b, row 4 left pair).

RP refers to the motive that participants intend to reverse the payoff difference between the transgressor and the victim, rewarded by their payoff difference in the opposite direction (that is, after intervention the victim would be better off than the transgressor). The parameter $\kappa$ controlling RP can be positive or negative, implying

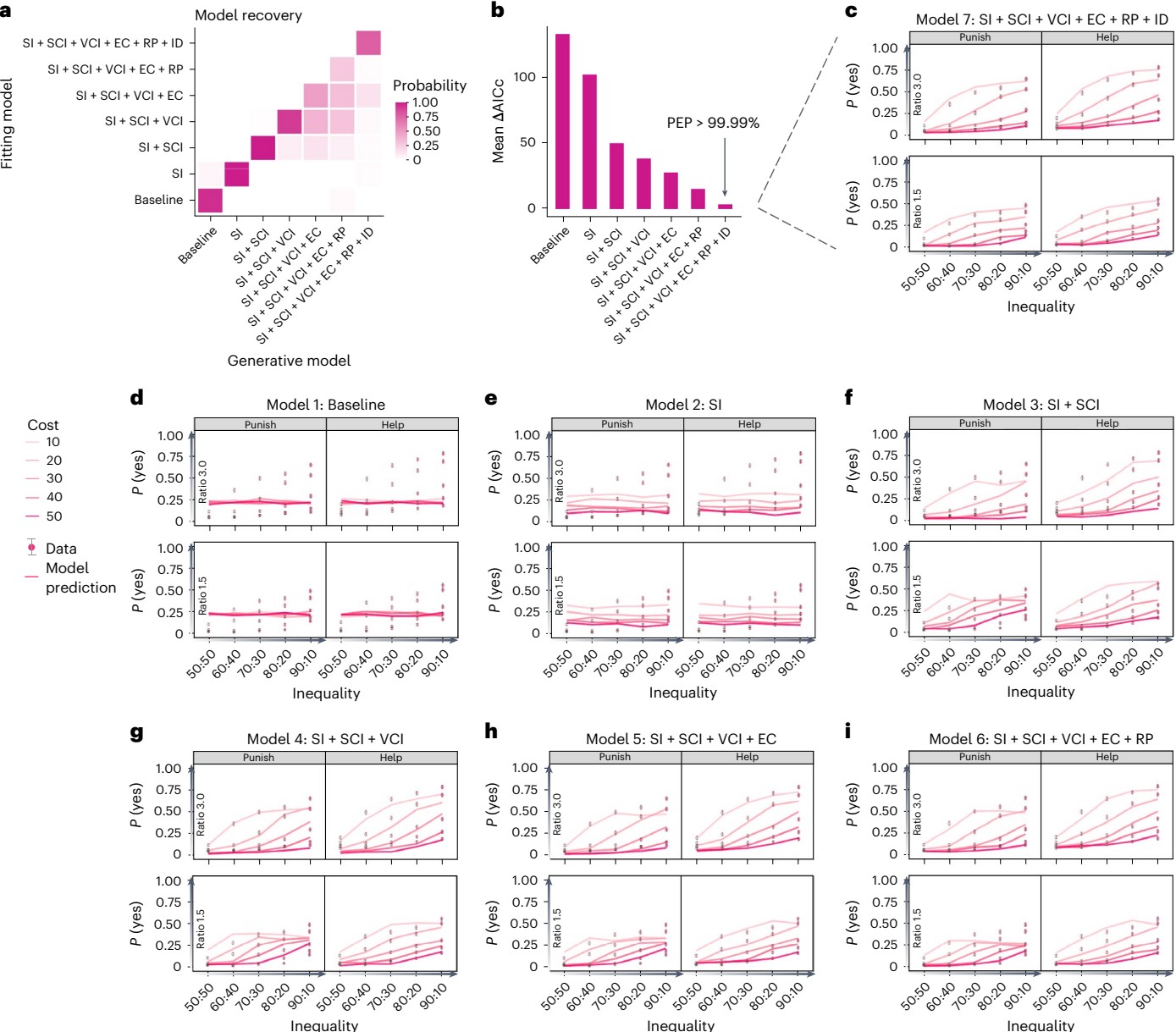

**Fig. 3 | Modeling results of the seven-motive motive cocktail model compared with alternative models. a**, Model recovery analysis. Each model was used to generate 100 synthetic datasets, for each of which model fitting and comparison were performed. Each column is for one generative model. Each row is for one fitting model. The color in each cell codes the probability that the synthetic datasets from the generative model in the column are best fit by the fitting model in the row, with darker color indicating higher probability. **b**, Model comparison results. For each participant, the model with the lowest AICc was used as a reference to compute ΔAICc by subtracting it from the AICc of the other models (ΔAICc = AICc − AICc$_{lowest}$). Lower ΔAICc indicates better fit. The PEP of a model

is a group-level measure of the likelihood that the model outperforms all other models. The name of a model (for instance, SI + SCI) conveys the motives included in its utility calculation. **c**–**i**, Separate data versus model predictions for the seven models compared in **b**. The title of each panel indicates the model name. The probability of intervention, *P*(yes), is plotted against the inequality (from 50:50 to 90:10). Different colors code different levels of intervention cost (from 10 to 50; darker color for higher cost). Each subpanel corresponds to one scenario and impact-ratio condition. The circles and error bars respectively denote the mean and s.e.m. across participants (*N* = 157). The solid lines denote the predictions of the models.

willingness or reluctance to reverse others' economic status, making the term a generalized form of rank reversal aversion[23,24]. Individuals with more positive *κ* are more willing to punish or help when the impact (cost × ratio) is large enough (relative to the inequality) to yield a rank reversal between the transgressor and the victim (Fig. 2b, row 1 right pair).

ID refers to people's tendency to behave as if they are underestimating the inequality between others as the intervention cost increases. We defined two types of ID motive: inaction ID (controlled by $\eta_{no}$) and action ID (controlled by $\eta_{yes}$), representing diminished

awareness of inequality when choosing not to intervene and when opting to intervene, respectively. ID motives are compounds that are not just the lack of motivation to reduce inequality as characterized by smaller *γ* (VCI), but capture the modulation of SI on VCI in both directions. Participants are less likely to intervene when they have larger $\eta_{no}$, which differs from smaller *γ* in that it may cause no intervention even when transgressor–victim inequality is high (Fig. 2b, row 2 right pair). Conversely, participants with larger $\eta_{yes}$ are more likely to intervene, as if they believe inequality is always minimized following a costly intervention (Fig. 2b, row 3 right pair).

Many of these motives would remain unidentifiable in a task involving only two parties, testing exclusively either punishment or helping scenarios, or lacking variation in cost or impact ratio. However, in our intervene-or-watch task, the seven motives forecast unique effects on intervention decisions, thus making them distinguishable in behavioral data. Subsequent modeling analysis validated each parameter's discernibility, even under simultaneous modeling (Methods and Supplementary Fig. 2).

### The motive cocktail model best predicts human behaviors

We assessed the seven socioeconomic motives' contribution to altruistic behavior by incrementally incorporating them into utility calculations, creating a series of increasingly complex computational models. The introduction of different motives follows a descending order depending on how central and established a specific motive is in the literature of 3PP and 3PH. We then compared these models' predictive power for the behavioral patterns observed in experiment 1. This solution-oriented approach is similar to the idea of 'quasicomprehensive exploration' introduced by a recent study on spatial working memory[20]. Starting from a baseline coin-flipping model, which intervened at a fixed probability, and an SI model, we introduced five motive classes as utility terms in the following order: SCI, VCI, EC, RP and ID. This process yielded seven different models (Methods) with different predictions (Fig. 3). We used maximum-likelihood estimation to fit each model to individual participants' decisions, and the corrected Akaike information criterion (AICc)[25] to evaluate each model's relative goodness of fit, accounting for complexity. We also computed the protected exceedance probability (PEP)[26] to provide a group-level measure that a model outperforms others.

The full motive cocktail model that includes all the motives best predicted participants' decisions (lowest AICc, PEP > 99.99% among the seven models). A model recovery analysis (Methods) further confirmed that the best performance of the full model was real and could not be attributed to model misidentification: among the 700 synthetic datasets generated by the six alternative models, none was misidentified as the full model (Fig. 3a). Integrating each motive class (SI, SCI, VCI, EC, RP and ID) into our models led to considerable improvements in their fits (as indicated by lower AICc values in Fig. 3b).

The full model closely mirrored changes in participants' intervention probabilities across the 100 experimental conditions (Fig. 3c), successfully predicting the main and interaction effects of different variables (lines in Fig. 1e–l). In contrast, alternative models failed to replicate certain patterns within the data (Fig. 3d–i). A supplementary analysis that compared more model variants further demonstrated the necessity of the ID assumption (the interaction items) in the full model as well as the nonlinear modulation of SI on the VCI (Supplementary Fig. 3) in fitting the behavioral data. The ID term follows the form of a sigmoid function (Supplementary Fig. 4b), which has the desired mathematical property of ensuring that its value is between 0 and 1.

To conclude, participants' third-party intervention decisions were jointly driven by SI and the seven socioeconomic motives, including the two ID terms.

### Justice warriors, pragmatic helpers and rational moralists

Our intervene-or-watch task, with its 100 factorially designed conditions, yielded a multifaceted profile that captured not only the collective behavioral tendencies but also the nuanced 3PP and 3PH behaviors of individual participants. A clustering analysis of the behavioral patterns of the 157 participants revealed that they were best summarized by three distinct clusters (Methods and Fig. 4a,b). Among them, the justice warriors (35% of participants) had an overall high probability to intervene, especially when the transgressor–victim inequality was high and the cost was relatively low (Fig. 4j). The pragmatic helpers (18%) also had a high probability to intervene, but were insensitive to inequality or cost, and preferred helping over punishment (Fig. 4k). The rational moralists (47%) barely intervened unless their intervention cost was minimal (Fig. 4l). The full motive cocktail model accurately predicted not only the average behavior (Fig. 4i) but also the behavioral patterns specific to each individual cluster (Fig. 4j–l).

These marked individual differences were associated with different combinations of motive parameters (Fig. 4c–e). Kruskal–Wallis tests with Bonferroni correction revealed significant differences across the three clusters for three out of the seven motive parameters (Fig. 4f–h and Supplementary Fig. 5): action ID $\eta_{yes}$ ($H(2) = 22.18$, $P < 0.001$, with $H(2)$ denoting the $X^2$ statistic with two degrees of freedom), RP $\kappa$ ($H(2) = 15.57$, $P < 0.001$) and inaction ID $\eta_{no}$ ($H(2) = 9.71$, $P = 0.008$). The highest values of $\eta_{yes}$, $\kappa$ and $\eta_{no}$ respectively occurred for justice warriors, pragmatic helpers and rational moralists. To unravel the relationship of these parameters with the observed individual differences, we carried out a series of correlation analyses between individuals' parameter values and their sensitivities to different variables at the group level (multiple comparisons corrected for each parameter using false discovery rate; Supplementary Fig. 6), where a participant's sensitivity to a variable was defined as the normalized intervention probability difference after the corresponding variable was dichotomized. The observed behavioral differences across clusters coincide with the correlational effects of these parameters (Fig. 4m–r) and agreed with the insights we obtained through simulation (Fig. 2). For example, higher $\eta_{yes}$ implies increased tendency to perceive one's action as effective in reducing inequality, irrespective of the actual impact, when the intervention cost is high. Indeed, individuals with higher $\eta_{yes}$ were less sensitive to the impact ratio. Justice warriors, those who had the highest $\eta_{yes}$ among the three clusters, were least sensitive to the impact ratio (Fig. 4n).

### Replication in a preregistered, large online experiment

To test whether our findings can be generalized to a large population with different cultural backgrounds, we performed a preregistered,

**Fig. 4 | Three types of 3PP and 3PH behavior: justice warriors, pragmatic helpers and rational moralists. a**, Illustration of the three behavioral types. **b**, The *k*-means clustering performance of behavioral patterns was best for three clusters. Higher silhouette value indicates larger ratio of between-cluster to within-cluster distance. **c–e**, The median value of motive parameters for each cluster. The outer contour of the spider plot indicates the maximal normalized parameter value. **f–h**, Action ID $\eta_{yes}$ (**f**), RP $\kappa$ (**g**) and inaction ID $\eta_{no}$ (**h**) parameters compared across clusters. The highest values of $\eta_{yes}$, $\kappa$ and $\eta_{no}$ respectively occurred for justice warriors (J, $N = 55$), pragmatic helpers (P, $N = 28$) and rational moralists (R, $N = 74$). Conventions follow Fig. 1e–h. *$0.01 \leq P < 0.05$, **$0.001 \leq P < 0.01$, ***$P < 0.001$. Pairwise comparison results were from two-tailed post hoc comparisons following Kruskal–Wallis tests, Bonferroni corrected (see Supplementary Section 3 for statistical details). **i–l**, Intervention probability $P$(yes) in 100 conditions for all participants (**i**) and each cluster (**j–l**), with data (top) versus motive cocktail model predictions (bottom). Heatmaps arranged as in Fig. 2b; darker colors indicate higher $P$(yes). **m–r**, The three parameters ($\eta_{yes}$, $\kappa$ and $\eta_{no}$) contribute to the behavioral differences across clusters. Each panel is for one main or interaction effect (as in Fig. 1), with the bar height denoting the effect size in each cluster. Arrows indicate significant correlations between parameters and behavioral measures, and how parameters modulate behavioral measures (arrow orientation) at the group level (Supplementary Fig. 6). For example, panel **m** shows that higher $\kappa$ and higher $\eta_{no}$ were respectively associated with higher and lower overall $P$(yes), which coincides with the high $P$(yes) observed in pragmatic helpers (**k**) and low $P$(yes) in rational moralists (**l**). Sensitivity for a variable was calculated as the normalized intervention probability difference between high and low conditions. 'Low inequality' refers to 60:40 and 50:50; 'high inequality' refers to 90:10, 80:20 and 70:30. 'Low cost' and 'high cost' refer to cost ≤ 20 and cost > 20, respectively. 'Low ratio' and 'high ratio' refer to impact ratios of 1.5 and 3, respectively. Credit: **a**, head icon, X. Mai.

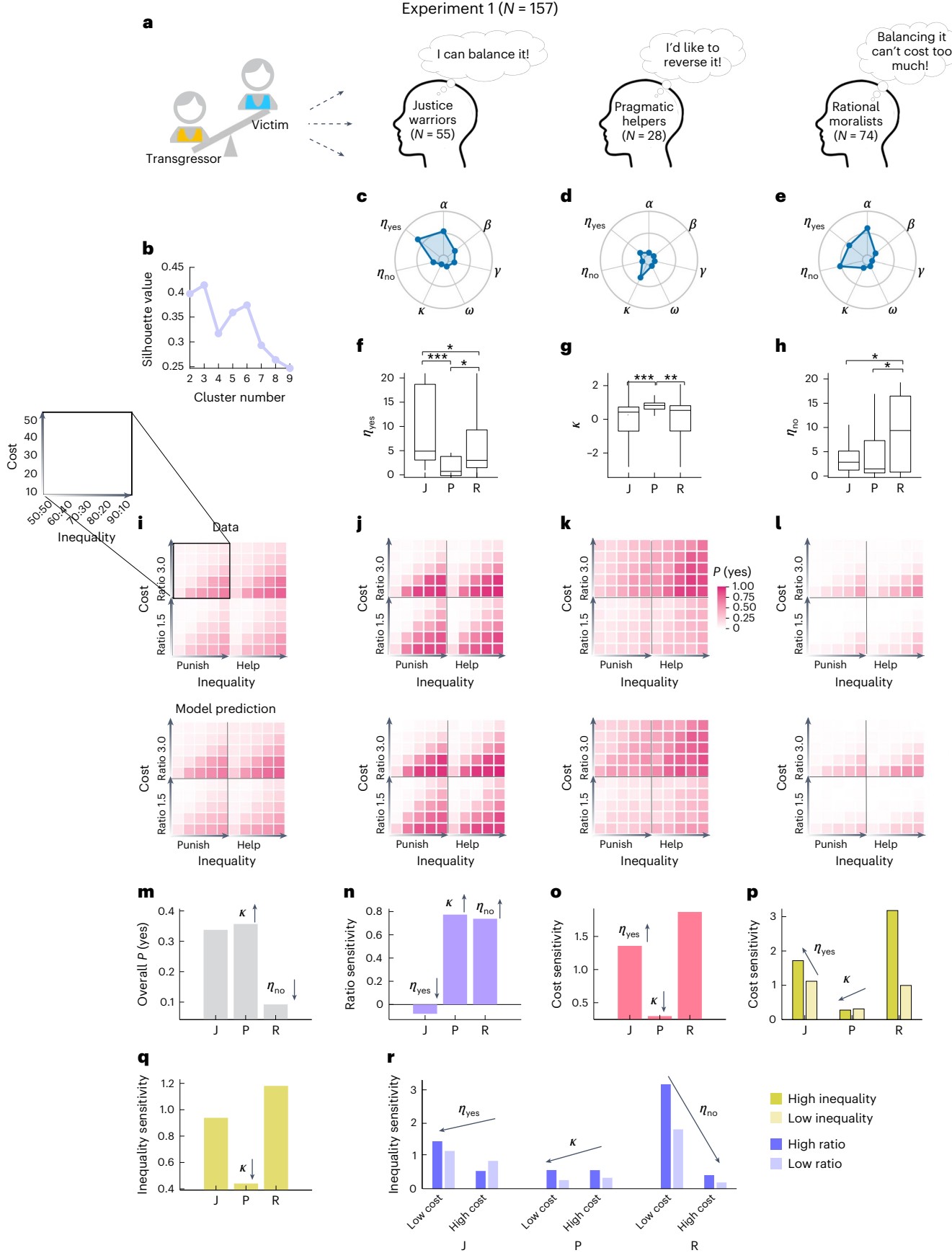

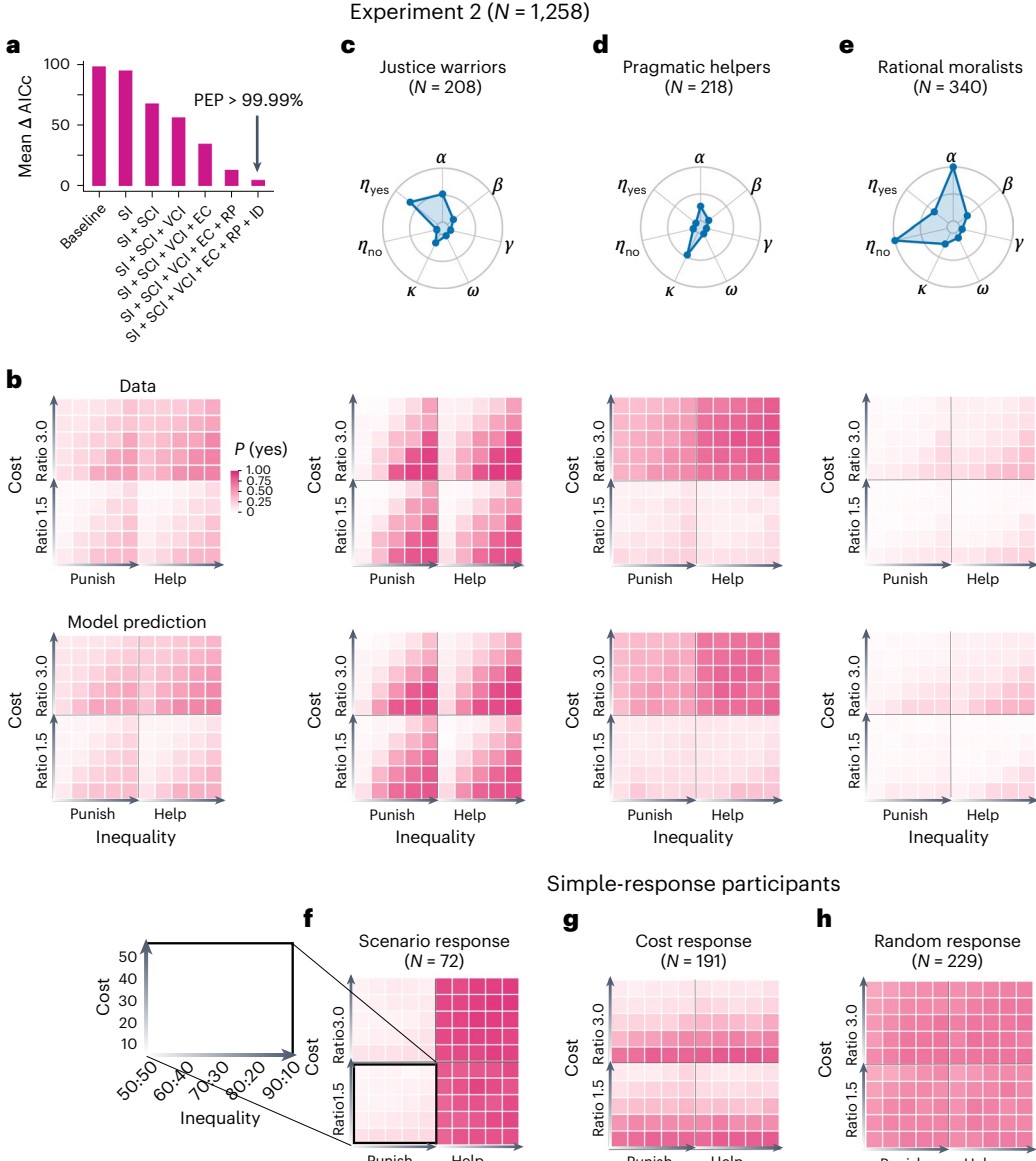

**Fig. 5 | Major findings in the preregistered, large-scale online experiment 2. a**, Model comparison results. As in experiment 1, the full motive cocktail model best fit participants' decision behaviors, as indicated by the lowest ΔAICc and a PEP over 99.9%. **b**, Data versus model prediction. As in experiment 1, the full model can accurately predict not only participants' average behaviors ($N = 1,258$), but also that of individual clusters (justice warriors, $N = 208$; pragmatic helpers, $N = 218$; rational moralists, $N = 340$). **c–e**, The median value of the motive parameters for the first three clusters. These three clusters had behavioral patterns and parameter combinations similar to those of the justice warriors, pragmatic helpers and rational moralists identified in experiment 1. **f–h**, Data

for the three additional clusters observed in experiment 2. These three clusters were best fit by a simple-response model (model 9) instead of by the motive cocktail model. **f**, The scenario response cluster ($N = 72$), where participants varied their choices only with the scenario, consistently choosing 'yes' for the helping scenario but 'no' for the punishment scenario. **g**, The cost response cluster ($N = 191$), where participants varied their choices only with the cost of intervention. **h**, The random response cluster ($N = 229$), where participants seemed to choose randomly, without responding to any variables. These patterns are clues to low effort or less engaged participation, which is more frequent among online participants. Conventions follow Fig. 4.

large-scale online experiment using the same experimental procedures, with 1,258 participants (all students, sample size predetermined on the basis of a model-based power analysis, Supplementary Fig. 7) from over 60 countries (or regions, Supplementary Table 4). All major statistical and modeling findings of experiment 1 were replicated in experiment 2 (Fig. 5; see Supplementary Table 5 for the GLMM results).

As in experiment 1, the full motive cocktail model outperformed the other models and accurately captured the behavioral patterns in experiment 2 (Fig. 5a,b; see Supplementary Fig. 8 for model recovery analysis). The behavioral patterns of the 1,258 participants were best captured by six clusters (Supplementary Fig. 9), in which the first three clusters agreed with those in experiment 1—justice warriors (16.60%, Fig. 5c),

pragmatic helpers (17.30%, Fig. 5d) and rational moralists (27.00%, Fig. 5e). As in experiment 1, each of these three clusters was best fit by the full motive cocktail model (or its derivatives; Supplementary Fig. 9b). The remaining three clusters of participants (39.10%, Fig. 5f–h) seemed to respond to one single stimulus dimension (for instance, always help but seldom punish) or even purely randomly; these choice behaviors were best described by a simple-response model that linearly combines different independent variables (Methods and Supplementary Fig. 9b). These choice patterns likely resulted from these participants' less engaged participation (lower attention check accuracy than participants in the first three clusters: $t(1,256) = -9.78$, $P < 0.001$), which is more common in online settings, rather than representing real-world behavioral patterns.

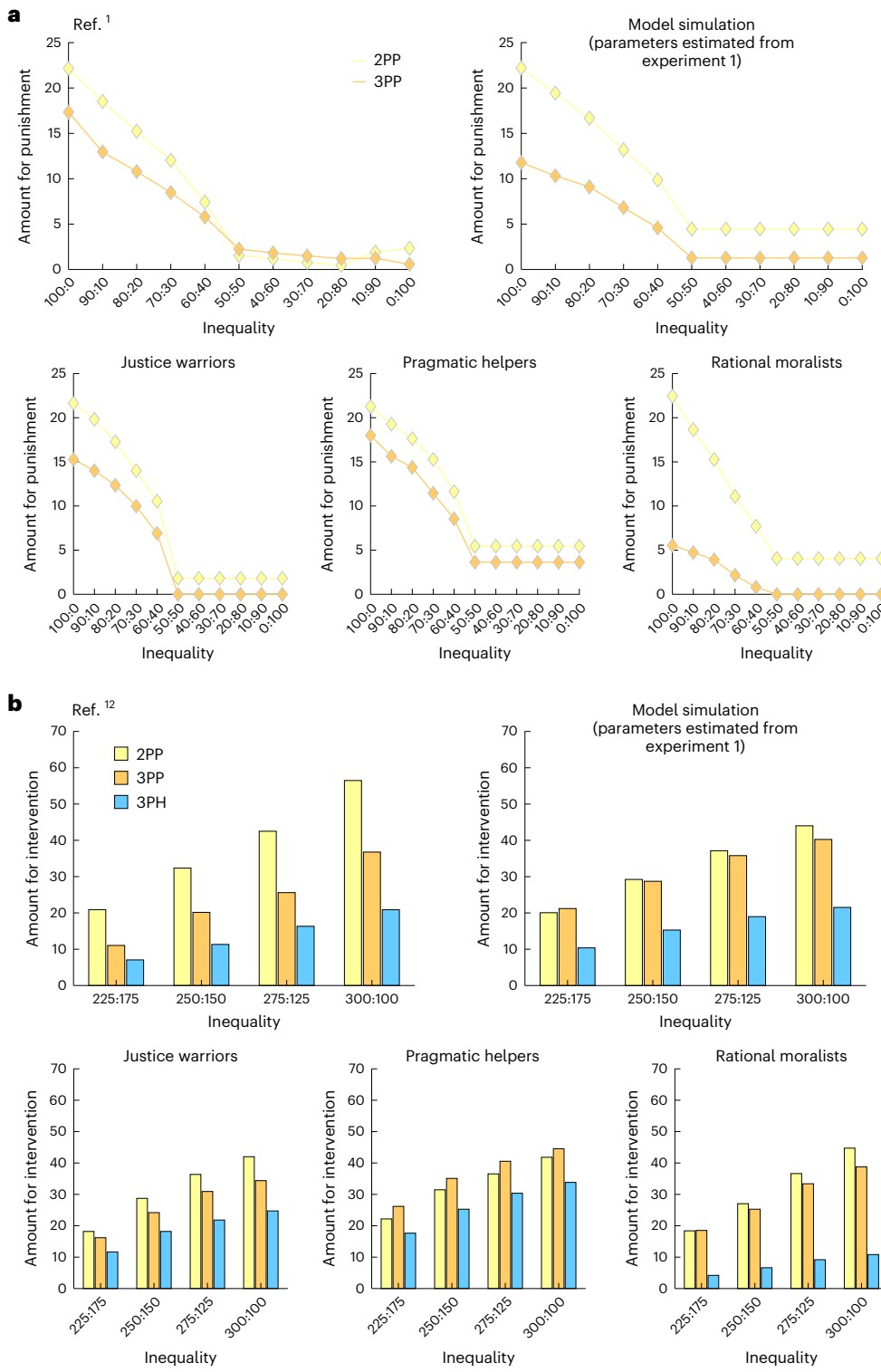

**Fig. 6 | Quantitative predictions of the motive cocktail model for more phenomena.** We used the full motive cocktail model estimated from the intervene-or-watch task (3PP and 3PH) to simulate the 2PP as well as the 3PP and 3PH behaviors in previous publications. In each panel (**a** or **b**), the upper left plot is the data; the upper right and three lower plots are model simulations respectively based on the estimated parameters of all participants and the three clusters of our experiment 1. **a**, Reproduction of the 2PP and 3PP behaviors in Fig. 5 of ref. 1. The amount participants would use to punish the allocator in a dictator game is plotted as a function of the level of inequality favoring the allocator. In simulating 2PP behaviors, participants—as the second party (the receiver)—were treated as a third party who had all the motives of third parties except for EC. Our model simulation (with no free parameters) reproduced two effects in the data: (1) the amount participants use for punishment decreases almost linearly with the decrease of inequality when the inequality favors the allocator and is nearly zero when the inequality favors the receiver, and (2) 2PP is larger than 3PP. The simulation based on justice warriors' parameters best matched the data. **b**, Reproduction of the 2PP, 3PP and 3PH behaviors in ref. 12. The amount participants would use to intervene is plotted as a function of the level of inequality. The task scenario of ref. 12 differed from that of ref. 1 in that the first party steals from the second party, causing a more severe violation of social norms. In this case we assume that the EC is excluded from the motive cocktail for all intervention behaviors, which leads to larger amounts for punishment than helping. As in **a**, the simulation based on justice warriors' parameters best matches the data.

Upon completion of the experiment, participants were asked to fill out personality questionnaires that assessed their prosocial inclinations in everyday life, including a social value orientation scale (SVO)[27] to measure selfishness and the Interpersonal Reactivity Index[28] for empathy concern. We computed the Pearson correlation coefficients ($r$) between each participant's model parameters (from the motive cocktail model) and the participant's personality measures (Supplementary Figs. 10 and 11). In both experiments 1 and 2, we found that stronger self-centered disadvantageous inequality aversion ($\alpha$) or inaction ID ($\eta_{no}$) was associated with more selfishness. When one of these two parameters was controlled, the correlation between $\eta_{no}$ and selfishness (experiment 1, partial correlation coefficient $\rho = -0.22$, $P = 0.006$; experiment 2, $\rho = -0.16$, $P < 0.001$) was still significant, but the correlation between $\alpha$ and selfishness was significant only in experiment 2 (experiment 1, $\rho = -0.11$, $P = 0.16$; experiment 2, $\rho = -0.12$, $P < 0.001$). We also found that inaction ID ($\eta_{no}$) and action ID ($\eta_{yes}$) were associated with empathy in opposite directions. When one of these two parameters was controlled, the correlation between $\eta_{no}$ and empathy was still significant in both experiments (experiment 1, $\rho = -0.25$, $P = 0.002$; experiment 2, $\rho = -0.12$, $P < 0.001$), but the correlation between $\eta_{yes}$ and empathy was significant only in experiment 2 (experiment 1, $\rho = 0.12$, $P = 0.13$; experiment 2, $\rho = 0.12$, $P < 0.001$).

Before the main experiments, we recorded the amounts participants allocated to their receiver in a dictator game. Kruskal–Wallis tests revealed significant differences across the three clusters for both experiment 1 ($H(2) = 14.56$, $P < 0.001$) and experiment 2 ($H(2) = 46.72$, $P < 0.001$). In both experiments, rational moralists allocated least to their receiver (see Supplementary Fig. 12 for post hoc tests). We also found significant differences between the three clusters of participants in selfishness (Kruskal–Wallis tests: experiment 1, $H(2) = 11.70$, $P = 0.003$; experiment 2, $H(2) = 74.02$, $P < 0.001$) and empathy concern (experiment 1, $H(2) = 4.21$, $P = 0.122$; experiment 2, $H(2) = 21.32$, $P < 0.001$). According to the personality questionnaires, the rational moralists were the most selfish and the justice warriors had the highest empathy (see Supplementary Fig. 13 for post hoc tests), which echoes the highest inaction ID ($\eta_{no}$) in the former and highest action ID ($\eta_{yes}$) in the latter (Fig. 4f,h). We also report some exploratory analyses of cultural differences in Supplementary Section 5.

### The motive cocktail quantitatively reproduces more phenomena

To demonstrate that this motive cocktail estimated in participants' intervene-or-watch decisions underlies human responses to inequality in general, we performed an out-of-sample prediction, using an adapted version of the motive cocktail to simulate behavioral patterns in published studies with different experimental settings[1,12]. Indeed, we found that the motive cocktail model can predict the behavioral patterns in second-party punishment (2PP) as well as 3PP and 3PH (Fig. 6).

One robust phenomenon is that interveners spend more to penalize transgressors when they themselves are victims rather than unaffected third parties (that is, 2PP > 3PP). This can be explained by the motive of deterrence[7], which is not in conflict with our utility maximization framework. We integrate this by assuming that deterrence motives lead to reduced EC (parameter $\omega$) in second-party situations. More broadly, $\omega$ may decrease with social distance[29] and intent viciousness[30].

In our simulations, we model second-party interveners as having all the motives of third-party interveners except EC ($\omega = 0$, Methods). Using parameters estimated from experiment 1 participants, our model reproduces both the 2PP > 3PP phenomenon and the increase in punishment with increasing inequality observed in previous laboratory experiments[1,12]. For both experiments, simulations with the justice warriors' parameters best matched the data.

Stallen et al.[12] used a scenario where the first party robs the second party. The inequality here was caused by the more vicious intentions of the transgressor, thus triggering stronger 3PP than the same level of inequality caused by a dictator allocator (Supplementary Fig. 14). For this case, we assume that even unaffected third parties have no EC, allowing our model to reproduce the less common 3PP > 3PH phenomenon they observed.

## Discussion

While helping and punishment equally reduce VCI, they differ in their influences on SCI. Inequality aversion alone would predict a preference for punishment over helping, unless participants are more uncomfortable with their advantage over others than the reverse. However, participants in our experiments were more likely to help the victim than to punish the transgressor, a finding consistent with most studies[5,31–33]. The motive cocktail model can naturally explain the preference for helping over punishment, because it includes EC as a utility term: that is, people also care about the overall payoff of the transgressor and the victim. With an additional assumption that the motive of EC is weakened when the participant is the victim or when the transgressor violates social norms in a more aggressive way such as robbing or stealing from the victim[12,34], it can also explain why people spend more resources for 2PP than for 3PP[1,12] and why a reverse preference for punishment rather than helping is found in some studies[12,34], as our simulation shows (Fig. 6). Our model thus provides a unified account for 2PP, 3PP and 3PH behaviors.

One motive documented in previous studies, seemingly contradicting inequality aversion, is rank reversal aversion[23,24]. Our motive cocktail model includes a generalized form of this motive and reveals that participants in our experiment prefer to reverse the initial inequality, giving the victim an advantage over the transgressor, similar to the outcome in Shakespeare's *The Merchant of Venice*. This RP motive opposes rank reversal aversion, suggesting that the latter may apply only when the initial inequality is caused by luck[23,24], instead of by the intentional choice of the benefited party, as in our task and classic third-party intervention tasks[1,12].

In line with the joint functioning of multiple motives identified in our modeling analysis, we found a three-way interaction between cost, impact ratio and transgressor–victim inequality. Such an interaction was not reported in previous studies, probably because most studies used cost as a dependent rather than an independent variable, measuring the amount of money participants were willing to spend on the intervention, which would prevent such effects from being detected by usual statistical analysis. In contrast, the cost is manipulated by the experimenter in our task, resembling another type of real-world scenario where individuals are confronted with limited options when it comes to addressing others' inequalities.

Beyond individual differences in attention to others' inequality[35], we found that, even within the same individual, attention to others' inequality is modulated by the personal cost of intervening. The two forms of ID—inaction ID and action ID—have distinct psychological implications. The former assumes that people act as if increasingly ignoring the victim's inequality due to rising intervention costs, leading to reluctance to engage in potentially self-harming altruistic actions. Action ID assumes that people act as if ignoring the remaining inequality faced by the victim after their intervention, resulting in being willing to intervene even when it hardly improves equality. The co-existence of these two types of ID demonstrates motive diversity in altruistic behaviors across various social contexts. These findings have implications for addressing real-world social issues: reducing barriers and costs for reporting injustices can encourage public engagement against inequities, while emphasizing the resolution achieved by intervention can further encourage altruistic behavior.

In both the laboratory and the large-scale online experiments, we identified three types of intervener: justice warriors, pragmatic helpers and rational moralists, differing in intervention probability, sensitivity to variables such as cost and inequality, and preference for helping over punishment. The observed behavioral clustering aligns with previous

findings that most individuals possess some form of prosocial preference, with few being purely self-interested[36]. The motive parameters estimated from the motive cocktail model provide a multifacet measure of such individual differences, raising questions about how personal experiences, cultural background or genetic makeup may influence individuals' motives.

In sum, the proposed motive cocktail model extends the economic modeling of altruistic behaviors, enabling us to understand the cognitive processes behind human altruistic behaviors, measure individual differences related to psychiatric disorders and developmental trajectories, and more precisely predict behavior, guiding social policy-making to foster prosocial behaviors on a societal scale. By elucidating the cognitive processes underlying prosocial behavior and identifying various motives and individual differences, our model can provide insights into psychiatric disorders characterized by social dysfunction and inform future research on the neural basis of human morality and its disorders[37]. Our model and task framework can also be used to investigate the developmental trajectories of altruistic motives, guiding efforts to foster prosocial behaviors across life stages[38]. By capturing the interplay of multiple motives and their impact on behavioral patterns, our model enables more precise predictions of prosocial behavior. Leveraging insights from the motive cocktail model, interventions can be designed to account for individuals' diverse motivations, experiences and cross-cultural backgrounds[9], aiming to create a more cohesive and prosocial community. Meanwhile, further research is needed to bridge the gap between our simplified laboratory task and real-world applications.

We used a one-shot anonymous interaction setting, a common practice in previous studies[1,12,13,32,36,39–43], to minimize participants' concern for their own reputations, a motive that is instrumental to the long-term reciprocity in human society[44]. Consequently, our motive cocktail model, which adequately explained our data, excluded reputation as a motive. However, in real-world scenarios with more interaction opportunities, reputation concern is likely to influence 3PP and 3PH behaviors[3,6]. The victim's reputation (for example, once a transgressor or not) also matters, with reputation-based expectancies emerging early in human development[45]. Similarly, deterrence[7], reciprocity[8] or social norms beyond egalitarian distribution[10] are other real-world motives not examined in this Article. Integrating these motives into the motive cocktail model will be topics for future research. Whether the three types of intervener relate to the different cooperative types found in public goods games[46], thus connecting to a larger picture of human altruistic behaviors, also deserves future research.

## Methods
Both experiments 1 (in laboratory) and 2 (online) had been approved by the Ethics Committee of Beijing Normal University (CNL_A_0001_009 and IRB_A_0003_2020001).

### Experiment 1
**Participants.** Experiment 1 was conducted in a laboratory room at Beijing Normal University and 157 university students (59 males, mean age ± s.d. 21.24 ± 2.56) were recruited. No statistical methods were used to predetermine sample size. No participants were excluded from the subsequent analysis. Participants completed the screening form before the task to confirm that they had normal or corrected-to-normal vision and no history of psychiatric or neurological illness. All participants provided informed consent. On average, participants were compensated with ¥80 (range ¥60–120).

**Experimental procedure.** Participants were self-paced to read the instructions of the task. A quiz followed the completion of each subsection of the instruction. Participants proceeded to the next section of the instruction only if they gave the correct answer to the quiz. Before the formal task, participants underwent several practice

trials to ensure that they fully understood the rules of the game. The intervene-or-watch task (detailed below) lasted approximately 45 min. After completing the task, participants were asked whether they had any doubts or questions during the task in an open-ended question. In experiment 1, four participants reported doubts about whether all the players were real people. To examine whether participants who reported doubts used different strategies when compared with those who did not have doubts during the task, we conducted a GLMM similar to GLMM1 but added 'doubt' as an additional predictor (a categorical variable) in the model. We found that the predictor doubt could not predict participants' choice ($b = -2.74$, 95% CI [−7.19, 2.24], $P = 0.304$), and concluded that participants who reported doubts did not employ different strategies in the task. Therefore, all participants were included in the following analysis. In the final section, participants were asked to fill out a few personality questionnaires (detailed below), including measures of SVO, the Machiavellianism Scale (MACH–IV) and the Interpersonal Reactivity Index, to assess their prosocial personalities.

**The intervene-or-watch task and experimental design.** The intervene-or-watch task was a paradigm adapted from the 3PP task[1]. In the task, participants played the role of an unaffected third party who watched an anonymous dictator (transgressor) allocate amounts between himself/herself and an anonymous receiver (victim), and then decided whether to intervene. The stimuli were presented using the E-Prime 2.0 software (Psychology Software Tools). In each trial, the transgressor allocated the 100 game tokens between himself/herself and the victim, while the victim had to accept the offer without any other options. Participants were told that all offers between a transgressor and a victim were made by other real participants, and that their decisions would affect their own payoffs as well as those of the victims and the transgressors. In reality, the offers between the transgressors and the victims were generated by a custom code and were designed to disentangle different hypotheses. To give the participants a more realistic experience and to familiarize them with the roles in the game, they were instructed to play two trials of the dictator game, in which they played the role of transgressor and victim respectively. In the intervene-or-watch task, participants had 50 game tokens in each trial which could be used to reduce the payoff of the transgressor in the punishment scenario or increase the payoff of the victim in the helping scenario. To avoid serial or accumulative effects, participants were instructed that their payoff was independent across trials and would not be accumulated through the task. They were also informed that 10% of the trials would be randomly selected and implemented at the end of the study to determine the payoffs of all players (or roles). Specifically, participants' actual payment was calculated by adding a base payment to the average remaining tokens from these randomly selected trials, with each token being exchanged for ¥1. Additionally, participants were explicitly informed that the roles of the transgressor and the victim were played by different participants in each trial, hence encouraging them to make decisions based solely on the current situation. We are aware that our experimental setting included deception, in the sense that participants' intervention to the players in the dictator game was not really implemented. Nevertheless, all of the offers we used in the intervene-or-watch task were ones that real human players might make in the dictator game[47,48]. Such use of deception has been a common practice of previous studies[12,32]. Furthermore, participants' payoff was actually determined by the randomly selected 10% of their decisions, akin to a random lottery design[49], which did not involve deception.

Since all players in the task were anonymous, no reputation concern was involved in this task. The players also had no opportunities for interaction; thus, reciprocity could be excluded. Therefore, participants' decisions to help and to punish in the intervene-or-watch task were altruistic.

Each trial (Fig. 1c,d) began with a fixation cross (600–800 ms), followed by an inequality window (1,500 ms) displaying the allocation

between the transgressor and the victim, and an intervention offer window (1,500 ms) showing the intervention cost for the participant and the consequence of the intervention (impact ratio × intervention cost) to the transgressor or victim. Subsequently, in the decision window, participants were asked whether they would like to accept the intervention offer: yes (to intervene) or no (not to intervene). The intervention would only be implemented if participants chose yes. For example, if the intervention offer window displays an intervention cost of $x$ in a trial, a decision of intervention would result in the transgressor losing (or the victim gaining) $1.5x$ or $3.0x$ in the punishment (or helping) scenario. There was no time limit for the decision. A visual feedback window after the decision highlighted the selected choice in red. Four independent variables were varied across trials: scenario (punishment and helping), inequality (transgressor versus victim, 50:50, 60:40, 70:30, 80:20, 90:10, jitter ±2), cost (10, 20, 30, 40, 50) and impact ratio (1.5 and 3.0). This led to 100 unique conditions, with each condition repeated three times for each participant. The scenario variable varied between blocks and the other three variables were randomly interleaved within blocks. Before each block, participants were told whether the following section was the 'increase' condition (the helping scenario) or the 'reduce' condition (the punishment scenario). In total, each participant completed 300 trials in six blocks, with three blocks each for the punishment and helping scenarios. The main experiment of the intervene-or-watch task lasted 30.86 ± 3.25 min for experiment 1 and 33.97 ± 7.59 min for experiment 2. The main experiment included six blocks, with each block lasting around 5 min, followed by a 30 s rest between blocks.

**Personality questionnaires.** Following the intervene-or-watch task, participants completed several personality questionnaires that allowed us to access their prosocial tendencies in daily life. Specifically, SVO[27] was used to measure individual preference about how to allocate financial resources between themselves and others. A higher score on the SVO scale reflects a greater degree of concern for others' payoffs and, therefore, indicates a more prosocial personality. MACH−IV[50] was used to assess an individual's level of Machiavellianism, related to manipulative, exploitative, deceitful and distrustful attitudes. Higher scores on the MACH−IV scale are indicative of a more pronounced degree of Machiavellian traits. The Interpersonal Reactivity Index[28] was used to measure the multidimensional assessment of empathy, including (1) perspective-taking, assessing an individual's tendency to consider a situation from another's perspective; (2) fantasy, evaluating an individual's inclination to identify with the situation and emotions of characters in books, movies or theatrical performances; (3) empathy concern, measuring an individual's inclination to care about the feelings and needs of others; (4) personal distress, assessing an individual's tendency to experience distress and discomfort in challenging social situations.

**Model-free analysis.** Figures were generated using MATLAB R2020b (MathWorks) and R 4.2.1. All statistical analyses were conducted in R 4.2.1[51] and MATLAB R2020b. GLMMs assuming binomial distributed responses were used to model the probability of intervention, given various predictors (for instance, scenario, inequality) and their interactions. The GLMMs were implemented using the lme4 (v.1.1.30) package[52], with the fixed-effect coefficients output from the binomial GLMM on the logit scale and the significance of each coefficient determined by the $Z$ statistics. For significant main effects in GLMMs, two-tailed paired $t$-tests were used for pairwise comparisons for two adjacent conditions with Bonferroni correction. The standard linear mixed-effect models (LMMs), which assume that the error term is normally distributed, were estimated using the afex (v.1.2.1) package to model participants' decision times. For the estimation of marginal effects and the post hoc analysis, the emmeans (v.1.8.0) package was used[53]. Interaction contrasts were performed for significant

interactions and, when higher-order interactions were not significant, pairwise or sequential contrasts were performed for significant main effects. The null hypothesis testing reported in the main text (Kruskal–Wallis test and paired $t$-test) and in the Supplementary Information (Mann–Whitney test) were implemented in MATLAB R2020b using the Statistics and Machine Learning Toolbox.

GLMM1: participants' choices in all trials in experiment 1 are the dependent variable; fixed effects include an intercept, the main effects of the scenario, inequality, cost, ratio, trial number and all possible interaction effects of the independent variables; random effects include correlated random slopes of scenario, inequality, cost, ratio and trial number within participants and random intercept for participants. The scenario is a category variable. Trial number, inequality, cost and ratio are continuous variables that were normalized to $Z$ score before model estimation. The inclusion of trial number controls for time-related confounds, such as potential fatigue or practice effects. See Supplementary Table 1 for the statistical results of GLMM1.

GLMM2: participants' choices in all trials in experiment 2 are the dependent variable. The fixed and random effects remain the same as for GLMM1. See Supplementary Table 5 for the statistical results of GLMM2. Both the main and interaction effects of the independent variables on intervention decisions of experiment 1 (as in Fig. 1e–l) were replicated in experiment 2 (Supplementary Fig. 15a–m).

LMM1: participants' decision times for all trials in experiment 1 are the dependent variable. In addition to the fixed and random effects included in GLMM1, participants' intervention decisions (choice) are added as well. See Supplementary Table 6 and Supplementary Fig. 16 for the statistical results of LMM1.

We found an inverted U-shaped relationship between the intervention probability ($P$(yes)) and decision time (Supplementary Fig. 16j), which implies that participants made decisions with more difficulty when the decision uncertainty (or entropy) was higher. This result is in line with previous research demonstrating an inverted U-shaped relationship between confidence levels and decision times[54].

LMM2: participants' decision times for all trials in experiment 2 are the dependent variable. The fixed and random effects remain the same as for LMM1. See Supplementary Table 7 for the statistical results of LMM2. The inverted U-shaped relationship between the probability of intervention ($P$(yes)) and decision time was replicated in experiment 2 (Supplementary Fig. 17).

*Sensitivity analysis to different variables.* We measured participants' intervention sensitivity to different variables, which was defined as the normalized intervention probability difference after the corresponding variable was dichotomized (Fig. 4n–r and Supplementary Fig. 18). Specifically, participants' sensitivity to the main effects, including scenario, ratio, cost and inequality, was calculated as the intervention probability difference in the helping trials when compared with the punishment trials, the high-impact-ratio trials (3.0) compared with the low-impact-ratio trials (1.5), the low-cost trials (cost ≤ 20) compared with high-cost trials (cost > 20) and the high-inequality trials (that is, the inequality level between the transgressor and the victim is 80:20 and 90:10) compared with the low-inequality trials (70:30, 60:40 and 50:50), divided by their overall $P$(yes), respectively. For the interaction effects, the sensitivity (that is, the normalized intervention probability difference) was calculated in a similar way as the main effect, that is, marginalizing over the other variables.

**Behavioral modeling.** We assumed that participants would make decisions on each trial by calculating the utility of the two options (yes and no) and choosing the option with the higher utility. In the intervene-or-watch task, participants were given the context regarding inequality between a transgressor and a victim as well as other related variables (for instance, cost, impact ratio) from the perspective of a third party and afterwards made a decision between two alternatives,

yes (to intervene) and no (not to intervene). In general, participants calculated the utilities of the choices by estimating the reduction in inequality for others through their intervention and considering the associated cost to themselves. Specifically, if they chose 'yes' (decide to intervene), they could reduce the inequality between the transgressor and the victim to some extent but at a cost. In contrast, by choosing 'no' (decide not to intervene), they could retain the inequality between the transgressor and the victim without incurring any cost. To investigate how individuals make decisions in the intervene-or-watch task, we constructed a series of computational models with different utility calculation hypotheses (that is, combinations of multiple socioeconomic motives) and compared their goodnesses of fit.

Participants' choices were then modeled using the Softmax function[55], with the utilities of no intervention ($U_{no}$) and intervention ($U_{yes}$) from different models as the inputs:

$$P(\text{yes}) = \frac{1}{1 + e^{\lambda(U_{no} - U_{yes})}} \quad (1)$$

where the inverse temperature, parameter $\lambda \in [0, 10]$, controls the stochasticity of participants' choices, with a larger $\lambda$ corresponding to less noisy choices.

In the following descriptions, we will use $x_1$, $x_2$ and $x_3$ to denote the payoffs of the transgressor, the victim and the third party (participant) if the third party does not intervene (chooses 'no'), and use $x_1'$, $x_2'$ and $x_3'$ to denote the counterpart payoffs if the third party intervenes (chooses 'yes'). In particular, $x_3'$ is equal to $x_3 -$ cost in both scenarios. In the punishment scenario $x_1' = x_1 -$ impact ratio $\times$ cost and $x_2' = x_2$, while in the helping scenario $x_1' = x_1$ and $x_2' = x_2 +$ impact ratio $\times$ cost.

*Model 1. The baseline model.* We modeled each participant's choices of intervention in each trial (whether to choose the yes option) as outcomes from a Bernoulli distribution, where the intervention probability is controlled by a parameter $q \in [0, 1]$. For each participant, the probabilities of choosing the intervention ($P(\text{yes})$) and not choosing the intervention ($P(\text{no})$) are denoted as follows:

$$P(\text{yes}) = q \quad (2)$$

$$P(\text{no}) = 1 - q. \quad (3)$$

*Model 2. Self-interest model (SI).* The models based on socioeconomic motives started with SI, where participants only consider SI when making decisions, thus always leading to a reduced utility of the intervention. Participants' choices were then modeled using the Softmax function (equation 1).

$$U_{no} = x_3 \quad (4)$$

$$U_{yes} = x_3' \quad (5)$$

where $x_3$ denotes the payoff of the third party when choosing no (without intervention), which is always 50 tokens in each trial. $x_3'$ denotes the payoff of the third party after choosing yes (with intervention), which is equal to 50 − cost.

Building upon the SI model, the following hypothetical socioeconomic components were progressively introduced into the utility calculation and participants' choices were modeled using the Softmax function. The necessity of each component to explain participants' decisions was determined through model comparisons.

*Model 3. SI and self-centered inequality aversion aversion model (SI + SCI).* On the basis of the SI model, we added a self-centered inequality aversion (SCI) aversion component, which assumes that participants are averse to the inequality between themselves and others in both directions[11]. The self-centered disadvantageous Inequality aversion

denotes that participants are averse to others having more payoffs than themselves, while the self-centered advantageous Inequality aversion denotes that participants are averse to themselves having more payoffs than others. The contributions of self-centered disadvantageous and advantageous inequality[11] are controlled separately by the parameters $\alpha$ ($\alpha \in [0, 10]$) and $\beta$ ($\beta \in [0, 10]$) and are subtracted from the SI. Under the assumption of the SI + SCI model, participants are motivated to maximize their SI and meanwhile minimize the inequality between themselves and others, and then make a choice between no intervention and intervention on the basis of their respective utilities:

$$U_{no} = x_3 - \alpha \sum_{j=1}^{2} \max(x_j - x_3, 0) - \beta \sum_{j=1}^{2} \max(x_3 - x_j, 0) \quad (6)$$

$$U_{yes} = x_3' - \alpha \sum_{j=1}^{2} \max(x_j' - x_3', 0) - \beta \sum_{j=1}^{2} \max(x_3' - x_j', 0) \quad (7)$$

where $j$ denotes the index of the transgressor and victim; $x_1$ and $x_2$ represent the payoffs of the transgressor and the victim when the participant (third party) chooses no; $x_1'$ and $x_2'$ represent the payoffs of the transgressor and the victim after the intervention of the third party.

*Model 4. SI + SCI and victim-centered disadvantageous inequality aversion model (SI + SCI + VCI).* On the basis of the SI + SCI model, we introduced another previously proposed inequality component, the victim-centered disadvantageous inequality aversion (VCI). The VCI assumes that participants are averse to the transgressor having more payoff than the victim[13], with its contribution to the utility calculation determined by a parameter $\gamma$ ($\gamma \in [0, 10]$). Participants with larger $\gamma$ will be more willing to intervene in almost all punishment and helping scenarios. Within this model, participants were motivated to maximize SI and simultaneously minimize the two kinds of inequality aversion (SCI and VCI):

$$U_{no} = x_3 - \gamma \max(x_1 - x_2, 0) - \alpha \sum_{j=1}^{2} \max(x_j - x_3, 0)$$
$$- \beta \sum_{j=1}^{2} \max(x_3 - x_j, 0) \quad (8)$$

$$U_{yes} = x_3' - \gamma \max(x_1' - x_2', 0) - \alpha \sum_{j=1}^{2} \max(x_j' - x_3', 0)$$
$$- \beta \sum_{j=1}^{2} \max(x_3' - x_j', 0). \quad (9)$$

*Model 5. SI + SCI + VCI and efficiency concern model (SI + SCI + VCI + EC).* On the basis of the SI + SCI + VCI model, an efficiency concern (EC)[16] component was added to the model. EC assumes that participants are motivated to maximize the total payoff of others, which is weighted by parameter $\omega$ ($\omega \in [0, 10]$). Participants with larger $\omega$ will be more likely to intervene in the helping scenario, but not in the punishment scenario:

$$U_{no} = x_3 - \gamma \max(x_1 - x_2, 0) - \alpha \sum_{j=1}^{2} \max(x_j - x_3, 0)$$
$$- \beta \sum_{j=1}^{2} \max(x_3 - x_j, 0) + \omega(x_1 + x_2) \quad (10)$$

$$U_{yes} = x_3' - \gamma \max(x_1' - x_2', 0) - \alpha \sum_{j=1}^{2} \max(x_j' - x_3', 0)$$
$$- \beta \sum_{j=1}^{2} \max(x_3' - x_j', 0) + \omega(x_1' + x_2'). \quad (11)$$

**Model 6. SI + SCI + VCI + EC and reversal preference for victim-centered advantageous inequality model (SI + SCI + VCI + EC + RP).** On the basis of the SI + SCI + VCI + EC model, we introduced another component, the reversal preference for victim-centered advantageous inequality (RP), into the model. RP is mutually exclusive to VCI and assumes that participants prefer to reverse the economic status of the victim. That is, RP motivates participants to make the victim have more payoff than the transgressor by punishing the transgressor or helping the victim. The reversal preference is controlled by the parameter $\kappa$ ($\kappa \in [-10, 10]$). A positive value of $\kappa$ indicates that participants are in favor of the victim having more money than the transgressor, while a negative value indicates that they are averse to such reverse inequality. Participants with larger $\kappa$ will be more likely to intervene when the initial victim-centered disadvantageous inequality is small enough or the impact is large enough to guarantee an inequality reversal:

$$U_{no} = x_3 - \gamma \max(x_1 - x_2, 0) - \alpha \sum_{j=1}^{2} \max(x_j - x_3, 0)$$
$$-\beta \sum_{j=1}^{2} \max(x_3 - x_j, 0) + \omega(x_1 + x_2) + \kappa \max(x_2 - x_1, 0) \quad (12)$$

$$U_{yes} = x_3' - \gamma \max(x_1' - x_2', 0) - \alpha \sum_{j=1}^{2} \max(x_j' - x_3', 0)$$
$$-\beta \sum_{j=1}^{2} \max(x_3' - x_j', 0) + \omega(x_1' + x_2') + \kappa \max(x_2' - x_1', 0). \quad (13)$$

**Model 7. SI + SCI + VCI + EC + RP and inequality discounting model (the motive cocktail model, SI + SCI + VCI + EC + RP + ID).** On the basis of the SI + SCI + VCI + EC + RP model, we also included the inequality discounting (ID) component that we proposed. Thus, the motive cocktail model includes seven socioeconomic motives. ID is derived from the rational framework of economic decisions and is implemented to capture the interaction between SI and VCI. Specifically, ID assumes that people will systematically disregard the victim-centered disadvantageous inequality as costs increase. We proposed two types of ID: inaction ID (controlled by parameter $\eta_{no}$) and action ID (controlled by $\eta_{yes}$), which are respectively blind to the initial and residual disadvantageous inequalities between the transgressor and the victim under no intervention and intervention with rising costs, respectively. In the model fitting, the range of parameters $\eta_{no}$ and $\eta_{yes}$ is restricted to between 0 and 20.

Participants with larger $\eta_{no}$ would have a lower probability of intervening. The effect differs from victim-centered disadvantageous inequality aversion (small $\gamma$) in that at large $\eta_{no}$ the tendency to intervene would barely increase with inequality. Conversely, participants with larger $\eta_{yes}$, who subjectively exaggerate the reduction of inequality by intervention, would have a higher probability of intervening. Those with large $\eta_{yes}$ will have similarly high probability of intervening regardless of the impact ratio, as if they optimistically believe that the inequality would be minimized by any of their interventions:

$$U_{no} = x_3 - \gamma \max(x_1 - x_2, 0) \delta_{IID} - \alpha \sum_{j=1}^{2} \max(x_j - x_3, 0)$$
$$-\beta \sum_{j=1}^{2} \max(x_3 - x_j, 0) + \omega(x_1 + x_2) + \kappa \max(x_2 - x_1, 0) \quad (14)$$

$$\delta_{IID} = \frac{2}{1 + e^{\eta_{no}(\cos t/50)}} \quad (15)$$

$$U_{yes} = x_3' - \gamma \max(x_1' - x_2', 0) \delta_{AID} - \alpha \sum_{j=1}^{2} \max(x_j' - x_3', 0)$$
$$-\beta \sum_{j=1}^{2} \max(x_3' - x_j', 0) + \omega(x_1' + x_2') + \kappa \max(x_2' - x_1', 0) \quad (16)$$

$$\delta_{AID} = \frac{2}{1 + e^{\eta_{yes}(\cos t/50)}}. \quad (17)$$

**Redundancy checks on the parameter space.** In the estimated parameters, we observed three highly correlated pairs in the parameter space of the motive cocktail model: the values of parameter $\beta$ (self-centered advantageous inequality aversion) and $\gamma$ (victim-centered disadvantageous inequality aversion), $\alpha$ (self-centered disadvantageous inequality aversion) and $\omega$ (efficiency concern), $\gamma$ and $\eta_{no}$ (inequality inaction inattention). To exclude the possibility that the correlation was due to parameter redundancy in the model, we performed redundancy checks as follows. We first randomly shuffled participants' labels for different parameters to eliminate correlations in the shuffled parameters. On the basis of these shuffled parameters, we generated 157 synthetic datasets and used them to estimate the model parameters. We found little correlation between the parameters estimated from these synthetic datasets, which indicates that the high correlations found in the data reflect the behavioral characteristics of human participants rather than redundancy in the model itself (Supplementary Fig. 19).

**Model fitting and model comparison.** The behavioral modeling was implemented in MATLAB R2020b using custom codes. For each participant, we fit each model to their intervention decisions across all trials using maximum-likelihood estimates. The likelihood function derived from the binomial distribution was used to describe the relationship between participants' choice and the model's prediction. The function fmincon in MATLAB was used to search for the parameters that minimized negative log-likelihood. To increase the probability of finding the global minimum, we repeated the search process 500 times with different starting points. We compared the goodness of fit of each model on the basis of two metrics: the Akaike information criterion with a correction for sample size (AICc)[25] and the PEP of group-level Bayesian model selection[26]. The spm_BMS function of the SPM12 toolbox was used to perform the group-level Bayesian model selection. We chose to use the AICc as the metric of goodness of fit for model comparison for the following statistical reasons. First, the Bayesian information criterion is derived on the basis of the assumption that the 'true model' must be one of the models in the limited model set compared[56,57], which is unrealistic in our case. In contrast, AIC does not rely on this unrealistic true model assumption and instead selects out the model that has the highest predictive power in the model set[58]. Second, AIC is also more robust than the Bayesian information criterion for finite sample size[59].

**Model identifiability and parameter recovery analyses.** We further performed a model identifiability analysis to rule out the possibility of model misidentification in model comparisons. For each model, the parameters estimated from the data of all participants were used to generate a synthetic dataset of 157 participants. Each synthetic dataset regarding a specific model was then used to fit each of the seven alternative models and identify the best-fitting model by model comparison. We repeated the above procedure 100 times to calculate the percentage at which each model was identified as the best model on the basis of all synthetic datasets from a specific generating model. The highest percentage assigned to the same fitting model as the generating model suggests that the model is identifiable. To assess parameter recovery in the motive cocktail model (model 7: SI + SCI + VCI + EC + RP + ID), we computed the Pearson correlation between the parameters estimated from the 100 synthetic datasets (recovered parameters) and the parameters used to generate the synthetic datasets. A larger correlation coefficient between the recovered parameter and the estimated parameter indicates a non-redundancy in parameter space.

**Clustering analysis.** To gain further insight into whether the motive cocktail model (model 7: SI + SCI + VCI + EC + RP + ID) could explain the varying behavioral patterns of individuals, we classified participants'

intervention decisions using $k$-means clustering and then investigated the distributions of the estimated parameters across participants as well their unique contributions to behavioral patterns within each cluster. $k$-means clustering is an unsupervised machine learning algorithm relying on the Euclidean distance to classify each participant into a specific cluster with the nearest mean[60]. The clustering evaluation criterion was based on silhouette value, which denotes how well each participant was matched to its own cluster when compared with other clusters, with a higher silhouette value indicating that the clustering solution is more appropriate[61]. The optimal cluster solution for 157 participants in experiment 1 is 3 (Fig. 4b).

**Correlation analysis for parameters and personality measures.** To further validate the psychological basis of the hypothetical socioeconomic motives in the motive cocktail model, we calculated the Pearson correlation between the estimated parameters and the scores on the personality measurements. A similar correlation analysis between individuals' motive parameters and their sensitivity to different variables was carried out to unravel the contributions of the parameters to behavioral differences. Partial correlation was conducted when multiple parameters correlated with the same measurement to ensure that the observed relationships were not confounded by the potential influence of other variables. $\rho$, ranging between −1 and 1, quantifies the strength and direction of linear links between parameters and measured variables. For multiple comparisons, the false discovery rate was employed.

**Simulations to quantitatively reproduce previous phenomena.** We made slight modifications to the motive cocktail model and applied it to explain the intervention patterns in 2PP, 3PP and 3PH models in the following two studies. The adapted model could also be used to explain a broader range of phenomena in previous studies.

In a substudy conducted by Fehr and Fischbacher[1], participants attended a dictator game, which contains both 2PP condition and 3PP condition. At the beginning of the experiments, participants were randomly assigned either the role of the transgressor (player A) or the victim (player B). In the 2PP condition, the victim also acted as an intervener, who could punish the transgressor after observing the transfer from the transgressor accordingly. In the 3PP condition, the victim could only punish the dictator in another group (player A' and player B'), in which he/she served as an unaffected third party. A strategy method was implemented in the 3PP condition: the third party (player B) had to indicate how much she/he would punish the outgroup player A' for every possible transfer of A' to player B'. The results showed that the intervener as the victim exerted more punishment than the intervener as the third party for all transfer levels below 50 (2PP > 3PP), while the punishment was generally low and similar across transfer levels above 50 (Fig. 6a top left). In the study conducted by Stallen et al.[12], participants played three conditions of a justice game. In the 2PP games, the participants played the role of the partner (the victim), in which the taker (the transgressor) had the opportunity to take or steal chips (or payoff) from the victim, and afterward the victim was given the option of punishing the transgressor by spending chips of their own. In 3PP and 3PH games, participants played the role of an observer (the third party) to watch whether the transgressor stole chips from the victim and then decided whether to intervene to punish the transgressor or to compensate the victim, at their own cost. Every time participants needed to make a choice, all intervention costs ranging from 0 to 100 with a step of 10 were displayed on the screen. The results indicated that the intervener in the 2PP condition punished the transgressor more than in the 3PP condition (2PP > 3PP). In addition, the third party was more likely to punish than to compensate (3PP > 3PH, Fig. 6b top left).

For both studies, we simulated participants' choices by calculating the utility of selecting yes and no for each inequality level using equations (14)–(17). We assume that a second-party intervener, who is also the victim, is less concerned about overall welfare than is the third party. As the result, the second-party intervener has all the motives a third-party intervener would have except for EC. To implement this assumption, we replaced $x_3$ in equations (14)–(17) with $x_2$, and set the EC $\omega$ to 0 in the 2PP condition. The same lack-of-efficiency-concern assumption ($\omega = 0$) was implemented during the simulation of third-party punishment and compensation games in ref. 12. That is, we assume that the unaffected third party would ignore others' welfare in a robbery situation.

## Experiment 2

To further verify our findings and model specifications, we conducted experiment 2 using the same experimental paradigm as experiment 1 on an online participant platform (Prolific, https://www.prolific.co/) by recruiting a larger population with diverse cultural backgrounds.

**Preregistration.** Experiment 2 was preregistered on OSF (https://osf.io/gcsqp) on 29 September 2022. All methods and analyses followed the design and analysis plan in the preregistration, except that two additional models were tested: a model with lapse rate parameters and a simple-response model. This was due to more behavioral patterns being observed from the online experiment. Building on the results of the model-free analysis in experiment 1, we hypothesized that the main effect of inequality, intervention cost, impact ratio and the interaction of inequality × cost × ratio would be statistically significant, and that participants' intervention decisions would follow the patterns we observed in experiment 1. For the model-based analysis, we hypothesized that participants' decisions would be best described by the full motive cocktail model.

**Participants.** The criteria for participant recruitment were matched between experiments 2 and 1, including the age ranges (18–30 years old), student status and the degree of education. In addition, the study was only accessible to participants with an approval rate of over 90% in Prolific. We received 1,365 participants' submissions overall. One hundred and seven of them had an accuracy rate below 75% on the attention check task (see details below) and thus were rejected for further analysis. The final valid samples were 1,258 (621 male, 631 female, 6 genders unknown, aged 23.30 ± 2.89). No participants met the exclusion criterion of average decision time exceeding 2.5 s.d. from the mean decision time of all participants. All participants provided informed consent before the task to confirm that they took part in the study voluntarily, had normal or corrected-to-normal vision, and did not have a history of psychiatric or neurological illness. On average, participants were compensated with £9 (range £7–12).

**Determination of sample size.** The sample size for experiment 2 was predetermined using a parametric simulation method[62], derived from the motive cocktail model (the best-fitting model in experiment 1). The effect we focused on is the three-way interaction of inequality × cost × ratio (Fig. 1i). As compensation for the higher randomness of online participants' decisions, we added another two parameters, $P_{min}$ and $P_{max}$ (lapse rates), in the motive cocktail model to capture participants' minimal and maximal ($1 - P_{max}$) intervention probabilities. An online pilot study based on 32 participants showed that the motive cocktail model with lapse rates (see model 8 for more details) fit participants' behavior better. We therefore used model 8 to generate synthetic datasets to determine the sample size for experiment 2. Parameters $\alpha, \beta, \gamma, \omega, \eta_{no}, \eta_{yes}, \lambda$ were sampled from the gamma distribution, $\kappa$ was sampled from the normal distribution and $P_{min}$ and $P_{max}$ were sampled from the beta distribution. The generated intervention decisions of virtual participants were then exported to GLMM1 to obtain the effect size for each variable and their interactions. The power was defined as the percentage at which the three-way interaction effect reaches significance over a specific sample size. We tested different sample

sizes ranging from 100 to 1,500 virtual participants, with increments of 100. Within each sample size, we repeated the synthetic data generation and power calculation procedure 500 times. The power of the three-way interaction effect increased monotonically with sample size and achieved a power of 80% with at least 1,200 participants (Supplementary Fig. 7). Our final valid sample size was 1,258 participants from 66 countries (Supplementary Table 4).

**Experimental procedure.** The procedure of experiment 2 was the same as that of experiment 1, except that it was conducted on the Prolific platform, with the experimental paradigm coded using PsychoPy (v.2021.1.3) and PsychoJS (v.2021.1.3). Participants were informed that their base payment was £7 per hour, and 10% of trials would be randomly selected to determine their bonus after the experiment. The game tokens accumulated from these randomly selected trials would be exchanged for pennies at a 5:1 exchange rate. After the task, participants were asked "Did you think the experimenter had deceived you in any way at any point during the experiment?", with a binary choice of yes or no. Seventy-four participants answered yes, while the remaining 1,184 participants answered no. To investigate whether participants who had doubts (answered yes) employed different strategies when compared with those who did not have doubts (answered no) during the task, we conducted a GLMM (like GLMM2) and included doubt as a predictor (categorical variable) in the model. We found that the effect of doubt ($b = 0.15$, 95% CI [−0.09, 0.41], $P = 0.221$) was not statistically significant to predict participants' choices, suggesting that participants who reported doubts did not employ different strategies in the task. Therefore, all participants were included in the subsequent analysis.

**Attention check.** We used the same intervene-or-watch task in experiment 2 and included several attention checks during the task to ensure that participants remained constantly attentive to the current task. The attention checks consisted of 12 questions, with two questions interspersed in each block. For each block, the questions appeared randomly without telling the participants, and participants were asked to answer the questions with binary options about their last decision. Specifically, the questions were either "In the last trial, your decision was: yes/no?" or "The last trial was in the increase/reduce scenario?" in each block. Those (107 participants) who gave less than 75% accuracy in the attention checks (incorrect answers on more than three questions) were excluded from further analyses.

**Model-free analysis.** All 1,258 participants in experiment 2 were included in the model-free analysis (Supplementary Table 5). Among them, 492 (39.10%) out of 1,258 participants were best described by a simple-response model and were therefore excluded from the analyses in relation to the motive cocktail model. Specifically, only the remaining 60.90% of participants whose intervention patterns could be categorized as justice warriors, pragmatic helpers and rational moralists were included in the following analyses: data versus model prediction (Fig. 5), Kruskal–Wallis tests on the parameters $\eta_{yes}$, $\kappa$, $\eta_{no}$ (Supplementary Fig. 5), correlations between the parameters estimated from the motive cocktail model and the intervention sensitivities (Supplementary Fig. 18) as well as the personality measurements (Supplementary Fig. 11).

**Behavioral modeling.** *Model space.* We constructed two additional models (models 8 and 9) in experiment 2 to capture the behavioral patterns that online participants would make random choices in a certain amount of trials. Model 8 was constructed on the basis of the motive cocktail model. Model 9 is a simple-response model to capture the behavioral patterns of a proportion of participants in the online experiment 2 (39.10%) who only responded to some of the manipulated variables and seemed to entirely ignore the others.

*Model 8. The motive cocktail model with two lapse rate parameters.* The model assumes that participants make an intervention decision by considering both SI and all socioeconomic motives assumed in the motive cocktail model. However, participants' minimal and maximal intervention probabilities are bounded by two free parameters. Specifically, participants are willing to randomly intervene with a probability of $P_{min}$ ($P_{min} \in [0, 0.5]$). Meanwhile, they constrain their maximum intervention probability below $1 − P_{max}$ ($P_{max} \in [0, 0.5]$). The utility calculations and choice mapping remain the same as equations (14)–(17) and equation 1, respectively.

$$P(yes)' = P_{min} + (1 - P_{min} - P_{max})P(yes) \tag{18}$$

where $P(yes)$ represents the choice probability based on the motive cocktail model.

*Model 9. Simple-response model.* Some of these online participants were sensitive to only a few of the manipulated variables and seemed to use simple-response rules for responses. Thus, we also included a simple-response model that linearly combines different manipulated variables (scenario, inequality, cost and ratio) to describe participants' behavior:

$$P(yes) = \frac{1}{1 + e^{\gamma(\beta_1 \text{Scenario} + \beta_2 \text{Inequality} + \beta_3 \text{Cost} + \beta_4 \text{Ratio})}} \tag{19}$$

$$P(yes)' = P_{min} + (1 - P_{min} - P_{max})P(yes). \tag{20}$$

**Reporting summary**
Further information on research design is available in the Nature Portfolio Reporting Summary linked to this article.

## Data availability
Source data for Figs. 1–6 and most Supplementary Figures and Tables as well as all the raw data produced in this study are available at https://doi.org/10.17605/OSF.IO/6G293 ref. 63.

## Code availability
All codes from this study are available at https://doi.org/10.17605/OSF.IO/6G293 ref. 63.

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

## Acknowledgements

We thank E. Fehr, C. C. Ruff, F. Cushman and J. Gross for discussions, G. Chen for helping to program experiment 2 and H. Lu for statistical and visualization consultation. This study was partly supported by the Scientific and Technological Innovation (STl) 2030—Major Projects 2021ZD020050 (to C.L.), the National Natural Science Foundation of China (32171095 to H.Z., and 32271092, 32130045 to C.L.), funding from Peking–Tsinghua Center for Life Sciences (to H.Z.), the Major Project of National Social Science Foundation 19ZDA363 (to C.L.) and the Beijing Municipal Science and Technology Commission Z151100003915122 (to C.L.).

## Author contributions

Conceptualization: X.W., X.R., C.L., H.Z. Investigation: X.W. Data curation: X.W., X.R. Formal analysis: X.W., X.R. Methodology—development and design of methodology: X.W., X.R., C.L., H.Z. Methodology—creation of models: X.W., X.R., H.Z. Software: X.W. Visualization: X.W. Writing—original draft: .X.W, X.R. Writing—review & editing: X.W., X.R., H.Z., C.L. Funding acquisition: C.L., H.Z. Supervision: C.L., H.Z.

## Competing interests

The authors declare no competing interests.

## Additional information

**Correspondence and requests for materials** should be addressed to Chao Liu or Hang Zhang.

# Reporting Summary

## Statistics

For all statistical analyses, confirm that the following items are present in the figure legend, table legend, main text, or Methods section.

| n/a | Confirmed | |
|---|---|---|
| ☐ | ☒ | The exact sample size (*n*) for each experimental group/condition, given as a discrete number and unit of measurement |
| ☐ | ☒ | A statement on whether measurements were taken from distinct samples or whether the same sample was measured repeatedly |
| ☐ | ☒ | The statistical test(s) used AND whether they are one- or two-sided<br>*Only common tests should be described solely by name; describe more complex techniques in the Methods section.* |
| ☐ | ☒ | A description of all covariates tested |
| ☐ | ☒ | A description of any assumptions or corrections, such as tests of normality and adjustment for multiple comparisons |
| ☐ | ☒ | A full description of the statistical parameters including central tendency (e.g. means) or other basic estimates (e.g. regression coefficient) AND variation (e.g. standard deviation) or associated estimates of uncertainty (e.g. confidence intervals) |
| ☐ | ☒ | For null hypothesis testing, the test statistic (e.g. *F*, *t*, *r*) with confidence intervals, effect sizes, degrees of freedom and *P* value noted<br>*Give P values as exact values whenever suitable.* |
| ☒ | ☐ | For Bayesian analysis, information on the choice of priors and Markov chain Monte Carlo settings |
| ☒ | ☐ | For hierarchical and complex designs, identification of the appropriate level for tests and full reporting of outcomes |
| ☐ | ☒ | Estimates of effect sizes (e.g. Cohen's *d*, Pearson's *r*), indicating how they were calculated |

*Our web collection on statistics for biologists contains articles on many of the points above.*

## Software and code

Policy information about availability of computer code

| Data collection | The data of Experiment 1 (N = 157) were collected at Beijing Normal University using E-Prime 2.0 software (Psychology Software Tools, Inc., Sharpsburg, PA, USA). The data of Experiment 2 (N = 1258) were collected on the online participant recruitment platform Prolific (https://www.prolific.co/) using PsychoPy (v2021.1.3) and PsychoJS (v2021.1.3). |
|---|---|
| Data analysis | Figures were generated by MATLAB R2020b and R 4.2.1.<br>The behavioral modeling was implemented in MATLAB R2020b using custom codes. The fmincon function in Matlab was used to search for the parameters that minimized negative log likelihood. The spm_BMS function of the SPM12 toolbox was used to perform the group-level Bayesian model selection.<br>All statistical analyses were implemented in R 4.2.1 and Matlab R2020b. The GLMMs were implemented by the "lme4 v1.1.30" package. The standard linear mixed-effect models (LMM) were estimated using the "afex v1.2.1" package. For the estimation of marginal effects and the post hoc analysis, the "emmeans v1.8.0" package was used. Interaction contrasts were performed for significant interactions and, when higher-order interactions were not significant, pairwise or sequential contrasts were performed for significant main effects.<br>Kruskal-Wallis tests, Mann-Whitney U tests, and paired t-tests (based on the significant main effects in GLMMs) were implemented in MATLAB R2020b using the Statistics and Machine Learning Toolbox.<br>All codes from this study are available at https://doi.org/10.17605/OSF.IO/6G293 |

For manuscripts utilizing custom algorithms or software that are central to the research but not yet described in published literature, software must be made available to editors and reviewers. We strongly encourage code deposition in a community repository (e.g. GitHub). See the Nature Portfolio guidelines for submitting code & software for further information.

## Data

Policy information about availability of data

All manuscripts must include a data availability statement. This statement should provide the following information, where applicable:

- Accession codes, unique identifiers, or web links for publicly available datasets
- A description of any restrictions on data availability
- For clinical datasets or third party data, please ensure that the statement adheres to our policy

> Source data for Figures 1–6 and most Supplementary Figures and Tables as well as all the raw data produced in this study are available at https://doi.org/10.17605/OSF.IO/6G293

## Research involving human participants, their data, or biological material

Policy information about studies with human participants or human data. See also policy information about sex, gender (identity/presentation), and sexual orientation and race, ethnicity and racism.

| | |
|---|---|
| Reporting on sex and gender | A total of 157 participants (59 males and 98 females) were included in the data analysis of Experiment 1. A total of 1258 participants (621 males, 631 females, 6 non-binary or unknown) were included in the data analysis of Experiment 2. |
| Reporting on race, ethnicity, or other socially relevant groupings | We did not report any information on participants' race or ethnicity. |
| Population characteristics | The criteria for participant recruitment were matched for Experiments 1 and 2, including the age ranges (18–30 years old), student status and the degree of education (undergraduate students or above). |
| Recruitment | Participants in Experiment 1 were students at Beijing Normal University and were recruited through the university's online forum (http://www.ddw.zone/), campus advertisements and a Wechat Group built for participant recruitment. Participants in Experiment 2 were recruited from the Prolific platform (https://www.prolific.co/). All participants provided informed consent before the task to confirm they voluntarily took part in the study.<br>Our study sample consists primarily of university students aged 18-30, which may introduce self-selection and volunteer bias, potentially limiting the generalizability of our findings. Participants who volunteer for studies often differ in motivation or attitudes from those who do not, which could influence the results. Caution should be exercised when extending our conclusions to non-student populations and other age groups. |
| Ethics oversight | The study was approved by the Ethics Committee of Beijing Normal University. |

Note that full information on the approval of the study protocol must also be provided in the manuscript.

# Field-specific reporting

Please select the one below that is the best fit for your research. If you are not sure, read the appropriate sections before making your selection.

☐ Life sciences  ☒ Behavioural & social sciences  ☐ Ecological, evolutionary & environmental sciences

For a reference copy of the document with all sections, see nature.com/documents/nr-reporting-summary-flat.pdf

# Life sciences study design

All studies must disclose on these points even when the disclosure is negative.

| | |
|---|---|
| Sample size | *Describe how sample size was determined, detailing any statistical methods used to predetermine sample size OR if no sample-size calculation was performed, describe how sample sizes were chosen and provide a rationale for why these sample sizes are sufficient.* |
| Data exclusions | *Describe any data exclusions. If no data were excluded from the analyses, state so OR if data were excluded, describe the exclusions and the rationale behind them, indicating whether exclusion criteria were pre-established.* |
| Replication | *Describe the measures taken to verify the reproducibility of the experimental findings. If all attempts at replication were successful, confirm this OR if there are any findings that were not replicated or cannot be reproduced, note this and describe why.* |
| Randomization | *Describe how samples/organisms/participants were allocated into experimental groups. If allocation was not random, describe how covariates were controlled OR if this is not relevant to your study, explain why.* |
| Blinding | *Describe whether the investigators were blinded to group allocation during data collection and/or analysis. If blinding was not possible, describe why OR explain why blinding was not relevant to your study.* |

# Behavioural & social sciences study design

All studies must disclose on these points even when the disclosure is negative.

| | |
|---|---|
| Study description | This study investigated the numerous, interdependent socioeconomic motives for altruistic behaviors when human participants performed a a Intervene-or-Watch task in which they played the role of an unaffected third party watching an anonymous dictator ("transgressor") allocated amounts between himself/herself and an anonymous receiver ("victim"), and then decided whether to intervene. <br> The study is quantitative, where we manipulate four independent variables across trials: scenario (punishment and helping), inequality (transgressor vs victim, 50:50, 60:40, 70:30, 80:20, 90:10, jitter ± 2), cost (10, 20, 30, 40, 50), and impact ratio (1.5 and 3.0). The participants' intervention decision in each trial was recorded. |
| Research sample | In Experiment 1, there were 157 valid participants (59 males, mean age ± SD: 21.24 ± 2.56) , who were recruited in Beijing Normal University. In Experiment 2, there were 1258 valid participants (621 males, 631 females and 6 non-binary or unknown, aged 23.30 ± 2.89) from over 60 countries (or regions), recruited from the Prolific platform. The criteria for participant recruitment were matched between Experiment 1 and Experiment 2, including the age ranges (18–30 years old), student status and the degree of education. Our study sample, primarily university students aged 18-30, was chosen for comparability with previous research and convenience of access. This demographic focus provides insights into specific developmental stages, but may limit the representativeness and generalizability of our findings to the broader population. |
| Sampling strategy | All participants from Experiment 1 and 2 were randomly sampled. For Experiment 1, no statistical methods were used to predetermine sample size, but our sample size is comparable to that of similar laboratory experiments in social decision making (e.g., 112 in FeldmanHall et al., 2014, Nature Communications; 144 in Rockenbach & Milinski, 2006, Nature). For Experiment 2, the sample size was predetermined using a parametric simulation method to achieve 80% statistical power for the three-way interaction of inequality × cost × ratio we found in Experiment 1. |
| Data collection | For Experiment 1, data were collected using E-prime and stored on the stimulus presentation computers. For Experiment 2, data were collected and saved via Pavlovia (https://pavlovia.org/). All data were stored anonymously. For the Experiment 1 (in-laboratory), no one was present besides the participant and the researcher. For the online Experiment 2, only the participant was present during the study. The researcher was blinded to experimental condition and/or the study hypothesis. |
| Timing | Data were collected between December 2019 and April 2020 for Experiment 1, and between August and November 2022 for Experiment 2. In August 2022, we validated the online experimental program using a small pilot sample with N = 32 that was later included in  Experiment 2. |
| Data exclusions | In Experiment 1, no participants were excluded. <br> In Experiment 2, attention checks were performed during the study to ensure that participants maintained attentive to the task. The attention checks consisted of 12 questions, with two questions in each block, appearing at random time. Participants who completed the experiment but achieved an accuracy of less than 75% in the attention checks (i.e., incorrectly answered more than three questions) were excluded from further analyses (107 participants). For the final 1258 participants in Experiment 2, no participants were further excluded, while some of the analyses were conducted only on the participants whose intervention patterns were not categorized into simple-responses. |
| Non-participation | In Experiment 1, no participants dropped out. In Experiment 2, a total of 626 participants dropped out. Among them, 559 participants left the study early or withdrew their submission after completing the study (labeled as "returned" in Prolific), and 67 participants failed to complete the experiment in the 140-minute time limit (labeled as "time out" in Prolific; average participants took approximately 60 minutes to complete the experiment). |
| Randomization | This study had a within-participant design. |

# Ecological, evolutionary & environmental sciences study design

All studies must disclose on these points even when the disclosure is negative.

| | |
|---|---|
| Study description | *Briefly describe the study. For quantitative data include treatment factors and interactions, design structure (e.g. factorial, nested, hierarchical), nature and number of experimental units and replicates.* |
| Research sample | *Describe the research sample (e.g. a group of tagged Passer domesticus, all Stenocereus thurberi within Organ Pipe Cactus National Monument), and provide a rationale for the sample choice. When relevant, describe the organism taxa, source, sex, age range and any manipulations. State what population the sample is meant to represent when applicable. For studies involving existing datasets, describe the data and its source.* |
| Sampling strategy | *Note the sampling procedure. Describe the statistical methods that were used to predetermine sample size OR if no sample-size calculation was performed, describe how sample sizes were chosen and provide a rationale for why these sample sizes are sufficient.* |
| Data collection | *Describe the data collection procedure, including who recorded the data and how.* |
| Timing and spatial scale | *Indicate the start and stop dates of data collection, noting the frequency and periodicity of sampling and providing a rationale for these choices. If there is a gap between collection periods, state the dates for each sample cohort. Specify the spatial scale from which* |

| | the data are taken |
|---|---|
| Data exclusions | *If no data were excluded from the analyses, state so OR if data were excluded, describe the exclusions and the rationale behind them, indicating whether exclusion criteria were pre-established.* |
| Reproducibility | *Describe the measures taken to verify the reproducibility of experimental findings. For each experiment, note whether any attempts to repeat the experiment failed OR state that all attempts to repeat the experiment were successful.* |
| Randomization | *Describe how samples/organisms/participants were allocated into groups. If allocation was not random, describe how covariates were controlled. If this is not relevant to your study, explain why.* |
| Blinding | *Describe the extent of blinding used during data acquisition and analysis. If blinding was not possible, describe why OR explain why blinding was not relevant to your study.* |

Did the study involve field work? ☐ Yes ☐ No

## Field work, collection and transport

| | |
|---|---|
| Field conditions | *Describe the study conditions for field work, providing relevant parameters (e.g. temperature, rainfall).* |
| Location | *State the location of the sampling or experiment, providing relevant parameters (e.g. latitude and longitude, elevation, water depth).* |
| Access & import/export | *Describe the efforts you have made to access habitats and to collect and import/export your samples in a responsible manner and in compliance with local, national and international laws, noting any permits that were obtained (give the name of the issuing authority, the date of issue, and any identifying information).* |
| Disturbance | *Describe any disturbance caused by the study and how it was minimized.* |

# Reporting for specific materials, systems and methods

We require information from authors about some types of materials, experimental systems and methods used in many studies. Here, indicate whether each material, system or method listed is relevant to your study. If you are not sure if a list item applies to your research, read the appropriate section before selecting a response.

### Materials & experimental systems

| n/a | Involved in the study |
|---|---|
| ☒ | ☐ Antibodies |
| ☒ | ☐ Eukaryotic cell lines |
| ☒ | ☐ Palaeontology and archaeology |
| ☒ | ☐ Animals and other organisms |
| ☒ | ☐ Clinical data |
| ☒ | ☐ Dual use research of concern |
| ☒ | ☐ Plants |

### Methods

| n/a | Involved in the study |
|---|---|
| ☒ | ☐ ChIP-seq |
| ☒ | ☐ Flow cytometry |
| ☒ | ☐ MRI-based neuroimaging |

## Antibodies

| | |
|---|---|
| Antibodies used | *Describe all antibodies used in the study; as applicable, provide supplier name, catalog number, clone name, and lot number.* |
| Validation | *Describe the validation of each primary antibody for the species and application, noting any validation statements on the manufacturer's website, relevant citations, antibody profiles in online databases, or data provided in the manuscript.* |

## Eukaryotic cell lines

Policy information about cell lines and Sex and Gender in Research

| | |
|---|---|
| Cell line source(s) | *State the source of each cell line used and the sex of all primary cell lines and cells derived from human participants or vertebrate models.* |
| Authentication | *Describe the authentication procedures for each cell line used OR declare that none of the cell lines used were authenticated.* |
| Mycoplasma contamination | *Confirm that all cell lines tested negative for mycoplasma contamination OR describe the results of the testing for mycoplasma contamination OR declare that the cell lines were not tested for mycoplasma contamination.* |

| Commonly misidentified lines (See ICLAC register) | *Name any commonly misidentified cell lines used in the study and provide a rationale for their use.* |
|---|---|

# Palaeontology and Archaeology

| Specimen provenance | *Provide provenance information for specimens and describe permits that were obtained for the work (including the name of the issuing authority, the date of issue, and any identifying information). Permits should encompass collection and, where applicable, export.* |
|---|---|
| Specimen deposition | *Indicate where the specimens have been deposited to permit free access by other researchers.* |
| Dating methods | *If new dates are provided, describe how they were obtained (e.g. collection, storage, sample pretreatment and measurement), where they were obtained (i.e. lab name), the calibration program and the protocol for quality assurance OR state that no new dates are provided.* |

☐ Tick this box to confirm that the raw and calibrated dates are available in the paper or in Supplementary Information.

| Ethics oversight | *Identify the organization(s) that approved or provided guidance on the study protocol, OR state that no ethical approval or guidance was required and explain why not.* |
|---|---|

Note that full information on the approval of the study protocol must also be provided in the manuscript.

# Animals and other research organisms

Policy information about studies involving animals; ARRIVE guidelines recommended for reporting animal research, and Sex and Gender in Research

| Laboratory animals | *For laboratory animals, report species, strain and age OR state that the study did not involve laboratory animals.* |
|---|---|
| Wild animals | *Provide details on animals observed in or captured in the field; report species and age where possible. Describe how animals were caught and transported and what happened to captive animals after the study (if killed, explain why and describe method; if released, say where and when) OR state that the study did not involve wild animals.* |
| Reporting on sex | *Indicate if findings apply to only one sex; describe whether sex was considered in study design, methods used for assigning sex. Provide data disaggregated for sex where this information has been collected in the source data as appropriate; provide overall numbers in this Reporting Summary. Please state if this information has not been collected. Report sex-based analyses where performed, justify reasons for lack of sex-based analysis.* |
| Field-collected samples | *For laboratory work with field-collected samples, describe all relevant parameters such as housing, maintenance, temperature, photoperiod and end-of-experiment protocol OR state that the study did not involve samples collected from the field.* |
| Ethics oversight | *Identify the organization(s) that approved or provided guidance on the study protocol, OR state that no ethical approval or guidance was required and explain why not.* |

Note that full information on the approval of the study protocol must also be provided in the manuscript.

# Clinical data

Policy information about clinical studies
All manuscripts should comply with the ICMJE guidelines for publication of clinical research and a completed CONSORT checklist must be included with all submissions.

| Clinical trial registration | *Provide the trial registration number from ClinicalTrials.gov or an equivalent agency.* |
|---|---|
| Study protocol | *Note where the full trial protocol can be accessed OR if not available, explain why.* |
| Data collection | *Describe the settings and locales of data collection, noting the time periods of recruitment and data collection.* |
| Outcomes | *Describe how you pre-defined primary and secondary outcome measures and how you assessed these measures.* |

# Dual use research of concern

Policy information about dual use research of concern

## Hazards

Could the accidental, deliberate or reckless misuse of agents or technologies generated in the work, or the application of information presented in the manuscript, pose a threat to:

No | Yes

☐ | ☐ Public health

☐ | ☐ National security

☐ | ☐ Crops and/or livestock

☐ | ☐ Ecosystems

☐ | ☐ Any other significant area

## Experiments of concern

Does the work involve any of these experiments of concern:

No | Yes

☐ | ☐ Demonstrate how to render a vaccine ineffective

☐ | ☐ Confer resistance to therapeutically useful antibiotics or antiviral agents

☐ | ☐ Enhance the virulence of a pathogen or render a nonpathogen virulent

☐ | ☐ Increase transmissibility of a pathogen

☐ | ☐ Alter the host range of a pathogen

☐ | ☐ Enable evasion of diagnostic/detection modalities

☐ | ☐ Enable the weaponization of a biological agent or toxin

☐ | ☐ Any other potentially harmful combination of experiments and agents

# Plants

| Seed stocks | *Report on the source of all seed stocks or other plant material used. If applicable, state the seed stock centre and catalogue number. If plant specimens were collected from the field, describe the collection location, date and sampling procedures.* |
|---|---|
| Novel plant genotypes | *Describe the methods by which all novel plant genotypes were produced. This includes those generated by transgenic approaches, gene editing, chemical/radiation-based mutagenesis and hybridization. For transgenic lines, describe the transformation method, the number of independent lines analyzed and the generation upon which experiments were performed. For gene-edited lines, describe the editor used, the endogenous sequence targeted for editing, the targeting guide RNA sequence (if applicable) and how the editor was applied.* |
| Authentication | *Describe any authentication procedures for each seed stock used or novel genotype generated. Describe any experiments used to assess the effect of a mutation and, where applicable, how potential secondary effects (e.g. second site T-DNA insertions, mosiacism, off-target gene editing) were examined.* |

# ChIP-seq

## Data deposition

☐ Confirm that both raw and final processed data have been deposited in a public database such as GEO.

☐ Confirm that you have deposited or provided access to graph files (e.g. BED files) for the called peaks.

| Data access links<br>May remain private before publication. | *For "Initial submission" or "Revised version" documents, provide reviewer access links.  For your "Final submission" document, provide a link to the deposited data.* |
|---|---|
| Files in database submission | *Provide a list of all files available in the database submission.* |
| Genome browser session<br>(e.g. UCSC) | *Provide a link to an anonymized genome browser session for "Initial submission" and "Revised version" documents only, to enable peer review.  Write "no longer applicable" for "Final submission" documents.* |

## Methodology

| Replicates | *Describe the experimental replicates, specifying number, type and replicate agreement.* |
|---|---|
| Sequencing depth | *Describe the sequencing depth for each experiment, providing the total number of reads, uniquely mapped reads, length of reads and whether they were paired- or single-end.* |
| Antibodies | *Describe the antibodies used for the ChIP-seq experiments; as applicable, provide supplier name, catalog number, clone name, and lot number.* |
| Peak calling parameters | *Specify the command line program and parameters used for read mapping and peak calling, including the ChIP, control and index files used.* |

| Data quality | *Describe the methods used to ensure data quality in full detail, including how many peaks are at FDR 5% and above 5-fold enrichment.* |
|---|---|
| Software | *Describe the software used to collect and analyze the ChIP-seq data. For custom code that has been deposited into a community repository, provide accession details.* |

# Flow Cytometry

## Plots

Confirm that:

☐ The axis labels state the marker and fluorochrome used (e.g. CD4-FITC).

☐ The axis scales are clearly visible. Include numbers along axes only for bottom left plot of group (a 'group' is an analysis of identical markers).

☐ All plots are contour plots with outliers or pseudocolor plots.

☐ A numerical value for number of cells or percentage (with statistics) is provided.

## Methodology

| Sample preparation | *Describe the sample preparation, detailing the biological source of the cells and any tissue processing steps used.* |
|---|---|
| Instrument | *Identify the instrument used for data collection, specifying make and model number.* |
| Software | *Describe the software used to collect and analyze the flow cytometry data. For custom code that has been deposited into a community repository, provide accession details.* |
| Cell population abundance | *Describe the abundance of the relevant cell populations within post-sort fractions, providing details on the purity of the samples and how it was determined.* |
| Gating strategy | *Describe the gating strategy used for all relevant experiments, specifying the preliminary FSC/SSC gates of the starting cell population, indicating where boundaries between "positive" and "negative" staining cell populations are defined.* |

☐ Tick this box to confirm that a figure exemplifying the gating strategy is provided in the Supplementary Information.

# Magnetic resonance imaging

## Experimental design

| Design type | *Indicate task or resting state; event-related or block design.* |
|---|---|
| Design specifications | *Specify the number of blocks, trials or experimental units per session and/or subject, and specify the length of each trial or block (if trials are blocked) and interval between trials.* |
| Behavioral performance measures | *State number and/or type of variables recorded (e.g. correct button press, response time) and what statistics were used to establish that the subjects were performing the task as expected (e.g. mean, range, and/or standard deviation across subjects).* |

## Acquisition

| Imaging type(s) | *Specify: functional, structural, diffusion, perfusion.* |
|---|---|
| Field strength | *Specify in Tesla* |
| Sequence & imaging parameters | *Specify the pulse sequence type (gradient echo, spin echo, etc.), imaging type (EPI, spiral, etc.), field of view, matrix size, slice thickness, orientation and TE/TR/flip angle.* |
| Area of acquisition | *State whether a whole brain scan was used OR define the area of acquisition, describing how the region was determined.* |

Diffusion MRI   ☐ Used   ☐ Not used

## Preprocessing

| Preprocessing software | *Provide detail on software version and revision number and on specific parameters (model/functions, brain extraction, segmentation, smoothing kernel size, etc.).* |
|---|---|
| Normalization | *If data were normalized/standardized, describe the approach(es): specify linear or non-linear and define image types used for transformation OR indicate that data were not normalized and explain rationale for lack of normalization.* |

| Normalization template | *Describe the template used for normalization/transformation, specifying subject space or group standardized space (e.g. original Talairach, MNI305, ICBM152) OR indicate that the data were not normalized.* |
|---|---|
| Noise and artifact removal | *Describe your procedure(s) for artifact and structured noise removal, specifying motion parameters, tissue signals and physiological signals (heart rate, respiration).* |
| Volume censoring | *Define your software and/or method and criteria for volume censoring, and state the extent of such censoring.* |

## Statistical modeling & inference

| Model type and settings | *Specify type (mass univariate, multivariate, RSA, predictive, etc.) and describe essential details of the model at the first and second levels (e.g. fixed, random or mixed effects; drift or auto-correlation).* |
|---|---|
| Effect(s) tested | *Define precise effect in terms of the task or stimulus conditions instead of psychological concepts and indicate whether ANOVA or factorial designs were used.* |

Specify type of analysis: ☐ Whole brain ☐ ROI-based ☐ Both

| Statistic type for inference<br><br>(See Eklund et al. 2016) | *Specify voxel-wise or cluster-wise and report all relevant parameters for cluster-wise methods.* |
|---|---|
| Correction | *Describe the type of correction and how it is obtained for multiple comparisons (e.g. FWE, FDR, permutation or Monte Carlo).* |

## Models & analysis

| n/a | Involved in the study |
|---|---|
| ☐ | ☐ Functional and/or effective connectivity |
| ☐ | ☐ Graph analysis |
| ☐ | ☐ Multivariate modeling or predictive analysis |

| Functional and/or effective connectivity | *Report the measures of dependence used and the model details (e.g. Pearson correlation, partial correlation, mutual information).* |
|---|---|
| Graph analysis | *Report the dependent variable and connectivity measure, specifying weighted graph or binarized graph, subject- or group-level, and the global and/or node summaries used (e.g. clustering coefficient, efficiency, etc.).* |
| Multivariate modeling and predictive analysis | *Specify independent variables, features extraction and dimension reduction, model, training and evaluation metrics.* |

