## [Peer Review File · Nature Computational Science]

Peer Review Information

Journal: Nature Computational Science

Manuscript Title: The “motive cocktail” in altruistic behaviors

Corresponding author name(s): Dr Hang Zhang

Editorial Notes:

Reviewer Comments & Decisions:

Decision Letter, initial version:
--

Date: 28th May 24 14:41:56

Last Sent: 28th May 24 14:41:56

Triggered By: Fernando Chirigati

From: fernando.chirigati@us.nature.com

To: hang.zhang@pku.edu.cn

BCC: fernando.chirigati@us.nature.com

Subject: Decision on Nature Computational Science manuscript NATCOMPUTSCI-24-0738

Message: ** Please ensure you delete the link to your author homepage in this e-mail if you wish to forward it to your co-authors. **

Dear Dr Zhang,

Your manuscript "The “motive cocktail” in altruistic behaviors" has now been seen by 4 referees, whose comments are appended below. Please note that Referees #2 and #3 co-reviewed the paper (and therefore there is a single report for both of these referees).

You will see that while they find your work of interest, they have raised points that need to be addressed before we can make a decision on publication. Naturally, we will need you to address **all** of the points raised.

While we ask you to address all of the points raised, the following points need to be substantially worked on:

- The Introduction and Discussion sections can be improved as per Referee #4's suggestions.
- Some modeling aspects need more clarification, including: the rationale for introducing the inequality discounting term, the rationale for how the different motives are combined into the model, the use of AICc instead of BIC for comparing the models, whether or not the comparison of the different individual difference groups accounted for the uncertainty in the parameter estimates, the assumptions made to use the motive cocktail model to make out-of-sample predictions, etc.
- Some details on the experimental task also need clarification, including: the frequency of participants never choosing the help/punish option, the amount of time taken on average for each experiment, the demographics of the participants, etc.
- As recommended by Referee #4, please use some real world examples to help readers understand the motives outside the context of stylized games.
- Please clarify the broader impact of the results and justify the claims on the significance of the work for real world issues.
- Specific scenarios must be at least discussed in the paper (or incorporated into the model/study), such as influence of people's past behaviors, the case when people choose how much they are willing to donate, etc.
- Regarding Experiment 2, Referee #4 indicates that the raw data suggests that patterns of punishment and helping are different and therefore not exactly replicated. Please make sure to investigate this in detail.

Please use the following link to submit your revised manuscript and a point-by-point response to the referees' comments (which should be in a separate document to any cover letter):

[REDACTED]

** This url links to your confidential homepage and associated information about manuscripts you may have submitted or be reviewing for us. If you wish to forward this e-mail to co-authors, please delete this link to your homepage first. **

To aid in the review process, we would appreciate it if you could also provide a copy of your manuscript files that indicates your revisions by making use of Track Changes or similar mark-up tools. Please also ensure that all correspondence is marked with your Nature Computational Science reference number in the subject line.

In addition, please make sure to upload a Word Document or LaTeX version of your text, to assist us in the editorial stage.

To improve transparency in authorship, we request that all authors identified as ‘corresponding author’ on published papers create and link their Open Researcher and Contributor Identifier (ORCID) with their account on the Manuscript Tracking System (MTS), prior to acceptance. ORCID helps the scientific community achieve unambiguous attribution of all scholarly contributions. You can create and link your ORCID from the home page of the MTS by clicking on ‘Modify my Springer Nature account’. For more information please visit www.springernature.com/orcid.

We hope to receive your revised paper within three weeks. If you cannot send it within this time, please let us know.

Best,
Fernando

--

Fernando Chirigati, PhD
Chief Editor, Nature Computational Science
Nature Portfolio

Reviewers comments:

Reviewer #1 (Remarks to the Author):

In the current study, the authors employed a newly developed third-party intervene task and computational modeling analyses to examine the effects of multiple motives on prosocial behaviors. These motives included two variants of self-centered inequality concern, victim-centered inequality concern, efficiency concern, reversal preference, and two non-linear combinations of elementary motives – inequality discounting. Across two experiments, the authors showed (N=157) and replicated (N=1258) their findings that individuals’ prosocial behaviors are governed by seven different motives. They also showed that the winning model can explain phenomena observed in other studies. The

authors did comprehensive model analyses to develop and validate their models, and the paper is well written. The findings shed new lights into the cognitive computational mechanisms of prosocial behavior, and will be of great interest to researchers in the field. However, I have several concerns regarding the authors' modeling approach and their explanations for the findings.

First, for the inequality concern, the author differentiated three different motives – two self-centered and one victim-centered motive, but for the reversal preference, they only considered potential reversal between dictator and receiver and not considered reversal between the participants themselves and the other two parties. I'm wondering if the authors have ever tried to do similar differentiations for reversal preference as for inequality concern, or for what reason they did not test potential reversal between the participant and the other two players.

Second, an issue related to my first point is about the explanation for why the authors observed reversal preference, rather than reversal aversion as in previous study. In the scenario of the current study, the initial inequality is intentionally generated by the dictator, rather than by luck, so it is highly likely that the reversal preference may reflect a strong preference to punish the transgressor with a bad intention (even in the helping scenario, rank reversal can be seen as a punishment for the transgressor). I'm wondering if the authors have any additional measures to test this potential hypothesis.

Third, can the authors more explicitly explain the rationale of introducing the inequality discounting term. I can understand the effect of the delta parameters on the utility, but I find it difficult to see why the delta equals to $2/(1+e^{\eta(\text{cost}/50)})$. Is it because the authors observed any model-free effects which can be best captured by such a discounting term? Therefore, it is also hard to understand why the authors proposed the other variant models. For now, it looks like the authors randomly tried several variant models and take the winning one, and there is no specific reason to explain why they assumed such a discounting term.

Fourth, for model construction, the authors added different motives one by one in to the model in a specific sequence, and not considered all possible combinations (for example there is no SI+SCI+VCI+RP model). I know it is next to impossible to test all the combinations, but I'm wondering whether the authors have a good rationale to justify such an approach to develop the models.

Fifth, the authors used AICc as the measure of model comparison. Their models are

nested to each other, and BIC is a recommended measure for model comparison among nested models. I'm wondering why the authors used AICc, rather than BIC for model comparison.

Minor:

I suggest the authors to include individual data points in figure 4f-h and figure s4. I cannot see any number or label on figure s5, please change the font size and enlarge the subplots.

Reviewers #2 and #3 (Remarks to the Author):

In this paper, the authors seek to address a gap in the literature on prosocial behavior, namely the possibility that prosocial behavior is driven by a collection of cooccurring motives. Using computational modeling, this paper is effective in untangling the influence of seven motives which drive a participant's decisions in allocating their resources to address a social injustice (a fake participant in a dictator game allocating more resources to themselves than to another fake participant). The paper also uses a clustering analysis to classify participants into groups based on their prosocial behavior patterns. Overall, we found the paradigm to be well-explained and the results to be compelling. However, we feel there are a few areas that could be clarified or discussed further.

One major point which should be expanded upon is the broader impact of these findings. The authors make a strong case for the "motive cocktail" model's ability to describe prosocial behaviors, but the impact this has on broad, societal goals could be stronger. Why do the authors believe that identifying these multiple motives, their interactions, and broad behavioral patterns is important for tackling prosocial goals?

One point that should be further discussed is the impact of reputation. While the authors make a strong case for eliminating a participant's motive to foster a good reputation, this would not entirely rule out the possibility for manipulating the reputation of transgressor and the victim in dictator game. If the real participant were to have multiple encounters with the dictator game participants, while remaining anonymous themselves, there could be an influence on how the participant allocates their resources based on observing their past actions. In other words, the current paradigm is effective in describing how people make third-party prosocial decisions when they know nothing of the other two parties other than the current situation, but in many real-world scenarios, participants may have other feelings based on observing their past behaviors. For example, if the current victim

was a transgressor in the past, or vice versa, how might this impact how participants allocate resources in the current trial, etc.? While it is not possible to address this with the current design, we feel this should be discussed.

The results presented pertain mainly to third-party scenarios. The authors do discuss two-party scenarios, but there is no discussion about whether or how the behavioral patterns they identify might relate to prosocial behavior in other scenarios, such as a traditional dictator game. In other words, do the authors think that participant's behavioral pattern (justice warrior, pragmatic helper, rational moralist) might predict how they themselves allocate resources in a traditional dictator game or how it might relate to general prosociality in different scenarios?

Relatedly, the authors correlate individual differences (e.g., selfishness, empathy) with the motive cocktail measures but did not see whether they varied in the different behavioral patterns identified in the cluster analysis. It is worth discussing, if not reporting, these relationships. As it stands, the individual difference analyses (line 434+) lack a bit of explanation/interpretation.

In the online replication, cluster analyses revealed 6 behavioral patterns, but the three not found in experiment 1 are largely ignored. This should be explained further, as it may be that categorizing participants into only 3 behavioral patterns is not capturing the full picture.

On lines 567-569, it is argued that cost being manipulated rather than chosen by participants resembles real-world scenarios. While this is true of some scenarios (being presented with preset donation options or rounding up change at a cash register) it is not true of all scenarios. People often do need to choose how much they are willing to donate. So while the paper does describe behavior in situations in which cost is manipulated, it should be mentioned as a limitation that it is unknown how these motives or behavioral patterns would influence behavior outside of this.

Lines 243-244 – “Inequality discounting refers to people’s tendency to underestimate the inequality between others as the intervention cost increases.” Would discounting not just refer to a decrease in motivation or willingness to right the inequality, as opposed to an underestimation of it? Similarly, the “rational moralists” are stated as believing that the situation is already fair. Could it not be that they just aren’t willing to incur the cost required to make it fair, unless the cost is comparatively low? This is evidenced in the thought bubble in Figure 4a reading “It’s already fair,” when an alternative could be “I’m not willing to expend the required resources to make it fair.”

Reviewer #4 (Remarks to the Author):

[Note that I have previously reviewed this paper for another journal. This review has been edited from my previous referee report, taking into account the changes that have been made since the previous version of the manuscript.]

This paper reports the results of a small lab experiment and a larger pre-registered online replication testing the motives underlying third-party helping and punishment in humans. In the experiments, participants witness splits of money between two individuals and are given the opportunity to intervene at a personal cost by either helping the victim or punishing the transgressor. In a within-subjects design, various aspects of the interactions are manipulated, including the intervention cost, the amount of inequality, and the impact-to-cost ratio. The results show that a “cocktail” of motives (including self-interest, inequality aversion, efficiency, reversal preferences, and newly-defined “inequality discounting”) explains participants’ behaviour in the experiments. The paper explores individual differences in these motives and argues that the motive cocktail model can also explain behavioural patterns in other settings.

I appreciated several aspects of the paper. The expansive within-subjects experimental design is an excellent way to explore the motives underlying third-party helping and punishment. My colleagues and I have recently used a similar within-subjects design to study the motives underlying punishment, albeit on a much smaller scale (Claessens et al., pre-print). I was impressed by the scope of the computational models that are fitted and compared throughout the paper. I also appreciated the large-scale replication study, which lends further credence to the paper’s claims. Finally, I would like to thank the authors for documenting the data and code and making them available for reviewers – I was able to examine the data myself and reproduce the main figures of the paper.

That said, I do have several concerns with the paper in its current form. For clarity, I have organised my concerns in line with the different sections of the paper.

I felt that the Introduction section of the paper did not sufficiently explore alternative motives for third-party helping and punishment. This section does describe Fehr & Schmidt’s (1999) utility model of inequality aversion, but it disregards other utility models based on reciprocity (Gintis, 2000), reputation (Jordan et al., 2016), and social norms (Kimbrough & Vostroknutov, 2016) on the tenuous grounds that “most 3PP or 3PH actions do not yield any direct benefit to the third party”. Other motives remain

unexplored, such as accounts of third-party punishment based on deterrence (Delton & Krasnow, 2017). I do not think that the current version of the paper discusses these alternative accounts in enough detail, in the introduction or the discussion sections. By discussing this diversity of accounts in more detail, the paper could make it clearer that it is focusing on only a subset of motives thought to underlie human altruism and punishment. Moreover, the introduction could be further improved by amending statements that are vague and/or unsupported by the cited evidence, such as “highlighting humans’ innate aversion to inequality” (line 50) or “the utility of (in)equality is computed in the human brain” (line 56).

Regarding the experimental task itself, there were several details that I was unclear on. First, how frequently did participants never choose the help/punish option? The reason I ask is that in our recent study of costly punishment (Claessens et al. pre-print) a significant portion of participants (~40-50%) chose never to punish, because doing so would reduce their bonus payment. I wonder if this is quite a common behaviour in the current study but is being clustered by the algorithm into the “rational moralist” bucket. Second, how much time did the experiment take, on average? I am a little concerned that fatigue could be affecting the results, particularly in Experiment 2. Third, am I correct in thinking that all participants in Experiment 2 were students? If so, this is quite an important demographic point to make clear earlier in the paper.

The first section of the Results describes the behavioural patterns from Experiment 1. For example, one result is that participants are less likely to intervene as the cost of the intervention increases. These findings were not at all surprising to me. Moreover, is it really true that the interaction between these variables (e.g., inequality x cost x ratio) has not been documented before? It was self-evident to me that these variables would have a combinatorial effect on behaviour. I’m also not sure how this interaction effect “pose[s] challenges for previous decision models that assume a linear combination of simple motives” (line 192-193). For these reasons, I think it would be better if these results (and the same results from Experiment 2) were moved to the supplement.

The next Results sections described the seven socioeconomic motives and the “motive cocktail” model. While it was good to have a clear description of the different motives, I think it also would have been useful to include some real world examples to help readers understand the motives outside the context of stylised games, e.g. Alice gives Bob less than she keeps for herself, but then Charlie intervenes to ensure that they both end up with the same amount (minimising victim-centred inequality). Real world examples like this might be especially useful for the newly-defined “inequality discounting” motives, which I admit it took me a few reads to properly understand. Since these new motives

are a key contribution of the paper (all other motives are drawn from past research), it's important that these are made intuitive for readers.

When comparing the different individual difference groups ("justice warriors", "pragmatic helpers", and "rational moralists") in the next Results section, I'm not sure the Kruskal-Wallis tests, post-hoc tests, or correlations are valid because the outcome variables are parameters that are estimated with uncertainty (e.g., α , β) rather than observed data. Do these analyses somehow account for the uncertainty in the parameter estimates?

Regarding Experiment 2, the paper claims that all major findings were replicated. However, examining the raw data suggests that patterns of punishment and helping are quite different in the second study: in general, people tend to punish and help at higher levels online compared to in the lab. The cluster analysis from the second experiment also identified six clusters, rather than three. These additional clusters are disregarded as a result of the lack of engagement in online studies (lines 432-433) but is there any evidence that this is actually the case? Since around 40% of the sample fit into these clusters, they should be summarised alongside the named clusters in the main text. Moreover, the cross-cultural variation in this larger sample is not currently exploited to its full potential. It would be even more interesting to see how the frequency of different individual difference types varies across cultures (this point is hinted at in the Discussion section).

The final Results section attempts to use the motive cocktail model to make out-of-sample predictions. While I appreciate the attempt to generalise the model to other settings, I'm not sure how valid it is for a model fitted to data on third-party interactions to be used to explain behaviour in second-party settings. Some assumptions are made to justify this leap, but despite the rewrite I'm still not sure I fully understand these assumptions. I'm also not clear about why the patterns of behaviour in Fehr & Fischbacher (2004) or Stallen et al. (2018) are seen as "puzzling" in the first place. For example, participants are more likely to punish the transgressor in second-party situations when they themselves were harmed (Fig 6a). Rather than being puzzling, this behaviour makes perfect sense when we consider that punishment may serve deterrent motives to discourage harm to oneself in the future (Delton & Krasnow, 2017). Without fully explaining how the model can be validly generalised to these settings or why these particular behavioural patterns are seen as puzzling to explain, I'm afraid that this section is more confusing than illuminating.

Like the Introduction section, I felt that the Discussion section could be improved by

referring to the additional motives that were unable to be explored in this particular experiment, such as reciprocity, reputation, social norms, deterrence, etc. It might also be worth discussing previous work that has captured individual differences in altruistic behaviour. For example, Kurzban & Houser (2001) find evidence for different cooperative types in a public goods game setting. Are these types similar or different to the types identified in the current research? It would also be good to tighten up some of the conclusions that are drawn in this section. For example, one claim is that the identification of an action inequality discounting motive in the study aligns with theories of warm glow, but I'm not sure exactly how this follows. The paper also makes some claims about the significance of this work for real world issues (e.g. policy formation), but this attempt at generalisability doesn't seem warranted to me given the lack of ecological validity in the lab-based experimental design (see also IJzerman et al. 2020).

Overall, I thought that the paper used a novel methodology to tackle an interesting question, and the findings have implications for our understanding of third-party helping/punishment in humans. By dealing with the above concerns and perhaps editing the text in places to improve clarity (some sections and figure legends I found quite dense to read), this could be a useful addition to the literature on human altruistic behaviour.

Please feel free to get in contact with me if you have any questions about this review.

Review signed: Scott Claessens (scott.claessens@gmail.com)

References

- Claessens, S., Atkinson, Q., & Raihani, N. (pre-print). Why do people punish? Evidence for a range of strategic concerns. PsyArXiv. <https://doi.org/10.31234/osf.io/ys6rm>
- Delton, A. W., & Krasnow, M. M. (2017). The psychology of deterrence explains why group membership matters for third-party punishment. *Evolution and Human Behavior*, 38(6), 734-743.
- Fehr, E., & Fischbacher, U. (2004). Third-party punishment and social norms. *Evolution and Human Behavior*, 25(2), 63-87.
- Fehr, E., & Schmidt, K. M. (1999). A theory of fairness, competition, and cooperation. *The Quarterly Journal of Economics*, 114(3), 817-868.
- Gintis, H. (2000). Strong reciprocity and human sociality. *Journal of Theoretical Biology*, 206(2), 169-179.
- IJzerman, H., Lewis Jr, N. A., Przybylski, A. K., Weinstein, N., DeBruine, L., Ritchie, S. J., ...

& Anvari, F. (2020). Use caution when applying behavioural science to policy. *Nature Human Behaviour*, 4(11), 1092-1094.

Jordan, J. J., Hoffman, M., Bloom, P., & Rand, D. G. (2016). Third-party punishment as a costly signal of trustworthiness. *Nature*, 530(7591), 473-476.

Kimbrough, E. O., & Vostroknutov, A. (2016). Norms make preferences social. *Journal of the European Economic Association*, 14(3), 608-638.

Kurzban, R., & Houser, D. (2001). Individual differences in cooperation in a circular public goods game. *European Journal of Personality*, 15(S1), S37-S52.

Stallen, M., Rossi, F., Heijne, A., Smidts, A., De Dreu, C. K., & Sanfey, A. G. (2018). Neurobiological mechanisms of responding to injustice. *Journal of Neuroscience*, 38(12), 2944-2954.

Reviewer #4 (Remarks on code availability):

There are README files online with metadata about the different folders/files and instructions for running the code. As stated in my review, I was able to run the code and reproduce the main figures from the paper.

Author Rebuttal to Initial comments

Manuscript No. NATCOMPUTSCI-24-0738A

Reply to Reviewers

We sincerely thank all the reviewers for their thorough evaluation of our manuscript and for providing valuable feedback. We appreciate these constructive comments and suggestions, which have significantly contributed to improving the quality of our work. Our replies below are in blue. Large additions or changes in the manuscript are also marked in blue.

Deputy Editor's Note:

- The Introduction and Discussion sections can be improved as per Referee #4's suggestions.
- Some modeling aspects need more clarification, including: the rationale for introducing the inequality discounting term, the rationale for how the different motives are combined into the model, the use of AICc instead of BIC for comparing the models, whether or not the comparison of the different individual difference groups accounted for the uncertainty in the parameter estimates, the assumptions made to use the motive cocktail model to make out-of-sample predictions, etc.
- Some details on the experimental task also need clarification, including: the frequency of participants never choosing the help/punish option, the amount of time taken on average for each experiment, the demographics of the participants, etc.
- As recommended by Referee #4, please use some real world examples to help readers understand the motives outside the context of stylized games.
- Please clarify the broader impact of the results and justify the claims on the significance of the work for real world issues.
- Specific scenarios must be at least discussed in the paper (or incorporated into the model/study), such as influence of people's past behaviors, the case when people choose how much they are willing to donate, etc.
- Regarding Experiment 2, Referee #4 indicates that the raw data suggests that patterns of punishment and helping are different and therefore not exactly replicated. Please make sure to investigate this in detail.

We appreciate the editor's effort and support as well as the opportunity to submit a revision. The list above is especially helpful in guiding the preparation of our revision. We have thoroughly revised our manuscript based on the reviewers' comments and provided a point-by-point response below.

Summary of major revisions:

- o We have revised the Introduction and Discussion of our manuscript following Reviewer #4's suggestions.
- o We have supplied the requested important details about the experiments and modeling.
- o We have discussed the broader impact of the results using real world examples.

- We have augmented our discussion of issues beyond the scope of the current task but of general interest to the community.
- We have reported the differences in the results of the two experiments and discussed their possible causes and implications.

Reviewers comments:

Reviewer #1 (Remarks to the Author):

In the current study, the authors employed a newly developed third-party intervene task and computational modeling analyses to examine the effects of multiple motives on prosocial behaviors. These motives included two variants of self-centered inequality concern, victim-centered inequality concern, efficiency concern, reversal preference, and two non-linear combinations of elementary motives – inequality discounting. Across two experiments, the authors showed (N=157) and replicated (N=1258) their findings that individuals' prosocial behaviors are governed by seven different motives. They also showed that the winning model can explain phenomena observed in other studies. The authors did comprehensive model analyses to develop and validate their models, and the paper is well written. The findings shed new lights into the cognitive computational mechanisms of prosocial behavior, and will be of great interest to researchers in the field. However, I have several concerns regarding the authors' modeling approach and their explanations for the findings.

Thank you for your constructive and insightful comments! We appreciate your positive feedback on the novelty and significance of our work, especially regarding the computational insights our work may provide for the cognitive mechanisms of prosocial behavior. We are also pleased to hear that our modeling is solid, our paper is well-written, and our findings will be of great interest to researchers in the field. Thank you for your encouraging words and support.

Q1: First, for the inequality concern, the author differentiated three different motives – two self-centered and one victim-centered motive, but for the reversal preference, they only considered potential reversal between dictator and receiver and not considered reversal between the participants themselves and the other two parties. I'm wondering if the authors have ever tried to do similar differentiations for reversal preference as for inequality concern, or for what reason they did not test potential reversal between the participant and the other two players.

Reply: Thanks for this great point and apologies for the confusion. In fact, the potential reversals between the participant and the other two parties have already been considered in our modeling, implicit in the motives of self-centered disadvantageous inequality aversion ("disadvantageous SCI", controlled by parameter α) and self-centered advantageous inequality aversion ("advantageous SCI", controlled by parameter β). On each trial, if participants did not intervene, disadvantageous SCI would apply to the inequality between the transgressor (i.e., dictator) and the participant, while advantageous SCI would apply to the inequality between the

participant and the victim (i.e., receiver) (Eq. 6). However, if participants intervened, reversals might occur between the participant and the other two parties, in which case disadvantageous SCI would apply to the reversed victim-participant inequality and advantageous SCI to the reversed participant-transgressor inequality (Eq. 7).

We apologize for not having made this point more explicit in the main text of the previous manuscript (though implicit in the equations defining the models). The confusion may also be partly due to the illustrative Fig. 2a for the SCI motives, which seems to imply disadvantageous SCI between the transgressor and the participant, and advantageous SCI between the participant and the victim. In the revised manuscript, we have added the following sentence in the legend of Fig. 2a to avoid confusion:

"Note that the illustration for SCI in the figure (disadvantageous SCI between self and transgressor, advantageous SCI between self and victim) may not apply to the inequality after intervention, where the direction of inequality between the participant and the other two parties might be reversed. Which type of SCI applies only depends on whether $\text{self} > \text{other}$ or $\text{self} < \text{other}$, irrespectively of the other is transgressor or victim."

Q2: Second, an issue related to my first point is about the explanation for why the authors observed reversal preference, rather than reversal aversion as in previous study. In the scenario of the current study, the initial inequality is intentionally generated by the dictator, rather than by luck, so it is highly likely that the reversal preference may reflect a strong preference to punish the transgressor with a bad intention (even in the helping scenario, rank reversal can be seen as a punishment for the transgressor). I'm wondering if the authors have any additional measures to test this potential hypothesis.

Reply: We agree with the reviewer's insight that the third party's seemingly contradictory motives of reversal preference and reversal aversion may arise in different scenarios, depending on the cause of the initial inequality. In previous studies where reversal aversion was reported (Li et al., 2022; Xie et al., 2017), the initial inequality between the first and second parties was caused by luck, instead of by the intentional choice of the first party as in our task and classic third-party intervention tasks (Fehr & Schmidt, 1999; Stallen et al., 2018). These two types of causes of initial inequality, with different real-life implications, can indeed lead to distinctively different responses in the third-party participants. For example, participants would hesitate to harm the first party whose advantage over the second party came by luck, a phenomenon termed "harm aversion" (Li et al., 2022); in contrast, participants are often willing to even sacrifice self-interest to punish the transgressor, as shown in our and previous third-party punishment studies (Fehr & Fischbacher, 2004; Stallen et al., 2018).

The reviewer's hypothesis about the influence of the first party's malicious intention on the third-party's reversal preference is an interesting one that deserves future research.

Though this hypothesis cannot be tested in the present study alone, we found preliminary evidence supporting it by comparing the results of our two experiments with those of two previous third-party punishment studies of comparable designs (Fehr & Fischbacher, 2004; Stallen et al., 2018), where the transgressor's behaviors are associated with different levels of malicious intentions (or, different extents of violating social norms). Among them, the initial inequality between the first and second parties in Fehr and Fischbacher (2004), like in our experiments, came from a dictator game, where the transgressor (dictator) allocates a fixed amount of money between themselves and the victim (receiver). In contrast, the transgressor in Stallen et al. (2018) was framed as more malicious (or, more severe violation of social norms), who "robbed" a specific amount from the victim. In the Fig. S14 below (included in the Supplement of the revised manuscript), we plot how the proportion of endowment the participants (as the third party) are willing to use to punish the transgressor varies with inequality (payoff ratio of transgressor to victim), separately for different experiments. Consistent with the hypothesis that more malicious transgressor incurs more severe third-party punishment, the punishment in Stallen et al. (2018) increases much faster with the level of inequality.

Fig. S14 | The influence of cause of inequality as well as level of inequality on the proportion of amounts participants (as a third party) used to punish the transgressor. The x-axis represents the ratio of the original allocation between the transgressor and the victim. A ratio of 1/1 means the amount of money is initially allocated equally between the transgressor and the victim, such as 50:50. A ratio of 9/1 means the transgressor has nine times the amount of money as the victim, such as 90:10. The y-axis represents the proportion of the amount that third parties (the participants) are willing to use for punishment relative to the maximum amount they have. For example, if a third party has up to 50 units and decides to use 10 units to punish

the transgressor, this is recorded as 0.2, indicating that the third party is willing to use 20% of their available amount to punish the transgressor. All the plotted experiments are comparable in that the total amount that the transgressors can allocate between themselves and the victims is twice the amount held by the third parties (the participants). To make the experiments further comparable, only data from the punishment scenario and from the conditions with an impact ratio of 3 are plotted, that is, participants' spending of one unit reduces the transgressor's amount by three units. Among the four experiments, the initial inequality between the first and second parties in Fehr and Fischbacher (2004), like in our two experiments, came from a dictator game, where the transgressor (dictator) allocates a fixed amount of money between themselves and the victim (receiver). In contrast, the transgressor in Stallen et al. (2018) was framed as more malicious (or, more severe violation of social norms), who "robbed" a specific amount from the victim. Note that compared with the other three experiments, the punishment in Stallen et al. (2018), increases much faster with the level of inequality.

Inspired by your question on reversal preference as well as by Reviewer #4's suggestion of exploring individual differences in our large sample, we also performed an additional statistical analysis to compare the strength of reversal preference—characterized by the κ parameter—in different cultural groups. First, we noticed that the median κ was smaller in our Experiment 1 (median = 0.008), whose participants were Chinese, compared to that of our online Experiment 2 (median = 3.560), whose participants came from multiple countries and regions (see Table S4 in the Supplement). We further categorized participants from both Experiment 1 and Experiment 2 into the East and West groups (see Supplementary Methods, "Exploratory analyses on cross-cultural differences", and our reply to Reviewer #4's Q6). We found that Western participants had significantly higher κ values than Eastern participants (see Fig. R1 below, Mann-Whitney U test: $Z = 6.02$, $p < 0.001$) (See Supplementary Methods and Fig. S21e).

Because our initial plan was not to study specific cultural backgrounds, we did not control for variables like sample size from different cultures, overseas experience, or immigration status. As a result, the sample sizes of different cultural groups were imbalanced; for different groups, the proportions of participants from the on-site Experiment 1 and the online Experiment 2 were also imbalanced. Therefore, we have included these results only as exploratory analyses (in response to Reviewer 4's suggestion) and explained their limitations.

Fig. R1 | Comparison of reversal preference (parameter κ) between the East and West groups, with participants combined from Experiments 1 and 2. Left panel: the κ value comparison between East and West groups. The bottom, middle, and top lines of the box plot respectively represent the first quartile, the median, and the third quartile of the data. The lines extending beyond the box refer to 1.5 times the interquartile range (IQR), i.e., the distance between the third quartile (Q3) and the first quartile (Q1). Each light-colored circle represents the κ value estimated from an individual participant. ***: $p < 0.001$ with Bonferroni corrections for seven comparisons (see Figure S9). Right panel: the distribution of κ values within each group. Red: East group. Blue: West group.

Q3: Third, can the authors more explicitly explain the rationale of introducing the inequality discounting term. I can understand the effect of the delta parameters on the utility, but I find it difficult to see why the delta equals to $2/(1+e^{\eta(\text{cost}/50)})$. Is it because the authors observed any model-free effects which can be best captured by such a discounting term? Therefore, it is also hard to understand why the authors proposed the other variant models. For now, it looks like the authors randomly tried several variant models and take the winning one, and there is no specific reason to explain why they assumed such a discounting term.

Reply: Apologies for not having made the rationale behind the inequality discounting term more explicitly. As now stated explicitly on p. 9, the introduction of the inequality discounting motives was motivated by the interaction effects we observed in the model-free statistical analysis:

“The remaining two motives under the class of “*inequality discounting*” are newly defined here to capture the interaction between self-interest and inequality aversion. They are partly motivated by the observed interaction effect that under higher intervention cost, participants’ probability of intervention not only was lower, but also increased more slowly with the transgressor-victim inequality (Fig. 1i).”

We had chosen the specific form $\delta = \frac{2}{1+e^{\eta(\text{cost}/50)}}$ for the inequality discounting term, because it is a commonly-used functional form that has the desired mathematical

property and meanwhile psychologically readily interpretable. We have added the following paragraphs in the Results (pp. 12–13) and Supplementary Methods (under “Additional variants for the motive cocktail model”) to clarify this:

“The inequality discounting term follows the form of a sigmoid function (Fig. S4b), which has the desired mathematical property of ensuring its value being between 0 and 1.”

“The inequality discounting term $\delta = \frac{2}{1 + e^{\eta(\text{cost}/50)}}$ follows the form of a sigmoid function (Fig. S4b), which has the desired mathematical property of ensuring the value of δ being between 0 and 1. That is, the value of δ is 1 for 0 cost and approaches 0 for high cost, while the parameter η controls the speed of this transition. Psychologically, this term can be interpreted as the probability or strength that the participant chooses to pay attention to a given transgressor-victim inequality, which, as a multiplying term for the magnitude of the inequality, modulates the effect of the latter on participants’ intervention decisions.”

We had included several variant models in the Supplement, because Prof. Ernst Fehr and several other researchers that we had discussed our work with had suggested some alternative forms to model the interaction between cost and inequality. Testing these alternative models allowed us to confirm that our initial choice of the discounting term was indeed the best fit for the empirical data, thereby strengthening the validity of our model.

With the revised manuscript augmented with the above clarification, we hope that its key modeling idea is now more accessible to the reader. Thanks for helping us to improve.

Fig. S4 | Illustration of linear and non-linear inequality discounting functions. a, Linear inequality discounting function (from the v3 model specified in the Supplementary Methods and Results), where inequality discounting is a linear function of the cost of intervention, $\delta =$

$\frac{2}{1+e^{\eta(\text{cost}/50)}}$. **b**, Non-linear inequality discounting function (v4 model, same as Model 7 in the main text) $\delta = \frac{2}{1+e^{\eta(\text{cost}/50)}}$. The x-axis represents the cost of intervention. The y-axis represents the degree of inequality discounting, where smaller values indicate stronger discounting for the victim-centered disadvantageous inequality. The parameter η controls the rate of discounting, with higher η resulting in a faster discounting of inequality with the increases in intervention cost.

Q4: Fourth, for model construction, the authors added different motives one by one in to the model in a specific sequence, and not considered all possible combinations (for example there is no SI+SCI+VCI+RP model). I know it is next to impossible to test all the combinations, but I'm wondering whether the authors have a good rationale to justify such an approach to develop the models.

Reply: We appreciate your understanding that it is impractical to test all possible combinations of motives in our modeling analysis. When developing the set of models, we introduced different motives following a descending order concerning how central and established a specific motive is in the literature of third-party punishment and helping. By progressively introducing the motives in this way, we could test whether each newly-introduced motive adds to the predictive power of the model.

In particular, among the motives we modeled, inequality aversion is the most central and established, widely used in explaining human altruistic behavior in second-party punishment, and third-party punishment and helping. Self-centered inequality aversion (SCI, including disadvantageous and advantageous inequality aversion) was proposed in Fehr and Schmidt (1999), a now-classic paper cited for more than 15000 times (according to Google Scholar), which parsimoniously explains a range of socioeconomic behaviors including second-party punishment. In the following years, Fehr and Fischbacher (2004) extend the concept of inequality aversion to incorporate the inequality between others to explain third-party punishment. Termed victim-centered inequality aversion (VCI) in our study, this variant of inequality aversion is later formulized in Zhong et al., (2016), receiving neural as well as behavioral evidence.

The motive of efficiency concern (EC) was originally proposed in the literature of resource allocation (Engelmann & Strobel, 2004). Despite that the motive itself is well-established (Hsu et al., 2008), efficiency concern alone cannot explain why third parties would scarify their own interests to reduce the payoff of others in the punishment scenario, an apparent lose-lose action that violates efficiency. That is probably why EC had not been investigated within the context of third-party punishment or help until our study. Therefore, we introduced this motive next to SCI and VCI.

The motive of reversal preference (RP) was inspired by recent findings of reversal aversion (Li et al., 2022; Xie et al., 2017) in the intervention behaviors of randomly-generated inequality. Considering that it is relatively new in the literature and also

discovered in scenarios different from classic third-party punishment or helping, we introduced RP after EC.

Last, we introduced the motive of inequality discounting, which is proposed in the present study.

We have added the following sentence on p. 12 to describe the logic behind this progressive model-building process:

“We assessed the seven socioeconomic motives’ contribution to altruistic behavior by incrementally incorporating them into utility calculations, creating a series of increasingly complex computational models. The introduction of different motives follows a descending order concerning how central and established a specific motive is in the literature of third-party punishment and helping.”

Q5: Fifth, the authors used AICc as the measure of model comparison. Their models are nested to each other, and BIC is a recommended measure for model comparison among nested models. I’m wondering why the authors used AICc, rather than BIC for model comparison.

Reply: Sorry for the lack of clarification. Both the Akaike Information Criterion (AIC, Akaike, 1974) and Bayesian Information Criterion (BIC, Schwarz, 1978) are information-theoretic criterions for model comparison, neither of which depends on whether the models to be compared are nested to each other or not (Burnham et al., 2002). We have added the following clarification into the Methods (pp. 34–35).

“We chose to use the AICc as the metric of goodness-of-fit for model comparison for the following statistical reasons. First, BIC is derived based on the assumption that the “true model” must be one of the models in the limited model set one compares (Burnham et al., 2002; Gelman & Shalizi, 2013), which is unrealistic in our case. In contrast, AIC does not rely on this unrealistic “true model” assumption and instead selects out the model that has the highest predictive power in the model set (Gelman et al., 2014). Second, AIC is also more robust than BIC for finite sample size (Vrieze, 2012).”

Minor:

I suggest the authors to include individual data points in figure 4f-h and figure s4. I cannot see any number or label on figure s5, please change the font size and enlarge the subplots.

Reply: Thank you for such detailed suggestions. We have added individual data points into Fig. 4f–h and Fig. S4 (currently Fig. S13) as suggested. Besides enlarging the font size and subplots of Fig. S5 (currently Figs. S6 & S18), we have gone through all the figures and supplementary figures to ensure the sizes of fonts and subplots are large enough to read.

Reviewers #2 and #3 (Remarks to the Author):

In this paper, the authors seek to address a gap in the literature on prosocial behavior, namely the possibility that prosocial behavior is driven by a collection of cooccurring motives. Using computational modeling, this paper is effective in untangling the influence of seven motives which drive a participant's decisions in allocating their resources to address a social injustice (a fake participant in a dictator game allocating more resources to themselves than to another fake participant). The paper also uses a clustering analysis to classify participants into groups based on their prosocial behavior patterns. Overall, we found the paradigm to be well-explained and the results to be compelling. However, we feel there are a few areas that could be clarified or discussed further.

Thank you for your constructive and insightful comments. We are pleased that you found our paradigm well-explained and our results compelling. We also appreciate your recognition of our computational modeling approach to untangling the multiple co-occurring motives driving prosocial behavior.

Q1: One major point which should be expanded upon is the broader impact of these findings. The authors make a strong case for the "motive cocktail" model's ability to describe prosocial behaviors, but the impact this has on broad, societal goals could be stronger. Why do the authors believe that identifying these multiple motives, their interactions, and broad behavioral patterns is important for tackling prosocial goals?

Reply: Thank you for highlighting the importance of addressing the broader impact of our findings. Altruism lies at the heart of human cooperative behavior, which is highlighted as one of the 125 Most Challenging Scientific Issues published by Science on its 125th anniversary, garnering widespread public interests. The motive cocktail model we developed here allows us to (1) understand the cognitive processes behind human altruistic behaviors, (2) measure individual differences, including those related to psychiatric disorders and developmental trajectories, and (3) more precisely predict behavior, guiding social policy making to foster prosocial behaviors on a societal scale.

Following your suggestions, we have enhanced our Discussion (p. 24):

"The "motive cocktail" model proposed in this study extends the economic modeling of altruistic behaviors and has important implications for understanding and promoting prosocial behavior on a societal level. By elucidating the cognitive processes underlying prosocial behavior and identifying new motives and individual differences, our model can provide insights into psychiatric disorders characterized by social dysfunction and inform future research on the neural basis of human morality and its disorders (Sanders, 2021). Our model and task framework can also be used to investigate the developmental trajectories of altruistic motives, guiding efforts to foster prosocial behaviors across life stages (Lockwood et al., 2021). By capturing the interplay of multiple motives and their impact on behavioral patterns, our model enables more precise predictions of prosocial behavior. Leveraging insights from the "motive cocktail" model,

interventions can be designed to account for the diverse motivations and experiences of individuals within society as well as cross culture background (Claessens et al., 2024), with the goal of creating a more cohesive and prosocial community. Meanwhile, further research needs to cover the gap between our over-simplified laboratory task and real-world applications."

Q2: One point that should be further discussed is the impact of reputation. While the authors make a strong case for eliminating a participant's motive to foster a good reputation, this would not entirely rule out the possibility for manipulating the reputation of transgressor and the victim in dictator game. If the real participant were to have multiple encounters with the dictator game participants, while remaining anonymous themselves, there could be an influence on how the participant allocates their resources based on observing their past actions. In other words, the current paradigm is effective in describing how people make third-party prosocial decisions when they know nothing of the other two parties other than the current situation, but in many real-world scenarios, participants may have other feelings based on observing their past behaviors. For example, if the current victim was a transgressor in the past, or vice versa, how might this impact how participants allocate resources in the current trial, etc.? While it is not possible to address this with the current design, we feel this should be discussed.

Reply: Thank you for this valuable point. Following your suggestions, we have expanded the last paragraph of the Discussion (pp. 24–25) to discuss the impact of reputation in third-party punishment and helping:

“Limitations. We used a one-shot anonymous interaction setting, a common practice in previous studies (Dawes et al., 2007; Fehr & Charness, 2023; Fehr & Fischbacher, 2004; Fehr & Gächter, 2002; FeldmanHall et al., 2014; Rockenbach & Milinski, 2006; Stallen et al., 2018; van Baar et al., 2019; Wang et al., 2024; Zhong et al., 2016), to minimize participants' concern for their own reputation, a motive that is instrumental to the long-term reciprocity in human society (Milinski et al., 2002). Consequently, our motive cocktail model, which adequately explained our data, excluded reputation as a motive. But in real-world scenarios with more interaction opportunities, reputation concern is likely to influence 3PP and 3PH behaviors (Bénabou & Tirole, 2006; Jordan et al., 2016). The victim's reputation (e.g., once a transgressor or not) also matters, with reputation-based expectancies emerging early in human development (Ting et al., 2019). Similarly, deterrence (Delton & Krasnow, 2017), reciprocity (Gintis, 2000), or social norms beyond egalitarian distribution (Kimbrough & Vostroknutov, 2016) are other real-world motives not examined here. Integrating these motives into the motive cocktail model will be topics for future research.”

Q3: The results presented pertain mainly to third-party scenarios. The authors do discuss two-party scenarios, but there is no discussion about whether or how the behavioral patterns they identify might relate to prosocial behavior in other scenarios,

such as a traditional dictator game. In other words, do the authors think that participant's behavioral pattern (justice warrior, pragmatic helper, rational moralist) might predict how they themselves allocate resources in a traditional dictator game or how it might relate to general prosociality in different scenarios?

Reply: Yes, an additional analysis shows that the three groups did differ in the resources they themselves allocated to the receiver in a traditional dictator game. (As described in the Methods, before the Intervene-or-Watch task, the participants were asked to play the roles of transgressor and victim to have first-hand experience and to familiarize themselves with the roles in the game). We now reported the result of this additional analysis on p. 17:

"Before the main experiments, we recorded the amounts participants allocated to their receiver in a dictator game. Kruskal-Wallis tests revealed significant differences across the three clusters for both Experiment 1 ($H(2) = 14.56, p < 0.001$) and Experiment 2 ($H(2) = 46.72, p < 0.001$). In both experiments, rational moralists allocated least to their receiver (see Fig. S12 for post-hoc tests)."

Fig. S12 | The amount different clusters of participants allocated to the anonymous receiver when acting as the dictator in the dictator game before the main experiment. a, Experiment 1. b, Experiment 2. Each data point (gray circle) denotes one participant. The bottom, middle, and top lines of the box plot respectively represent the first quartile, the median, and the third quartile of the data. The lines extending beyond the box refer to 1.5 times the interquartile range (IQR), i.e., the distance between the third quartile (Q3) and the first quartile (Q1). ***: $p < 0.001$ after multi-comparison corrections. DG: dictator game. J: justice warriors. P: pragmatic helpers. R: rational moralists.

Q4: Relatedly, the authors correlate individual differences (e.g., selfishness, empathy) with the motive cocktail measures but did not see whether they varied in the different behavioral patterns identified in the cluster analysis. It is worth discussing, if not

reporting, these relationships. As it stands, the individual difference analyses (line 434+) lack a bit of explanation/interpretation.

Reply: Thank you for your insightful advice. Following your suggestions, we have performed an additional analysis to compare the personality measures of the three clusters of participants and reported its results in Supplementary Fig. S13 (reproduced below). We also expanded the paragraph you referred to with the following sentences (p. 17):

"We also found significant differences between the three clusters of participants in selfishness (Kruskal-Wallis tests, Experiment 1: $H(2) = 11.70$, $p = 0.003$; Experiment 2: $H(2) = 74.02$, $p < 0.001$) and empathy concern (Experiment 1: $H(2) = 4.21$, $p = 0.122$; Experiment 2: $H(2) = 21.32$, $p < 0.001$). According to the personality questionnaires, the rational moralists were the most selfish and the justice warriors had the highest empathy (see Fig. S13 for post-hoc tests), which echoes the highest inaction inequality discounting (η_{no}) in the former and highest action inequality discounting (η_{yes}) in the latter (Fig. 4h and 4f)."

Note that in Experiment 1, though the post-hoc differences between justice warriors and the other two clusters did not reach significance (probably due to the smaller sample size compared to Experiment 2), justice warriors had the highest empathy, consistent with our findings in Experiment 2.

Fig. S13 | Differences between justice warriors (J), pragmatic helpers (P), and rational moralists (R) in the probability of accepting the intervention offer and in personality measures. For each panel, the bottom, middle, and top lines of the box plot respectively represent the first quartile, the median, and the third quartile of the data. The lines extending beyond the box refer to 1.5 times the interquartile range (IQR), i.e., the distance between the third quartile (Q3) and the first quartile (Q1). Each gray circle represents one participant. ***, ** and *: $p < 0.001$, $p < 0.01$ and $p < 0.05$ after multi-comparison corrections.

Q5: In the online replication, cluster analyses revealed 6 behavioral patterns, but the three not found in experiment 1 are largely ignored. This should be explained further, as it may be that categorizing participants into only 3 behavioral patterns is not capturing the full picture.

Reply: Apologies for the confusion. Due to space limit, in the main text we did not give a full description of the three additional clusters found in the online Experiment 2, but

had plot them in the previous Supplement. Following the reviewers' suggestions, we have moved the plots into the main text (see Fig. 5). As evident in Fig. 5 f–h (reproduced below), these participants exhibited behaviors responsive to only one stimulus dimension or even random behaviors: Participants in Cluster 4 ("scenario response") varied their choices only with the scenario, consistently choosing "yes" for the helping scenario but "no" for the punishment scenario; participants in Cluster 5 ("cost response") varied their choices only with the cost of intervention; participants in Cluster 6 ("random response") seemed to choose randomly, without responding to any variables. These patterns are clues of low effort or engaging participation, which is more frequent among online participants. Indeed, we found that these participants' choice behaviors were best described by a simple-response model, instead of any motive model we tested (see Fig. S9b). In other words, their behaviors were also inconsistent with previous research. These participants' choice behaviors probably reflected their less engagement in the experiment, as indicated by their significantly lower accuracy in attention checks during the Intervene-or-Watch task (participants in Clusters 4, 5, 6 vs. participants in Clusters 1, 2, 3: 0.93 ± 0.08 vs. 0.97 ± 0.06 ; $t(1256) = -9.78$, $p < 0.001$). Therefore, these three clusters are unlikely to represent human behaviors in the real world. That is why we chose not to discuss them in more depth.

The term "heuristic model" in our previous manuscript may also be mis-leading. We have re-named it "simple-response model" and expanded the relevant sentences in the main text (p. 17):

"The remaining 3 clusters of participants (39.10%, Fig. 5f-h) seemed to respond to one single stimulus dimension (e.g., always help but seldom punish) or even purely randomly, whose choice behaviors were best described by a simple-response model that linearly combines different independent variables (see Methods and Fig. S9b). These choice patterns likely resulted from these participants' less engaging participation (lower attention check accuracy than participants in the first three clusters: $t(1256) = -9.78$, $p < 0.001$), which is more common in online settings, rather than representing real-world behavioral patterns."

We have also included the above description about Clusters 4, 5 and 6 into the legend of Fig. 5.

Experiment 2 (N = 1258)

Fig. 5 | All major findings were replicated in the pre-registered, large-scale online Experiment 2. **a**, Model comparison results. As in Experiment 1, the full motive cocktail model best fit participants' decision behaviors, as indicated by the lowest $\Delta AICc$ and a PEP over 99.9%. **b**, Data versus model prediction. As in Experiment 1, the full model can accurately predict not only participants' average behaviors, but also that of individual clusters. **c–e**, The median value of the motive parameters for the first three clusters. The three clusters had similar behavioral patterns and parameter combinations to those of the justice warriors, pragmatic helpers, and rational moralists identified in Experiment 1. **f–h**, Data of the three clusters newly observed in Experiment 2. These three clusters were best fit by a simple-response model (Model 9) instead of by the motive cocktail model. **f**, The scenario response cluster, where participants varied their choices only with the scenario, consistently choosing "yes" for the

helping scenario but “no” for the punishment scenario. **g**, The cost response cluster, where participants varied their choices only with the cost of intervention. **h**, The random response cluster, where participants seemed to choose randomly, without responding to any variables. These patterns are clues of low effort or engaging participation, which is more frequent among online participants. Conventions follow Fig. 4.

Q6: On lines 567-569, it is argued that cost being manipulated rather than chosen by participants resembles real-world scenarios. While this is true of some scenarios (being presented with preset donation options or rounding up change at a cash register) it is not true of all scenarios. People often do need to choose how much they are willing to donate. So while the paper does describe behavior in situations in which cost is manipulated, it should be mentioned as a limitation that it is unknown how these motives or behavioral patterns would influence behavior outside of this.

Reply: Apologies for the confusion. By stating that cost being manipulated resembles real-world scenarios, we did not intend to claim this is the only way to resemble real-world scenarios. Instead, we agree with you that cost being chosen by participants themselves are also common in real-world scenarios.

When we contrasted cost as independent variable with cost as dependent variable, our focus was not that the former was more “real-world” than the latter. We just tried to explain why the three-way interaction effect involving cost seemed to have never been reported before. It is because detecting such effect would be statistically less straightforward for previous tasks with cost as dependent variable.

We have rephrased the relevant sentences to avoid such confusion (p. 23):

“In line with the joint functioning of multiple motives identified in our modeling analysis, we found a three-way interaction between cost, impact ratio, and the inequality between the transgressor and the victim. Such interaction was not reported in previous studies, probably because most studies used cost as a dependent rather than an independent variable, measuring the amount of money participants were willing to spend on the intervention, which would prevent such effects from being detected by usual statistical analysis. In contrast, the cost is manipulated by the experimenter in our task, resembling another type of real-world scenarios where individuals are confronted with limited options when it comes to addressing others’ inequalities.”

Meanwhile, we have three reasons to believe that some common cognitive mechanisms underly these two types of tasks. First, many behavioral findings observed in our Intervene-or-Watch task are consistent with findings from previous studies where participants freely choose their costs. For example, participants are more likely to help the victim than to punish the transgressor (Batistoni et al., 2022; FeldmanHall et al., 2014; Singh & Garfield, 2022; Wiessner, 2020); intervention increases with the increased inequality between the transgressor and the victim (Egas

& Riedl, 2008; Jordan et al., 2014; Stallen et al., 2018; Zhong et al., 2016) and with the increased impact-to-cost ratio of the intervention (Egas & Riedl, 2008). Second, from the decision-theoretic perspective (Luce, 1959), these two types of tasks are closely related. When people choose how much they are willing to donate, the decision-making process can be understood as first proposing options and then choosing among the options. Third, according to our out-of-sample prediction analysis (Fig. 6), the motive cocktail model estimated in our task can quantitatively predict the behavioral patterns observed in previous research with cost as dependent variable, further supporting common cognitive mechanisms behind these two types of tasks.

Q7: Lines 243-244 – “Inequality discounting refers to people’s tendency to underestimate the inequality between others as the intervention cost increases.” Would discounting not just refer to a decrease in motivation or willingness to right the inequality, as opposed to an underestimation of it? Similarly, the “rational moralists” are stated as believing that the situation is already fair. Could it not be that they just aren’t willing to incur the cost required to make it fair, unless the cost is comparatively low? This is evidenced in the thought bubble in Figure 4a reading “It’s already fair,” when an alternative could be “I’m not willing to expend the required resources to make it fair.”

Reply: We understand your concern and we apologize for not having made our reasoning more explicit. We would hesitate to interpret inequality discounting as “not willing to expend the required resources to make it fair”, for two reasons.

First, even without the inequality discounting terms, a trade-off between self-interest (SI) and victim-centered inequality aversion (VCI) is inherent in the models that consist of the SI and VCI terms (i.e., Models 4, 5, and 6), with parameter γ indicating how strongly the participant is willing to expend the required resources to reduce the inequality. However, participants’ intervention choices were better fit by Model 7 with the introduction of inequality discounting terms (Fig. 3), which suggests that inequality discounting is something beyond the lack of motivation to reduce inequality.

Second and more important, we have defined two types of inequality discounting motives: *inaction inequality discounting* (controlled by η_{no}) and *action inequality discounting* (controlled by η_{yes}), which respectively modulate the strength of VCI before and after intervention (Eqs. 14-17). Both higher η_{no} and higher η_{yes} correspond to greater discounting of VCI. Among these two different inequality discounting terms, *inaction* inequality discounting is close to the reviewers’ interpretation, with high- η_{no} participants unwilling to intervene under high cost. However, *action* inequality discounting is the opposite, where higher η_{yes} (i.e., greater discounting of the remaining inequality after intervention) means participants would be optimistic about the inequality reduction gain from the intervention when the intervention cost is high. In other words, even when the intervention cannot reduce the inequality effectively, as soon as the cost is high, high- η_{yes} participants behave as if the inequality can be successfully reduced by their intervention, thus more willing (instead of unwilling, as the reviewers’ interpretation would suggest) to intervene.

Meanwhile, we agree with the reviewers that the behavioral results of the current study may not allow us to attribute “inequality discounting” exclusively to the change of perception. To avoid confusion, we have weakened our statement by adding “to behave as if” in the referred sentence as well as revised the sentences following it to clarify our points above (p. 10):

“Inequality discounting refers to people’s tendency to behave as if they are underestimating the inequality between others as the intervention cost increases. We defined two types of inequality discounting motives: *inaction inequality discounting* (controlled by η_{no}) and *action inequality discounting* (controlled by η_{yes}), which represent a diminished awareness of inequality when choosing not to intervene and when opting to intervene, respectively. Inequality discounting motives are “compounds” that are not just the lack of motivation to reduce inequality as characterized by smaller γ (victim-centered inequality aversion), but capture the modulation of self-interest on victim-centered inequality aversion in both directions. Participants with larger η_{no} would be less likely to intervene, which differs from that of smaller γ in that it may cause no intervention even when the transgressor-victim inequality is high (Fig. 2b, row 2 right pair). Conversely, participants with larger η_{yes} would be more likely to intervene, as if they believe inequality is always minimized following a costly intervention (Fig. 2b, row 3 right pair).”

We have also gone through the manuscript to soften our tones in similar circumstances, such as on p. 5:

“We called the compound motives “inequality discounting”, which refers to people’s tendency to behave as if they are underestimating the inequality between others as the intervention cost increases.”

As to the rational moralist’s thought bubble in Fig. 4a, we are afraid we could not follow the reviewers’ exact revision suggestion, because rational moralists are characterized by both high η_{no} (inaction inequality discounting) and high η_{yes} (action inequality discounting). Though “I’m not willing to expend the required resources to make it fair” is consistent with the consequence of the high η_{no} , the phrase is incompatible with the high η_{yes} , which encourages intervention by amplifying the inequality reduction gain from the intervention. In the spirit of the reviewers’ suggestion, we have changed the phrase to “Balancing it can’t cost so much!” to capture both the inaction and action inequality discounting characteristic of rational moralists.

Reviewer #4 (Remarks to the Author):

[Note that I have previously reviewed this paper for another journal. This review has been edited from my previous referee report, taking into account the changes that have been made since the previous version of the manuscript.]

I appreciated several aspects of the paper. The expansive within-subjects experimental design is an excellent way to explore the motives underlying third-party helping and punishment. My colleagues and I have recently used a similar within-subjects design to study the motives underlying punishment, albeit on a much smaller scale (Claessens et al., pre-print). I was impressed by the scope of the computational models that are fitted and compared throughout the paper. I also appreciated the large-scale replication study, which lends further credence to the paper's claims. Finally, I would like to thank the authors for documenting the data and code and making them available for reviewers – I was able to examine the data myself and reproduce the main figures of the paper.

That said, I do have several concerns with the paper in its current form. For clarity, I have organised my concerns in line with the different sections of the paper.

Thank you for such positive and detailed feedback! We appreciate your recognition of our expansive within-subjects experimental design and its effectiveness in exploring the motives behind third-party helping and punishment. We also appreciate your acknowledgment of the scope of our computational models and the value of our large-scale replication study.

We are also grateful for your time and effort in reviewing our data and code. Thank you for your valuable suggestions and for taking time to review our work so thoroughly.

Q1: I felt that the Introduction section of the paper did not sufficiently explore alternative motives for third-party helping and punishment. This section does describe Fehr & Schmidt's (1999) utility model of inequality aversion, but it disregards other utility models based on reciprocity (Gintis, 2000), reputation (Jordan et al., 2016), and social norms (Kimbrough & Vostroknutov, 2016) on the tenuous grounds that "most 3PP or 3PH actions do not yield any direct benefit to the third party". Other motives remain unexplored, such as accounts of third-party punishment based on deterrence (Delton & Krasnow, 2017). I do not think that the current version of the paper discusses these alternative accounts in enough detail, in the introduction or the discussion sections. By discussing this diversity of accounts in more detail, the paper could make it clearer that it is focusing on only a subset of motives thought to underlie human altruism and punishment. Moreover, the introduction could be further improved by amending statements that are vague and/or unsupported by the cited evidence, such as "highlighting humans' innate aversion to inequality" (line 50) or "the utility of (in)equality is computed in the human brain" (line 56).

Reply: We appreciate your suggestion of providing a more comprehensive introduction to the alternative motives that have been proposed for third-party helping and punishment. Thank you also for referring us to (Delton & Krasnow, 2017) and your

PsyArXiv paper (Claessens et al., 2024), both of which we have read carefully and found to be very helpful.

Following your suggestions, we have re-written the second paragraph of the Introduction (p. 3):

"According to one line of theories, third-party intervention serves as a strategic means to obtain future rewards, by signaling one's trustworthiness to potential cooperators (Bénabou & Tirole, 2006; Jordan et al., 2016) or deterring potential transgressors from harming oneself or valued others (Delton & Krasnow, 2017). However, third-party intervention in one-shot, anonymous scenarios (Fehr & Fischbacher, 2004) aligns more with the strong reciprocity theory (Gintis, 2000), where individuals may reward cooperation, punish non-cooperation, or more generally, sanction violations of social norms (Claessens et al., 2024; Kimbrough & Vostroknutov, 2016) even without prospect of personal gain. These two lines of theories are not necessarily conflicting; the motives for sanctioning norm violations can be viewed as internalized external motivations. A widely observed norm in human societies is egalitarian distribution. By quantifying inequality—a violation of this norm—as a loss in a utility maximization framework, Fehr and Schmidt (1999) provide a unified explanation for various social-economic phenomena, including altruistic punishment and helping behaviors (Fehr & Fischbacher, 2004; Stallen et al., 2018; Zhong et al., 2016). Human representation of inequality is further supported by neuroimaging studies (Hsu et al., 2008; Stallen et al., 2018; Tricomi et al., 2010)."

Following your suggestions, we have also removed the clause "highlighting humans' innate aversion to inequality", and rephrased "the utility of (in)equality is computed in the human brain" for clarity (see the paragraph above).

We have also enhanced the Discussion to make it more explicit that our paper focuses on only a subset of motives underlying human altruism and punishment. Please see our reply to your Q8.

Q2: Regarding the experimental task itself, there were several details that I was unclear on. First, how frequently did participants never choose the help/punish option? The reason I ask is that in our recent study of costly punishment (Claessens et al. pre-print) a significant portion of participants (~40-50%) chose never to punish, because doing so would reduce their bonus payment. I wonder if this is quite a common behaviour in the current study but is being clustered by the algorithm into the "rational moralist" bucket. Second, how much time did the experiment take, on average? I am a little concerned that fatigue could be affecting the results, particularly in Experiment 2. Third, am I correct in thinking that all participants in Experiment 2 were students? If so, this is quite an important demographic point to make clear earlier in the paper.

Reply: These are indeed important details, which we have clarified in the revised manuscript. Below we organize our reply to your questions under the corresponding sub-headings.

The proportion of participants who never chose the help/punish option

We added a new subheading in the Supplement "The proportion of participants who never chose the help/punish option" with the following paragraphs:

"In our experiments, some participants never chose to intervene either in the punishment scenario, or in the helping scenario, or in both. Please see Table S8 for their proportions in each experiment.

As shown in Table S9, participants who neither punished nor helped were clustered into rational moralists. Those who never punished but sometimes helped were clustered into either rational moralists or pragmatic helpers, depending on their behavioral patterns in the helping scenarios."

Table S8. The proportions of participants who never chose to punish, who never chose to help, and who never chose to punish and help.

	% Never punish	% Never help	% Never punish and help
Experiment 1 (n = 157)	17.83%	8.28%	7.64%
Experiment 2 (n = 1258)	9.14%	3.66%	2.78%
Experiment 2 (simple-response participants excluded, n = 766)	12.53%	6.01%	4.57%

As now shown in Supplementary Table S8, in our experiments, there were less than 20% participants who never chose to punish. The proportion of participants who never chose to help or those who neither punished nor helped was even lower, below 10%. All these proportions were much lower than the ~40-50% reported in Claessens et al. (2024). The following differences in experimental design may have contributed to this seemingly discrepancy between our and your studies.

First, in our study, most of the time (4/5 of trials) the transgressor starts with a payoff higher than that of the victim. This transgressor-victim inequality setting is different from your study (Claessens et al., 2024), where in Games A–D, the payoff of the transgressor (P2) after stealing is the same as or still lower than that of the victim (P1). Only in Games E and F the transgressor becomes better off than the victim after stealing. The difference in inequality setting is probably also why the proportion of punishment was significantly lower in your Games A–D compared to your Games E–F.

Second, the level of inequality between the transgressor and the victim could be much larger in our study, up to 9:1 (i.e., the 90:10 inequality condition). In contrast, the inequality was at most 9:5 (i.e., Game F) in your study.

In the 50:50 and 60:40 inequality conditions of our study, where the inequality is at a similar level as in your study, the proportion of participants who never punished was also high (Experiment 1: 36.31%, Experiment 2: 18%, Experiment 2 with simple-response participants excluded: 26.89%), while this proportion was much lower in our 80:20 and 90:10 inequality conditions (Experiment 1: 17.83%, Experiment 2: 13.04%, Experiment 2 with simple-response participants excluded: 18.54%).

These findings agree with the effects of inequality aversion. As we noticed, inequality aversion also corresponds to the strongest motives observed in your study (i.e., Avoid DI and Egalitarian in Table 2). Also, as reported in Claessens et al. (2024), "Participants were also more likely to punish when targets' stealing behaviour generated inequalities, specifically in Games E and F ($b = 2.42$, $SE = 0.44$, $p < .001$)."

Another observation of note in our Table S8 is that the proportion of participants who never punished was much higher than those who never helped, which echoes our findings in the main text that participants were more likely to help than to punish (Fig. 1e and Fig. S15a).

To summarize, whether one chooses "never to punish" in a specific experiment is influenced by the experimental settings, instead of solely determined by personal traits.

You were insightful that participants who never chose to punish may be clustered into rational moralists, though the exact finding is more nuanced because we tested both punishment and helping scenarios. As now shown in Table S9, participants who neither punished nor helped were clustered into rational moralists. However, those who never punished but sometimes helped were clustered into either rational moralists or pragmatic helpers, depending on their behavioral patterns in the helping scenarios.

Table S9. The proportions of justice warriors, pragmatic helpers, and rational moralists who never chose to punish, who never chose to help, and who never chose to punish and help.

		% Never punish	% Never help	% Never punish and help
Experiment 1	Justice warriors	0%	0%	0%
	Pragmatic helpers	17.86%	0%	0%
	Rational moralists	31.08%	17.57%	16.22%

	Justice warriors	0%	0%	0%
Experiment 2	Pragmatic helpers	6.88%	0%	0%
	Rational moralists	23.82%	13.53%	10.29%

Experiment duration and fatigue

The whole experiment lasted approximately 1 hour, including the instructions, practice, main experiment and post-experiment questionnaires. The main experiment of the Intervene-or-Watch task lasted 30.86 ± 3.25 minutes for Experiment 1 and 33.97 ± 7.59 minutes for Experiment 2. The main experiment included 6 blocks, with each block lasting around 5 minutes, followed by a 30-second rest between blocks. We have added this information in the Methods on p. 27.

We had designed and analyzed the experiments in a way that may minimize the influence of practice and fatigue effects, if any. First, different conditions of inequality, cost, and impact ratio were randomly intermixed in each block; the punishment and helping blocks were randomly intermixed, whose order was counterbalanced across participants. Second, each block lasted only 5 minutes, with a 30-second rest between blocks. Third, in our GLMM1, GLMM2, LMM1, and LMM2 statistical models, we included the fixed effects of the trial number and its interactions with all the independent variables we tested, as well as the random effect of the trial number. As we noted on p. 28, "The inclusion of trial number controls for time-related confounds, such as potential fatigue or practice effects."

Demographic information of the participants

Yes, similar to Experiment 1, all the participants in our online Experiment 2 were students (student status according to Prolific), which had been reported in the Methods on p. 37:

"The criteria for participant recruitment were matched between Experiments 2 and 1, including the age ranges (18–30 years old), student status and the degree of education. In addition, the study was only accessible to participants with an approval rate of over 90% in Prolific."

These participants were recruited following the requirements stated in our pre-registration for Experiment 2 (<https://osf.io/gcsqp/>, Data Collection Procedure): "Participants will be recruited through the online labor market Prolific Academic. Participants will be paid £6-12 for agreeing to participate. The prescreen participants for this study include 1) age from 18 to 30, 2) fluent in English, 3) student status, 4) normal or corrected-to-normal vision, and 5) approval rate in Prolific from 90-100."

Following your suggestion, we have revised the sentences in the Results to make the student status information more visible (see p. 6 and p. 16):

(For Experiment 1)

"In Experiment 1, there were 157 participants (all students)."

(For Experiment 2)

"To test whether our findings can be generalized to a large population with different cultural backgrounds, we performed a pre-registered, large-scale online experiment using the same experimental procedures, with 1258 participants (all students, sample size pre-determined based on a model-based power analysis, Fig. S7) from over 60 countries (or regions, see Table S4)."

Q3: The first section of the Results describes the behavioural patterns from Experiment 1. For example, one result is that participants are less likely to intervene as the cost of the intervention increases. These findings were not at all surprising to me. Moreover, is it really true that the interaction between these variables (e.g., inequality \times cost \times ratio) has not been documented before? It was self-evident to me that these variables would have a combinatorial effect on behaviour. I'm also not sure how this interaction effect "pose[s] challenges for previous decision models that assume a linear combination of simple motives" (line 192-193). For these reasons, I think it would be better if these results (and the same results from Experiment 2) were moved to the supplement.

Reply: Thank you for these detailed suggestions. We agree with you that the statistical results of Experiment 2, which is a replication of that of Experiment 1, may be moved to the Supplement. We have done so in our revision.

However, we maintain the statistical results of Experiment 1 in the main text, for the following considerations. First, we do need "model-free" statistical reports to convince the reader that the behavioral findings in our Intervene-or-Watch task are consistent with the literature. In our task, participants decide whether to accept a specific intervention offer, instead of freely choosing the amount used for intervention as more commonly found in the literature. We understand that this forced-choice setting may appear natural to you, who used similar intervention offers in Claessens et al. (2024), but not necessarily self-evident to every reader. Reviewers #2 and #3, for example, show some doubt whether conclusions from such forced-choice task can be generalized to tasks where people freely choose the intervention cost. Statistical reports comparable to previous findings would help to convince readers like them. Second, to our knowledge, the three-way interaction of inequality \times cost \times ratio was undocumented before. In previous studies, cost was typically measured as a dependent variable by asking participants how much money they would be willing to pay to punish or help, which would not allow for a straightforward examination of how cost interacts with other variables to influence punishment or helping behaviors. Moreover, most studies either did not vary the impact ratio as an independent variable

(Fehr & Fischbacher, 2004; Stallen et al., 2018), or did not use a factorial design (Claessens et al., 2024), which also prevent such interaction being revealed. Third, presentation of these "model-free" behavioral patterns before diving into modeling analysis would make the paper more comprehensible, especially to readers who are not familiar with computational modeling.

Following your suggestion, we have removed the paragraph containing the sentence "pose[s] challenges for previous decision models that assume a linear combination of simple motives".

Q4: The next Results sections described the seven socioeconomic motives and the "motive cocktail" model. While it was good to have a clear description of the different motives, I think it also would have been useful to include some real world examples to help readers understand the motives outside the context of stylised games, e.g. Alice gives Bob less than she keeps for herself, but then Charlie intervenes to ensure that they both end up with the same amount (minimising victim-centred inequality). Real world examples like this might be especially useful for the newly-defined "inequality discounting" motives, which I admit it took me a few reads to properly understand. Since these new motives are a key contribution of the paper (all other motives are drawn from past research), it's important that these are made intuitive for readers.

Reply: Thank you for this great suggestion! We also found the descriptions of behavioral strategies in the Table 1 of Claessens et al. (2024) instructive. We have composed examples with fictitious characters, Alice, Bob and Charlie in a resource allocation game as well as some real-life scenarios to help the reader understand the motives used in the motive cocktail model. We have added these examples to the Supplement (Tables S2 and S3) and cited them in the main text (p. 8):

"Besides self-interest (the core of classical economic models), we considered five classes of computationally well-defined socioeconomic motives (Fig. 2a), which expand into seven motive terms in utility calculation (see Table S2 and Table S3 for examples in fictitious characters and real-life scenarios)."

Table S2. Fictitious examples to illustrate the motives in the motive cocktail model.

Motive	Fictitious example
Self-interest	Alice allocates resources, keeping a larger share for herself and giving a smaller portion to Bob. Charlie, a third party, observes this unequal distribution between Alice and Bob but chooses not to intervene due to the personal cost involved in taking action.

Self-centered inequality aversion	Disadvantageous inequality aversion: Alice allocates resources, keeping more for herself than she gives to Bob. Charlie, observing this, punishes Alice to ensure that Alice does not end up with more than Charlie himself. In this case, Charlie acts to minimize his own disadvantageous inequality relative to Alice. Advantageous inequality aversion: Alice allocates resources, keeping more for herself than she gives to Bob. Charlie, observing this, helps Bob to ensure that Bob does not end up with less than Charlie himself. In this case, Charlie acts to minimize his own advantageous inequality relative to Bob.
Victim-centered inequality aversion	Alice allocates resources, keeping more for herself than she gives to Bob. Charlie, observing this unequal distribution, intervenes by either punishing Alice or helping Bob, with the goal of equalizing their final outcomes. In this case, Charlie acts to minimize the disadvantageous inequality experienced by the victim, Bob.
Efficiency concern	Alice allocates resources, keeping more for herself than she gives to Bob. Charlie, observing this unequal distribution, chooses to help Bob rather than punish Alice. This action ensures that the total sum of resources for Alice and Bob increases. In this case, Charlie acts to maximize the overall payoff for others.
Reversal preference	Reversal aversion: Alice allocates resources, keeping more for herself than she gives to Bob. Charlie has an opportunity to intervene by either punishing Alice or helping Bob. However, Charlie realizes that such intervention would result in Bob having more than Alice. Charlie decides not to intervene, demonstrating reversal aversion—a preference to avoid reversing the original inequality. Reversal preference: In the same scenario, where Alice keeps more for herself than she gives to Bob, Charlie has the opportunity to intervene. Despite recognizing that intervention would result in Bob having more than Alice, Charlie chooses to intervene anyway. This demonstrates reversal preference—a willingness to create reverse inequality in the process of addressing the original imbalance.

Inaction inequality discounting	Alice allocates resources, keeping more for herself than she gives to Bob. Charlie observes this unequal distribution and has an opportunity to intervene by either punishing Alice or helping Bob. However, recognizing that such intervention would be costly for himself, Charlie chooses to disregard the disadvantageous inequality Bob is experiencing. Charlie acts as if he cannot see the inequality and decides not to intervene, effectively discounting the observed inequality to justify his inaction.
Action inequality discounting	Alice allocates resources, keeping more for herself than she gives to Bob. Charlie observes this unequal distribution and has an opportunity to intervene by either punishing Alice or helping Bob. Although Charlie recognizes that his intervention would only slightly reduce the disadvantageous inequality Bob is experiencing, and that Bob would still end up with less than Alice, he decides to take action anyway. Charlie justifies his intervention by discounting the remaining inequality, believing that his effort balances out the persisting disparity.

Table S3. Real-life examples to illustrate the motives in the motive cocktail model.

Motive	Real-life example
Self-interest	Scenario: In a community garden, volunteers are needed to help with a variety of tasks such as weeding, planting, and watering. Example: A community member notices that the garden needs attention and that there's a sign-up sheet for volunteers. Despite having free time, they decide not to sign up or participate, preferring to use their leisure time for personal activities rather than contributing to the community project.
Self-centered inequality aversion	Scenario: A company is distributing annual bonuses to its employees based on performance. Example 1 (disadvantageous inequality aversion): An employee learns that their colleague in the same role received a larger bonus. The employee appeals to management for a bonus increase to ensure they don't earn less than their peer.

Example 2 (advantageous inequality aversion): A team leader discovers they received a significantly larger bonus than their team members, despite similar contributions. The leader advocates for their team members to receive larger bonuses to reduce their own discomfort with having a much higher bonus than their peers.

Victim-centered inequality aversion

Scenario: In a small office, the manager consistently assigns the most desirable projects and clients to one team member, Mark, while giving less appealing tasks to another team member, Sarah.

Example: A third team member, Lisa, observes this pattern of unequal distribution of work. Despite not being directly affected, Lisa decides to intervene. She speaks to the manager, advocating for a more balanced distribution of projects. Lisa suggests either reassigning some of Mark's high-profile projects to Sarah or providing Sarah with additional resources and support to enhance her current projects. Lisa's primary motivation is to reduce the disadvantage experienced by Sarah, aiming to equalize opportunities and recognition between Mark and Sarah.

Efficiency concern

Scenario: During a community park cleanup event, volunteer Alex is actively picking up litter, while volunteer Sam is merely observing and giving occasional directions.

Example: A third volunteer, Jordan, notices this situation. Instead of confronting Sam about their lack of hands-on participation, Jordan chooses to assist Alex in collecting trash. Jordan's decision is motivated by the desire to maximize the overall amount of litter removed from the park. By focusing on increasing the total output of the cleanup effort rather than ensuring equal participation, Jordan prioritizes the efficiency and overall impact of the group's work.

Reversal preference

Scenario: In a small tech startup, the CEO has allocated a limited budget for employee bonuses. The senior developer, Tom, receives a significantly larger bonus than the junior developer, Emily, despite Emily having contributed crucial work to a recent successful project.

Example 1 (reversal aversion): The HR manager, Sarah, notices this disparity and has the authority to adjust the bonuses. She considers redistributing some of Tom's bonus to Emily or advocating for an increase in Emily's bonus. However, Sarah calculates that any meaningful adjustment would result in Emily's total compensation (base salary plus bonus) exceeding Tom's. Concerned about creating a reversed inequality where the junior developer earns more than the senior developer, Sarah decides not to intervene, demonstrating reversal aversion.

Example 2 (reversal preference): In the same situation, the HR manager, Mike, also notices the bonus disparity between Tom and Emily. Mike recognizes that adjusting the bonuses would likely result in Emily's total compensation surpassing Tom's. Despite this, Mike decides to intervene by recommending a significant increase to Emily's bonus, acknowledging her crucial contributions to the recent project. This action demonstrates Mike's reversal preference, as he is willing to create a reversed inequality to address the original imbalance and recognize Emily's performance.

Inaction inequality discounting

Scenario: In a sports team, the coach favors certain players over others, giving them more playtime.

Example: A teammate notices the favoritism but chooses not to speak up because challenging the coach could cost them their own playtime or position, thus ignoring the inequality.

Action inequality discounting

Scenario: In a volunteer group, one volunteer does most of the work but gets the same recognition as others.

Example: Another volunteer decides to speak up and advocate for more recognition for the hard-working volunteer, even though the overall recognition still remains somewhat unequal, believing their effort will partially balance the inequality.

Q5: When comparing the different individual difference groups ("justice warriors", "pragmatic helpers", and "rational moralists") in the next Results section, I'm not sure the Kruskal-Wallis tests, post-hoc tests, or correlations are valid because the outcome

variables are parameters that are estimated with uncertainty (e.g., α , β) rather than observed data. Do these analyses somehow account for the uncertainty in the parameter estimates?

Reply: We appreciate the reviewer's insightful observation regarding the uncertainty in parameter estimates. We understand your concern but believe that our statistical analyses for the estimated model parameters (α , β , etc.) remain valid and informative, for the following reasons.

First, all observed data, whether derived from questionnaires or behavioral tasks, inherently contain some level of uncertainty. For example, an individual may give different responses each time when answering the same question repeatedly. The uncertainty from parameter estimation is not qualitatively different from the uncertainty from observed data, when statistical tests are applied.

Second, the estimated model parameters can be viewed as sophisticated summary statistics of the observed behavioral data. They are not qualitatively different from the summary statistics that are commonly used in behavioral studies, such as the accuracy or mean response time across trials or participants.

Third, not only statistically sound, it is also a widely accepted practice in psychology and cognitive neuroscience to apply conventional statistical tests (such as t -tests) to model parameters estimated from maximum likelihood estimates like ours. For example, the Science paper of (Bahrami et al., 2010) used t -tests to compare the estimated slope parameters of different psychometric curves, and the Nature Neuroscience paper of (Behrens et al., 2007) used t -tests to compare the learning rate parameters estimated in different time periods.

Fourth, the Kruskal-Wallis test and associated post-hoc analyses are non-parametric methods, which are generally more robust to violations of distributional assumptions compared to parametric tests like t -tests.

R6: Regarding Experiment 2, the paper claims that all major findings were replicated. However, examining the raw data suggests that patterns of punishment and helping are quite different in the second study: in general, people tend to punish and help at higher levels online compared to in the lab. The cluster analysis from the second experiment also identified six clusters, rather than three. These additional clusters are disregarded as a result of the lack of engagement in online studies (lines 432-433) but is there any evidence that this is actually the case? Since around 40% of the sample fit into these clusters, they should be summarised alongside the named clusters in the main text. Moreover, the cross-cultural variation in this larger sample is not currently exploited to its full potential. It would be even more interesting to see how the frequency of different individual difference types varies across cultures (this point is hinted at in the Discussion section).

Reply: Thank you for the detailed suggestions. In the revised manuscript, we have provided additional texts, figures, and results of new analyses based on these suggestions.

Presentation of the three additional clusters

We think that people tend to punish and help at higher levels online (Experiment 2) compared to in the lab (Experiment 1) is related to the higher proportion of less engaging participants online, as unfolded below.

Due to space limit, in the main text we did not give a full description of the three additional clusters found in the online Experiment 2, but had plot them in the previous Supplement. Following the reviewers' suggestions, we have moved the plots into the main text (see Fig. 5). As evident in Fig. 5 f–h (reproduced below), these participants exhibited behaviors responsive to only one stimulus dimension or even random behaviors: Participants in Cluster 4 ("scenario response") varied their choices only with the scenario, consistently choosing "yes" for the helping scenario but "no" for the punishment scenario; participants in Cluster 5 ("cost response") varied their choices only with the cost of intervention; participants in Cluster 6 ("random response") seemed to choose randomly, without responding to any variables. These patterns are clues of low effort or engaging participation, which is more frequent among online participants. Indeed, we found that these participants' choice behaviors were best described by a simple-response model, instead of any motive model we tested. In other words, their behaviors were also inconsistent with previous research. These participants' choice behaviors probably reflected their less engagement in the experiment, as indicated by their significantly lower accuracy in attention checks during the Intervene-or-Watch task (participants in Clusters 4, 5, 6 vs. participants in Clusters 1, 2, 3: 0.93 ± 0.08 vs. 0.97 ± 0.06 ; $t(1256) = -9.78$, $p < 0.001$). Therefore, these three clusters are unlikely to represent human behaviors in the real world. That is why we chose not to discuss them in more depth.

Experiment 2 (N = 1258)

Fig. 5 | All major findings were replicated in the pre-registered, large-scale online Experiment 2. a, Model comparison results. As in Experiment 1, the full motive cocktail model best fit participants' decision behaviors, as indicated by the lowest $\Delta AICc$ and a PEP over 99.9%. **b**, Data versus model prediction. As in Experiment 1, the full model can accurately predict not only participants' average behaviors, but also that of individual clusters. **c–e**, The median value of the motive parameters for the first three clusters. The three clusters had similar behavioral patterns and parameter combinations to those of the justice warriors, pragmatic helpers, and rational moralists identified in Experiment 1. **f–h**, Data of the three clusters newly observed in Experiment 2. These three clusters were best fit by a simple-response model (Model 9) instead of by the motive cocktail model. **f**, The scenario response cluster, where participants varied their choices only with the scenario, consistently choosing "yes" for the

helping scenario but "no" for the punishment scenario. **g**, The cost response cluster, where participants varied their choices only with the cost of intervention. **h**, The random response cluster, where participants seemed to choose randomly, without responding to any variables. These patterns are clues of low effort or engaging participation, which is more frequent among online participants. Conventions follow Fig. 4.

The term "heuristic model" in our previous manuscript may also be mis-leading. We have re-named it "simple-response model" and expanded the relevant sentences in the main text (p. 17):

"The remaining 3 clusters of participants (39.10%, Fig. 5f-h) seemed to respond to one single stimulus dimension (e.g., always help but seldom punish) or even purely randomly, whose choice behaviors were best described by a simple-response model that linearly combines different independent variables (see Methods and Fig. S9b). These choice patterns likely resulted from these participants' less engaging participation (lower attention check accuracy than participants in the first three clusters: $t(1256) = -9.78$, $p < 0.001$), which is more common in online settings, rather than representing real-world behavioral patterns."

Following your suggestions, we have also added the behavioral patterns of the three additional clusters in Fig. 5, naming them as "scenario response", "cost response", and "random response". Because participants in these additional clusters were not best fit by the motive cocktail model but instead by the simple-response model, we did not present their estimated parameters or predictions from the motive cocktail model.

Cross-cultural variation

We appreciate your suggestion to explore cross-cultural variations in our sample. While this is indeed an intriguing avenue for research, there are several limitations that constrain our ability to draw robust conclusions about cultural differences. First, our primary objective in Experiment 2 was to replicate the main findings from Experiment 1. Exploring cultural differences would be beyond our pre-registered scope. Second, because we did not initially plan to study specific cultural backgrounds, we did not control for variables such as sample size from different cultures, overseas experience, or immigration status. Third, as a consequence, the sample size from different cultural groups could be imbalanced.

Despite these limitations, we recognized the value in exploring the cultural aspects of our data. We performed exploratory analyses by categorizing participants from both Experiments 1 and 2 into Eastern and Western cultural backgrounds. To ensure comparable decision-making processes across groups, we first excluded participants whose choice behaviors were best described by the simple-response model (Model 9 in the main text) that linearly combines different independent variables (see Methods and Fig. S9b). This step was necessary because the proportions of simple-response

participants differed substantially between the on-site Experiment 1 (0%) and online Experiment 2 (39.11%). We then categorized participants into Eastern and Western groups based on their countries of origin (Markus & Kitayama, 1991). To minimize the confounding effects of individuals living in different cultural areas, we excluded participants whose records spanned both Eastern and Western regions in terms of nationality, country of birth, or country of residence. For example, a participant born in China but currently holding Western nationality or living in a Western country would be excluded from this analysis. After these exclusions, our final sample consisted of 158 participants in the East group (all from Experiment 1, except for one participant) and 355 participants in the West group (all from Experiment 2). This imbalance in group sizes and experiment representation should be considered when interpreting the results. See Table S10 for detailed country distributions.

Table S10. Participants' nationality distributions in the east and west groups.

Culture	Nationality	Number of participants	Percentage
East	China	157	30.60%
	Indonesia	1	0.19%
West	Italy	102	19.88%
	Portugal	73	14.23%
	Greece	54	10.53%
	Spain	38	7.41%
	Germany	19	3.70%
	France	11	2.14%
	United Kingdom	11	2.14%
	Netherlands	10	1.95%
	Belgium	7	1.36%
	United States	6	1.17%
	Canada	5	0.97%
	Austria	5	0.97%
	Ireland	4	0.78%
	Sweden	4	0.78%
	Australia	3	0.58%
Finland	2	0.39%	
Turkey	1	0.19%	

In Supplementary Table S11, we reported the frequencies of justice warriors, pragmatic helpers, and rational moralists separately for the East and West groups. According to chi-square test of independence, the relative frequencies of the three clusters were significantly different between the two groups ($\chi^2(2) = 7.92, p = 0.019$). Following proportion difference test with Bonferroni correction, the West group had relatively more pragmatic helpers ($Z = 2.77, p = 0.017$), while the proportions of justice warriors ($Z = -1.94, p = 0.053$) and rational moralists ($Z = -0.44, p = 0.657$) did not show significant differences between the two groups. Recall that pragmatic helpers had the highest parameter of reverse preference, κ (Figs. 4 & 5). The higher proportion of pragmatic helpers in the West group thus echoes the higher κ in the West group (Fig. S21, see also our response to Reviewer #1's Q2).

We have included these results in the Supplement (newly added "Exploratory analyses on cross-cultural differences" section) and described them briefly in the main text. Please also see Tables S10–11 and Figs. S20–21 for more details.

Table S11. Participants were clustered as justice warriors, pragmatic helpers, and rational moralists in both Eastern and Western cultures.

Group	Cluster	N	Percentage
East	Justice warriors	55	34.81%
	Pragmatic helpers	28	17.72%
	Rational moralists	75	47.47%
West	Justice warriors	93	26.20%
	Pragmatic helpers	101	28.45%
	Rational moralists	161	45.35%

Fig. S21 | Comparison of motive parameters between the East and West groups, with participants combined from Experiments 1 and 2. a–g, Motives parameter comparison between the East and West groups for the parameters α , β , γ , ω , κ , η_{no} and η_{yes} . The bottom, middle, and top lines of the box plot respectively represent the first quartile, the median, and the third quartile of the data. The lines extending beyond the box refer to 1.5 times the interquartile range (IQR), i.e., the distance between the third quartile (Q3) and the first quartile (Q1). Each light-colored circle represents the parameter value estimated from an individual participant. **h–i,** The distributions of parameter α and κ (the parameters with significant group differences) in each group. Red: East group. Blue: West group. $***$ and $*$ respectively denote $p < 0.001$ and $p < 0.05$, with Bonferroni corrections for seven comparisons.

Q7: The final Results section attempts to use the motive cocktail model to make out-of-sample predictions. While I appreciate the attempt to generalise the model to other settings, I'm not sure how valid it is for a model fitted to data on third-party interactions to be used to explain behaviour in second-party settings. Some assumptions are made to justify this leap, but despite the rewrite I'm still not sure I fully understand these assumptions. I'm also not clear about why the patterns of behaviour in Fehr & Fischbacher (2004) or Stallen et al. (2018) are seen as "puzzling" in the first place. For example, participants are more likely to punish the transgressor in second-party situations when they themselves were harmed (Fig 6a). Rather than being puzzling, this behaviour makes perfect sense when we consider that punishment may serve deterrent motives to discourage harm to oneself in the future (Delton & Krasnow, 2017). Without fully explaining how the model can be validly generalised to these settings or

why these particular behavioural patterns are seen as puzzling to explain, I'm afraid that this section is more confusing than illuminating.

Reply: Apologies for the confusion. Since your review last time, we had tried to address these questions by adding an additional supplementary figure (Fig. S1) as well as adding the following sentences in the Introduction (p. 3) to clarify why the behavioral patterns of Fehr & Fischbacher (2004) or Stallen et al. (2018) are “puzzling” in our view:

“For example, when a victim seeks revenge against the transgressor, a trade-off between self-interest and inequality reduction would predict either no punishment or full punishment to restore equality, depending on whether the impact ratio of the punishment is below or above a certain threshold (see Fig. S1). But people often choose to punish the transgressor without fully restoring equality (Fehr & Fischbacher, 2004), which some researchers explain by resorting to a separate personal tendency called “willingness to punish” (Stallen et al., 2018), a factor not motivated by socio-economic utilities.”

Self-centered inequality aversion model

Fig. S1 | Simulation of the behavior of an agent following the self-centered inequality aversion model. a–b, The punishment amount as a function of the impact ratio and the inequality level between the transgressor and the victim in (a) second-party punishment (“2PP”) and (b) third-party punishment (“3PP”). **c–d,** The remaining inequality after punishment, calculated by $\max(x_1' - x_2', 0)$, as a function

of impact ratio and inequality level in (c) 2PP and (d) 3PP. The color of the lines represents inequality levels, with darker colors indicating higher inequality and lighter colors indicating lower inequality between the transgressor and the victim. The x-axis represents the impact ratio (e.g., ratio = 2 indicates that the amount of punishment reduces the transgressor's resources by twice that amount). Note that the model predicts either no punishment or full punishment to restore equality, depending on whether the impact ratio of the punishment is below or above a certain threshold.

When describing a specific phenomenon as “puzzling”, we were based on the normative framework of utility maximization, assuming a trade-off between self-interest and inequality reduction.

Following your suggestions, we have removed the word “puzzling” in the sub-heading of the out-of-sample prediction section, which is now “The motive cocktail model quantitatively reproduces a broader range of phenomena”. What we highlight now is the quantitative prediction the motive cocktail model could achieve. We have also revised this section in the following aspects to address your concerns.

First, we agree with you that the motive of deterrence (Delton & Krasnow, 2017) can explain why the intervener would spend a greater amount to penalize the transgressor when they themselves are the victim instead of the unaffected third party. This account is not necessarily conflicting with the utility maximization framework of the motive cocktail model, but instead may be integrated into the latter by, for example, assuming that the motive of deterrence may lead one to ignore the others' welfare, corresponding to less efficiency concern (parameter ω). More generally, social distance (Tang et al., 2023) and intent viciousness (Gummerum & Chu, 2014) may modulate efficiency concern.

Second, we have rephrased the description of simulation settings to make it more comprehensible.

Third, we have added sentences on how the motive cocktail model can be generalized to more real-life scenarios that involve social distance, intent viciousness, etc.

The section now reads (p. 19):

“One may wonder whether the motive cocktail estimated in participants' Intervene-or-Watch decisions is specific to this specific third-party intervention task. To demonstrate that such motive cocktail underlies human responses to inequality in general, we performed an out-of-sample prediction, using the motive cocktail model (with slight adaptations) to simulate the behavioral patterns in published studies with different experimental settings (Fehr & Fischbacher, 2004; Stallen et al., 2018). Indeed, we found that the motive cocktail model can predict the behavioral patterns in second-party punishment (2PP) as well as 3PP and 3PH (Fig. 6).

One robust phenomenon is that interveners spend more to penalize transgressors when they themselves are victims rather than unaffected third parties (i.e., 2PP > 3PP). This

can be explained by the motive of deterrence (Delton & Krasnow, 2017), which is not in conflict with our utility maximization framework. We integrate this by assuming that deterrence motives lead to reduced efficiency concern (parameter ω) in second-party situations. More broadly, ω may decrease with social distance (Tang et al., 2023) and intent viciousness (Gummerum & Chu, 2014).

In our simulations, we model second-party interveners as having all the motives of third-party interveners except efficiency concern (i.e., $\omega=0$, see Methods). Using parameters estimated from Experiment 1 participants, our model reproduces both the 2PP>3PP phenomenon and the increase in punishment with increasing inequality observed in previous laboratory experiments (Fehr & Fischbacher, 2004; Stallen et al., 2018). For both experiments, simulations with the justice warriors' parameters best matched the data.

Stallen et al. (2018) used a scenario where the first party robs the second party. The inequality here was caused by the more vicious intentions of the transgressor, thus triggering stronger third-party punishment than the same level of inequality caused by a dictator allocator (See Fig. S14). For this case, we assume even unaffected third parties have no efficiency concern, allowing our model to reproduce the less common 3PP > 3PH phenomenon they observed.”

Please also see the “Simulations to quantitatively reproduce a broader range of phenomena” section in the Method for more details about the methodology of the simulation (pp. 36–37):

“We made slight modifications to the motive cocktail model and applied it to explain the intervention patterns in second-party punishment (2PP), third-party punishment (3PP), and third-party helping (3PH) models in the following two studies. The adapted model could also be used to explain a broader range of phenomena in previous studies.

In a sub-study conducted by Fehr and Fischbacher (2004), participants attended a dictator game, which contains both 2PP condition and 3PP condition. At the beginning of the experiments, participants were randomly assigned either the role of the transgressor (Player A) or the victim (Player B). In the 2PP condition, the victim also acted as an intervener who could punish his transgressor after observing the transfer from the transgressor accordingly. In the 3PP condition, the victim could only punish the dictator in another group (Player A' and Player B'), in which he/she served as an unaffected third party. A strategy method was implemented in the 3PP condition: the third-party (Player B) had to indicate how much he would punish the outgroup Player A' for every possible transfer of A' to Player B'. The results showed that the intervener as the victim exerted more punishment than the intervener as the third-party for all transfer levels below 50 (i.e., 2PP > 3PP), while the punishment was generally low and similar across transfer levels above 50 (Fig. 6a top left). In the study conducted by Stallen et al. (2018), participants played three conditions of a justice game. In the 2PP games, the participants played the role of the partner (the victim), in which the taker (the transgressor) had the opportunity to take or rob chips (or payoff) from the victim, and afterward, the victim was given the option of punishing the transgressor by spending chips of their own. In 3PP and 3PH games, participants played the role of an observer (the third-party) to watch whether the transgressor robbed chips from the victim and then

decided whether to intervene to punish the transgressor or to compensate the victim, at their own cost. Every time participants needed to make a choice, all intervention costs ranging from 0 to 100 with a step of 10 were displayed on the screen. The results indicated the intervener in the 2PP condition punished more on the transgressor than in the 3PP condition (i.e., $2PP > 3PP$). In addition, the third-party was more likely to punish than to compensate (i.e., $3PP > 3PH$, Fig. 6b top left).

For both studies, we simulated participants' choices by calculating the utility of selecting "yes" and "no" for each inequality level using Eqs. 14–17. We assume that a second-party intervener, who themselves are also the victim, is less concerned about the overall welfare than the third-party does. As the result, the second-party intervener has all the motives a third-party intervener would have except for efficiency concern. To implement this assumption, we replaced x_3 in Eqs. 14–17 with x_2 , and set the efficiency concern ω to 0 in the second-party punishment condition. The same lack-of-efficiency-concern assumption (i.e., $\omega = 0$) was implemented during the simulation of third-party punishment and compensation games in Stallen et al. (2018). That is, we assume that the unaffected third party would ignore others' welfare in a robbery situation."

Q8: Like the Introduction section, I felt that the Discussion section could be improved by referring to the additional motives that were unable to be explored in this particular experiment, such as reciprocity, reputation, social norms, deterrence, etc. It might also be worth discussing previous work that has captured individual differences in altruistic behaviour. For example, Kurzban & Houser (2001) find evidence for different cooperative types in a public goods game setting. Are these types similar or different to the types identified in the current research? It would also be good to tighten up some of the conclusions that are drawn in this section. For example, one claim is that the identification of an action inequality discounting motive in the study aligns with theories of warm glow, but I'm not sure exactly how this follows. The paper also makes some claims about the significance of this work for real world issues (e.g. policy formation), but this attempt at generalisability doesn't seem warranted to me given the lack of ecological validity in the lab-based experimental design (see also IJzerman et al. 2020).

Reply: Thank you for your thoughtful suggestions. Motives such as reputation and deterrence, which exist in many real-world scenarios but seldomly in the one-shot, anonymous scenarios of our experiments, are beyond the scope of the present work. To acknowledge this limitation of our work more explicitly, we have added the following sentences in the Discussion (p. 24):

"Limitations. We used a one-shot anonymous interaction setting, a common practice in previous studies (Dawes et al., 2007; Fehr & Charness, 2023; Fehr & Fischbacher, 2004; Fehr & Gächter, 2002; FeldmanHall et al., 2014; Rockenbach & Milinski, 2006; Stallen et al., 2018; van Baar et al., 2019; Zhong et al., 2016), to minimize participants' concern for their own reputation, a motive that is instrumental to the long-term reciprocity in human society (Milinski et al., 2002). Consequently, our motive cocktail model, which adequately explained our data, excluded reputation as a motive. But in real-world scenarios with more interaction opportunities, reputation concern is likely to influence 3PP and 3PH behaviors (Bénabou & Tirole, 2006; Jordan et al., 2016). The victim's reputation (e.g., once a transgressor or not) also matters, with reputation-based expectancies emerging early in human development (Ting et al., 2019). Similarly, deterrence (Delton & Krasnow, 2017), reciprocity (Gintis, 2000), or social norms beyond

egalitarian distribution (Kimbrough & Vostroknutov, 2016) are other real-world motives not examined here. Integrating these motives into the motive cocktail model will be topics for future research."

Thank you for the valuable reference of Kurzban & Houser (2001). Though the available evidence does not allow us to connect our third-party intervention types to cooperative types in public goods games, we think this is an interesting future direction to explore. We have added the following sentence at the end of the Discussion (p. 25):

"Whether the three types of interveners relate to the different cooperative types found in public goods games (Kurzban & Houser, 2001), thus connecting to a larger picture of human altruistic behaviors, also deserves future research."

Following your suggestions, we have removed the paragraph about "warm glow".

We agree with you that there is still a gap between our lab-based experimental design and real-world applications. We have acknowledged this by adding the sentence (p. 24):

"Meanwhile, further research needs to cover the gap between our over-simplified laboratory task and real-world applications."

Please also see our reply to Reviewers #2 and #3's Q1, where we have discussed the broader implications of our work as they suggested and ended up with the sentence above (see p. 24 of the main text).

Overall, I thought that the paper used a novel methodology to tackle an interesting question, and the findings have implications for our understanding of third-party helping/punishment in humans. By dealing with the above concerns and perhaps editing the text in places to improve clarity (some sections and figure legends I found quite dense to read), this could be a useful addition to the literature on human altruistic behaviour.

Thank you again for your constructive comments.

Please feel free to get in contact with me if you have any questions about this review.

Review signed: Scott Claessens (scott.claessens@gmail.com)

References

- Claessens, S., Atkinson, Q., & Raihani, N. (pre-print). Why do people punish? Evidence for a range of strategic concerns. PsyArXiv. <https://doi.org/10.31234/osf.io/ys6rm>
- Delton, A. W., & Krasnow, M. M. (2017). The psychology of deterrence explains why group membership matters for third-party punishment. *Evolution and Human Behavior*, 38(6), 734-743.
- Fehr, E., & Fischbacher, U. (2004). Third-party punishment and social norms. *Evolution and Human Behavior*, 25(2), 63-87.
- Fehr, E., & Schmidt, K. M. (1999). A theory of fairness, competition, and cooperation. *The Quarterly Journal of Economics*, 114(3), 817-868.

- Gintis, H. (2000). Strong reciprocity and human sociality. *Journal of Theoretical Biology*, 206(2), 169-179.
- IJzerman, H., Lewis Jr, N. A., Przybylski, A. K., Weinstein, N., DeBruine, L., Ritchie, S. J., ... & Anvari, F. (2020). Use caution when applying behavioural science to policy. *Nature Human Behaviour*, 4(11), 1092-1094.
- Jordan, J. J., Hoffman, M., Bloom, P., & Rand, D. G. (2016). Third-party punishment as a costly signal of trustworthiness. *Nature*, 530(7591), 473-476.
- Kimbrough, E. O., & Vostroknutov, A. (2016). Norms make preferences social. *Journal of the European Economic Association*, 14(3), 608-638.
- Kurzban, R., & Houser, D. (2001). Individual differences in cooperation in a circular public goods game. *European Journal of Personality*, 15(S1), S37-S52.
- Stallen, M., Rossi, F., Heijne, A., Smidts, A., De Dreu, C. K., & Sanfey, A. G. (2018). Neurobiological mechanisms of responding to injustice. *Journal of Neuroscience*, 38(12), 2944-2954.

Reviewer #4 (Remarks on code availability):

There are README files online with metadata about the different folders/files and instructions for running the code. As stated in my review, I was able to run the code and reproduce the main figures from the paper.

Thank you!

References in the Reply

- Akaike, H. (1974). A new look at the statistical model identification. *IEEE Transactions on Automatic Control*, 19(6), 716–723. <https://doi.org/10.1109/TAC.1974.1100705>
- Bahrami, B., Olsen, K., Latham, P. E., Roepstorff, A., Rees, G., & Frith, C. D. (2010). Optimally Interacting Minds. 329.
- Batistoni, T., Barclay, P., & Raihani, N. J. (2022). Third-party punishers do not compete to be chosen as partners in an experimental game. *Proceedings of the Royal Society B: Biological Sciences*, 289(1966), 20211773. <https://doi.org/10.1098/rspb.2021.1773>
- Behrens, T. E. J., Woolrich, M. W., Walton, M. E., & Rushworth, M. F. S. (2007). Learning the value of information in an uncertain world. *NATURE NEUROSCIENCE*, 10(9).
- Bénabou, R., & Tirole, J. (2006). Incentives and Prosocial Behavior. *American Economic Review*, 96(5), 1652–1678.
- Burnham, K. P., Anderson, D. R., & Burnham, K. P. (2002). *Model selection and multimodel inference: A practical information-theoretic approach* (2nd ed). Springer.
- Claessens, S., Atkinson, Q., & Raihani, N. (2024). Why do people punish? Evidence for a range of strategic concerns. <https://doi.org/10.31234/osf.io/ys6rm>
- Dawes, C. T., Fowler, J. H., Johnson, T., McElreath, R., & Smirnov, O. (2007). Egalitarian motives in humans. *Nature*, 446(7137), 794–796. <https://doi.org/10.1038/nature05651>
- Delton, A. W., & Krasnow, M. M. (2017). The psychology of deterrence explains why group membership matters for third-party punishment. *Evolution and Human Behavior*, 38(6), 734–743. <https://doi.org/10.1016/j.evolhumbehav.2017.07.003>

- Egas, M., & Riedl, A. (2008). The economics of altruistic punishment and the maintenance of cooperation. *Proceedings of the Royal Society B: Biological Sciences*, 275(1637), 871–878. <https://doi.org/10.1098/rspb.2007.1558>
- Engelmann, D., & Strobel, M. (2004). Inequality aversion, efficiency, and maximin preferences in simple distribution experiments. *American Economic Review*, 94(4), 857–869. <https://doi.org/10.1257/0002828042002741>
- Fehr, E., & Charness, G. (2023). Social preferences: Fundamental characteristics and economic consequences. <https://doi.org/10.5167/UZH-233086>
- Fehr, E., & Fischbacher, U. (2004). Third-party punishment and social norms. *Evolution and Human Behavior*, 25(2), 63–87. [https://doi.org/10.1016/S1090-5138\(04\)00005-4](https://doi.org/10.1016/S1090-5138(04)00005-4)
- Fehr, E., & Gächter, S. (2002). Altruistic punishment in humans. *Nature*, 415(6868), Article 6868. <https://doi.org/10.1038/415137a>
- Fehr, E., & Schmidt, K. M. (1999). A theory of fairness, competition, and cooperation. *The Quarterly Journal of Economics*, 114(3), 817–868. <https://doi.org/10.1162/003355399556151>
- FeldmanHall, O., Sokol-Hessner, P., Van Bavel, J. J., & Phelps, E. A. (2014). Fairness violations elicit greater punishment on behalf of another than for oneself. *Nature Communications*, 5(1), 5306. <https://doi.org/10.1038/ncomms6306>
- Gelman, A., Hwang, J., & Vehtari, A. (2014). Understanding predictive information criteria for Bayesian models. *Statistics and Computing*, 24(6), 997–1016. <https://doi.org/10.1007/s11222-013-9416-2>
- Gelman, A., & Shalizi, C. R. (2013). Philosophy and the practice of Bayesian statistics. *British Journal of Mathematical and Statistical Psychology*, 66(1), 8–38. <https://doi.org/10.1111/j.2044-8317.2011.02037.x>
- Gintis, H. (2000). Strong Reciprocity and Human Sociality. *Journal of Theoretical Biology*, 206(2), 169–179. <https://doi.org/10.1006/jtbi.2000.2111>
- Gummerum, M., & Chu, M. T. (2014). Outcomes and intentions in children's, adolescents', and adults' second- and third-party punishment behavior. *Cognition*, 133(1), 97–103. <https://doi.org/10.1016/j.cognition.2014.06.001>
- Hsu, M., Anen, C., & Quartz, S. R. (2008). The Right and the Good: Distributive Justice and Neural Encoding of Equity and Efficiency. *Science*, 320(5879), 1092–1095. <https://doi.org/10.1126/science.1153651>
- Jordan, J. J., Hoffman, M., Bloom, P., & Rand, D. G. (2016). Third-party punishment as a costly signal of trustworthiness. *Nature*, 530(7591), 473–476. <https://doi.org/10.1038/nature16981>
- Jordan, J. J., McAuliffe, K., & Warneken, F. (2014). Development of in-group favoritism in children's third-party punishment of selfishness. *Proceedings of the National Academy of Sciences*, 111(35), 12710–12715. <https://doi.org/10.1073/pnas.1402280111>
- Kimbrough, E. O., & Vostroknutov, A. (2016). Norms make preferences social. *Journal of the European Economic Association*, 14(3), 608–638. <https://doi.org/10.1111/jeea.12152>
- Kurzban, R., & Houser, D. (2001). Individual differences in cooperation in a circular public goods game. *European Journal of Personality*, 15(1_suppl), S37–S52. <https://doi.org/10.1002/per.420>
- Li, Y., Hu, J., Ruff, C. C., & Zhou, X. (2022). Neurocomputational evidence that conflicting prosocial motives guide distributive justice. *Proceedings of the National Academy of Sciences*, 119(49), e2209078119. <https://doi.org/10.1073/pnas.2209078119>
- Lockwood, P. L., Abdurahman, A., Gabay, A. S., Drew, D., Tamm, M., Husain, M., & Apps, M. A.

- (2021). Aging increases prosocial motivation for effort. *Psychological Science*, 32(5), 668–681.
- Luce, R. D. (1959). *Individual Choice Behavior: A theoretical analysis*, New York, NY: John Wiley and Sons. Inc.
- Markus, H. R., & Kitayama, S. (1991). Cultural variation in the self-concept. In *The self: Interdisciplinary approaches* (pp. 18–48). Springer.
- Milinski, M., Semmann, D., & Krambeck, H.-J. (2002). Reputation helps solve the 'tragedy of the commons.' *Nature*, 415(6870), 424–426. <https://doi.org/10.1038/415424a>
- Rockenbach, B., & Milinski, M. (2006). The efficient interaction of indirect reciprocity and costly punishment. *Nature*, 444(7120), Article 7120. <https://doi.org/10.1038/nature05229>
- Sanders, S. (2021). *125 questions: Exploration and Discovery*. Science/AAAS Custom Publishing Office: Washington, DC, USA.
- Schwarz, G. (1978). Estimating the Dimension of a Model. *The Annals of Statistics*, 6(2). <https://doi.org/10.1214/aos/1176344136>
- Singh, M., & Garfield, Z. H. (2022). Evidence for third-party mediation but not punishment in Mentawai justice. 21.
- Stallen, M., Rossi, F., Heijne, A., Smidts, A., De Dreu, C. K. W., & Sanfey, A. G. (2018). Neurobiological Mechanisms of Responding to Injustice. *The Journal of Neuroscience*, 38(12), 2944–2954. <https://doi.org/10.1523/JNEUROSCI.1242-17.2018>
- Tang, Z., Qu, C., Hu, Y., Benistant, J., Moisan, F., Derrington, E., & Dreher, J.-C. (2023). Strengths of social ties modulate brain computations for third-party punishment. *Scientific Reports*, 13(1), 10510. <https://doi.org/10.1038/s41598-023-37286-8>
- Tricomi, E., Rangel, A., Camerer, C. F., & O'Doherty, J. P. (2010). Neural evidence for inequality-averse social preferences. *Nature*, 463(7284), 1089–1091. <https://doi.org/10.1038/nature08785>
- van Baar, J. M., Chang, L. J., & Sanfey, A. G. (2019). The computational and neural substrates of moral strategies in social decision-making. *Nature Communications*, 10(1), 1483. <https://doi.org/10.1038/s41467-019-09161-6>
- Vrieze, S. I. (2012). Model selection and psychological theory: A discussion of the differences between the Akaike information criterion (AIC) and the Bayesian information criterion (BIC). *Psychological Methods*, 17(2), 228–243. <https://doi.org/10.1037/a0027127>
- Wang, H., Wu, X., Xu, J., Zhu, R., Zhang, S., Xu, Z., Mai, X., Qin, S., & Liu, C. (2024). Acute stress during witnessing injustice shifts third-party interventions from punishing the perpetrator to helping the victim. *PLOS Biology*, 22(5), e3002195. <https://doi.org/10.1371/journal.pbio.3002195>
- Wiessner, P. (2020). The role of third parties in norm enforcement in customary courts among the Enga of Papua New Guinea. *Proceedings of the National Academy of Sciences*, 117(51), 32320–32328. <https://doi.org/10.1073/pnas.2014759117>
- Xie, W., Ho, B., Meier, S., & Zhou, X. (2017). Rank reversal aversion inhibits redistribution across societies. *Nature Human Behaviour*, 1(8), 0142. <https://doi.org/10.1038/s41562-017-0142>
- Zhong, S., Chark, R., Hsu, M., & Chew, S. H. (2016). Computational substrates of social norm enforcement by unaffected third parties. *NeuroImage*, 129, 95–104. <https://doi.org/10.1016/j.neuroimage.2016.01.040>

Decision Letter, first revision:

Date: 25th July 24 09:46:16
Last Sent: 25th July 24 09:46:16
Triggered By: Fernando Chirigati
From: fernando.chirigati@us.nature.com
To: hang.zhang@pku.edu.cn
CC: computacionalscience@nature.com
BCC: fernando.chirigati@us.nature.com
Subject: AIP Decision on Manuscript NATCOMPUTSCI-24-0738A
Message: Our ref: NATCOMPUTSCI-24-0738A

25th July 2024

Dear Dr. Zhang,

Thank you for submitting your revised manuscript "The "motive cocktail" in altruistic behaviors" (NATCOMPUTSCI-24-0738A). It has now been seen by the original referees and their comments are below. The reviewers find that the paper has improved in revision, and therefore we'll be happy in principle to publish it in Nature Computational Science, pending minor revisions to satisfy the referees' final requests and to comply with our editorial and formatting guidelines.

We are now performing detailed checks on your paper and will send you a checklist detailing our editorial and formatting requirements in about 3 days. Please do not upload the final materials and make any revisions until you receive this additional information from us. We will ask you for a quick turnaround, if possible, so we can include your paper in our upcoming September issue.

TRANSPARENT PEER REVIEW

Nature Computational Science offers a transparent peer review option for original research manuscripts. We encourage increased transparency in peer review by publishing the reviewer comments, author rebuttal letters and editorial decision letters if the authors agree. Such peer review material is made available as a supplementary peer review file. **Please remember to choose, using the manuscript system, whether or not**

you want to participate in transparent peer review.

Please note: we allow redactions to authors' rebuttal and reviewer comments in the interest of confidentiality. If you are concerned about the release of confidential data, please let us know specifically what information you would like to have removed. Please note that we cannot incorporate redactions for any other reasons. Reviewer names will be published in the peer review files if the reviewer signed the comments to authors, or if reviewers explicitly agree to release their name. For more information, please refer to our FAQ page.

Thank you again for your interest in Nature Computational Science. Please do not hesitate to contact me if you have any questions.

Sincerely,
Fernando

--

Fernando Chirigati, PhD
Chief Editor, Nature Computational Science
Nature Portfolio

ORCID

Reviewer #1 (Remarks to the Author):

The authors have addressed all my concerns very well. Congrats the authors for this great paper!

Reviewer #2 and #3 (Remarks to the Author):

The authors have been very responsive to the reviews. We appreciated the response to our Q1 and found the description of the three things the motive cocktail model allowed

the researchers to accomplish to be very clear. We suggest that this sentence be included in the revised manuscript, in addition to the text that has already been added.

Author Rebuttal, first revision:

Final Decision Letter:

Date: 7th August 24 11:42:13
Last Sent: 7th August 24 11:42:13
Triggered By: Fernando Chirigati
From: fernando.chirigati@us.nature.com
To: hang.zhang@pku.edu.cn
BCC: computacionalscience@nature.com,rjsproduction@springernature.com,fernando.chirigati@us.nature.com
Subject: Decision on Nature Computational Science manuscript NATCOMPUTSCI-24-0738B
Message: Dear Dr Zhang,

We are pleased to inform you that your Article "The "motive cocktail" in altruistic behaviors" has now been accepted for publication in Nature Computational Science.

Once your manuscript is typeset, you will receive an email with a link to choose the appropriate publishing options for your paper and our Author Services team will be in touch regarding any additional information that may be required.

Please note that *Nature Computational Science* is a Transformative Journal (TJ). Authors may publish their research with us through the traditional subscription access route or make their paper immediately open access through payment of an article-processing charge (APC). Authors will not be required to make a final decision about access to their article until it has been accepted. Find out more about Transformative Journals

Authors may need to take specific actions to achieve compliance with funder and institutional open access mandates. If your research is supported by a funder that requires

immediate open access (e.g. according to Plan S principles) then you should select the gold OA route, and we will direct you to the compliant route where possible. For authors selecting the subscription publication route, the journal's standard licensing terms will need to be accepted, including self-archiving policies. Those licensing terms will supersede any other terms that the author or any third party may assert apply to any version of the manuscript.

Acceptance of your manuscript is conditional on all authors' agreement with our publication policies (see <https://www.nature.com/natcomputsci/for-authors>). In particular your manuscript must not be published elsewhere and there must be no announcement of the work to any media outlet until the publication date (the day on which it is uploaded onto our web site).

Before your manuscript is typeset, we will edit the text to ensure it is intelligible to our wide readership and conforms to house style. We look particularly carefully at the titles of all papers to ensure that they are relatively brief and understandable.

Once your manuscript is typeset, you will receive a link to your electronic proof via email with a request to make any corrections within 48 hours. If, when you receive your proof, you cannot meet this deadline, please inform us at rjsproduction@springernature.com immediately.

If you have queries at any point during the production process then please contact the production team at rjsproduction@springernature.com.

You may wish to make your media relations office aware of your accepted publication, in case they consider it appropriate to organize some internal or external publicity. Once your paper has been scheduled you will receive an email confirming the publication details. This is normally 3-4 working days in advance of publication. If you need additional notice of the date and time of publication, please let the production team know when you receive the proof of your article to ensure there is sufficient time to coordinate. Further information on our embargo policies can be found here:

<https://www.nature.com/authors/policies/embargo.html>

An online order form for reprints of your paper is available at <https://www.nature.com/reprints/author-reprints.html>. All co-authors, authors'

institutions and authors' funding agencies can order reprints using the form appropriate to their geographical region.

Best regards,
Fernando

--

Fernando Chirigati, PhD
Chief Editor, Nature Computational Science
Nature Portfolio

P.S. Click on the following link if you would like to recommend Nature Computational Science to your librarian: <https://www.springernature.com/gp/librarians/recommend-to-your-library>

** Visit the Springer Nature Editorial and Publishing website at www.springernature.com/editorial-and-publishing-jobs for more information about our career opportunities. If you have any questions please click here.**